# FastSurvival: Hidden Computational Blessings in Training Cox Proportional Hazards Models

**Jiachang Liu**[1], **Rui Zhang**[2], **Cynthia Rudin**[2]
[1]Cornell University, [2]Duke University
jiachang.liu@cornell.edu, r.zhang@duke.edu, cynthia@cs.duke.edu

## Abstract

Survival analysis is an important research topic with applications in healthcare, business, and manufacturing. One essential tool in this area is the Cox proportional hazards (CPH) model, which is widely used for its interpretability, flexibility, and predictive performance. However, for modern data science challenges such as high dimensionality (both $n$ and $p$) and high feature correlations, current algorithms to train the CPH model have drawbacks, preventing us from using the CPH model at its full potential. The root cause is that the current algorithms, based on the Newton method, have trouble converging due to vanishing second order derivatives when outside the local region of the minimizer. To circumvent this problem, we propose new optimization methods by constructing and minimizing surrogate functions that exploit hidden mathematical structures of the CPH model. Our new methods are easy to implement and ensure monotonic loss decrease and global convergence. Empirically, we verify the computational efficiency of our methods. As a direct application, we show how our optimization methods can be used to solve the cardinality-constrained CPH problem, producing very sparse high-quality models that were not previously practical to construct. We list several extensions that our breakthrough enables, including optimization opportunities, theoretical questions on CPH's mathematical structure, as well as other CPH-related applications.

## 1 Introduction

Survival analysis, which studies time-to-event data, is an important research topic with a wide range of real-world applications. In medicine, survival analysis has been employed to model when a patient will die [44, 38, 10]. In business, it is useful for attrition prediction [30] (when an employees resigns) and churn prediction [31] (when a customer unsubscribes), and in manufacturing, it is used to predict when a physical system breaks down [42, 50]. A fundamental tool in analyzing such data is the Cox proportional hazards (CPH) model [8], a linear model under the assumption that features have a multiplicative effect on the risk of failure/event. Simple yet powerful, the CPH model has enjoyed great popularity due to its modeling flexibility (when coupled with additive models [27, 28, 67, 11, 1]). Moreover, in contrast to black box models, it is both interpretable and accurate.

However, with the advent of larger sample and feature spaces and more complex data, new challenges arise in using the CPH model to its full potential. Ideally, practitioners want to produce CPH models repeatedly, with feature engineering and preprocessing between iterations. Additionally, they want the CPH model to identify important variables [64, 14], even in presence of highly correlated features. However, current optimization methods for training the CPH model do not meet these needs. Current algorithms [62, 22, 23, 54], based on the generic Newton's method, are computationally intensive. More importantly, due to both vanishing second-order derivatives [53] and the use of approximation strategies that trade precision for efficiency, existing optimization methods have trouble converging, either with the loss blowing up or the algorithm converging very slowly when we require the precision

necessary to handle correlated variables. The latter issue is the core reason for incorrect variable selection when features are highly correlated.

In this work, we propose new optimization methods to train the CPH model and show that there is not necessarily a precision-efficiency tradeoff. Despite the CPH model being seemingly amenable to classical optimization approaches such as coordinate descent, this has not been attempted for the original CPH loss function due to its daunting complexity. However, through careful examination, we show instead that the complexity of the CPH loss function is really a blessing, rather than a curse. We discover hidden mathematical structures that allow us to design very efficient algorithms. We show both the first and second-order derivatives at each coordinate can be computed exactly in linear time complexity ($O(n)$). Moreover, we show both derivatives are Lipschitz-continuous by making novel connections with the second and third central moment calculation in probability theory and statistics. All these discoveries lead us to design algorithms that essentially minimize a quadratic surrogate function and a cubic surrogate function, respectively. They are extremely easy to implement.

Empirically, we demonstrate the superior speed of our algorithms on large-scale datasets. In general, ours are significantly faster than all existing methods and rapidly converge to optimal high-precision solutions. Because our methods produce high-quality solutions, we apply them for variable selection in challenging regimes where features are highly correlated. We solve difficult cardinality-constrained CPH problems and produce models that are much sparser than the state-of-the-art methods.

In summary, our contributions are: (1) We find a *critical flaw* in the current optimization algorithms for the CPH method by pinpointing that they converge slowly with low precision. Sometimes, the loss does not converge and explodes. (2) To circumvent this issue, we propose novel algorithms that minimize a quadratic and a cubic surrogate function, respectively, with guaranteed convergence and loss descent at each iteration. The core novelty lies in discovering hidden mathematical structure, which allows for an efficiency way ($O(n)$) of calculating the second-order partial derivatives exactly. In addition, we show the first and second order partial derivatives are Lipschitz continuous. To calculate these Lipschitz constants, we leverage second and third central moments from theoretical statistics and probability theory. (3) Empirically, our method enjoys *fast speed* in training the loss function and results in *superior performance* when solving cardinality-constrained problems.

Our work constitutes a methodological breakthrough in training CPH models. At the end of the paper, we also discuss several exciting extensions and follow-up questions, showing how our new perspectives and discoveries open doors to many new research opportunities.

## 2 Preliminaries

Given a time-to-event dataset of $n$ samples with $\{\boldsymbol{x}_i, t_i, \delta_i\}_{i=1}^n$, where $\boldsymbol{x} \in \mathbb{R}^p$ is the feature vector with length $p$, $t_i \in \mathbb{R}$ is the observation time, and $\delta_i \in \{0, 1\}$ is an indicator with 1 indicating that a failure event has happened, the CPH model can be used to learn and predict the risk of failure, commonly known as the hazard function in survival analysis. The CPH model predicts the hazard $h_i(t)$ for sample $i$ in a semiparametric way [62]. For review of related work, please see Appendix B.

$$h_i(t) = h_0(t)e^{\boldsymbol{x}_i^T \boldsymbol{\beta}}, \tag{1}$$

where $h_0(t)$ is a baseline hazard function shared by all samples, and $\boldsymbol{\beta} \in \mathbb{R}^p$ is the parameter of interest. The nice thing about the CPH model is that $h_0(t)$ cancels out if we look only at the ratio of hazards of sample $i$ vs. all remaining samples at time $t_i$, *i.e.*,

$$\frac{h_i(t_i)}{\sum_{j \in R_i} h_j(t_i)} = \frac{e^{\boldsymbol{x}_i^T \boldsymbol{\beta}}}{\sum_{j \in R_i} e^{\boldsymbol{x}_j^T \boldsymbol{\beta}}}, \tag{2}$$

where $R_i := \{j \mid t_j \geq t_i\}$ is the set of indices whose observation time is greater than or equal to that of sample $i$. Such ratios are also called partial likelihoods. To estimate the parameter of interest, $\boldsymbol{\beta}$, we maximize the joint partial likelihood of all samples with failure events, which can be written as

$$L(\boldsymbol{\beta}) = \Pi_{i|\delta_i=1} \frac{e^{\boldsymbol{x}_i^T \boldsymbol{\beta}}}{\sum_{j \in R_i} e^{\boldsymbol{x}_j^T \boldsymbol{\beta}}}. \tag{3}$$

This is equivalent to minimizing the negative log partial likelihood [62], which is defined as

$$\ell(\boldsymbol{\beta}) = -\log L(\boldsymbol{\beta}) = \sum_{i=1}^{n} \delta_i \left[ \log \left( \sum_{j \in R_i} e^{\boldsymbol{x}_j^T \boldsymbol{\beta}} \right) - \boldsymbol{x}_i^T \boldsymbol{\beta} \right]. \tag{4}$$

The loss function $\ell(\boldsymbol{\beta})$, while convex, is very mathematically involved. In addition to the double sum, the inner summation over $j$ is with respect to a different index set $R_i$, for each outer summation index $i$. Such daunting complexity makes it difficult to employ first-order methods such as gradient descent because we cannot easily pick the right step size for each iteration, which plays a crucial role in practical running time. Therefore, past efforts have been focused on developing Newton-type (second-order) methods, where the loss function is approximated by a second-order Taylor expansion:

$$\ell(\boldsymbol{\beta} + \Delta\boldsymbol{\beta}) \approx \ell(\boldsymbol{\beta}) + \nabla_{\boldsymbol{\beta}}\ell(\boldsymbol{\beta})^T \Delta\boldsymbol{\beta} + \frac{1}{2}\Delta\boldsymbol{\beta}^T \nabla_{\boldsymbol{\beta}}^2 \ell(\boldsymbol{\beta})\Delta\boldsymbol{\beta} := f(\Delta\boldsymbol{\beta}). \tag{5}$$

The function $f(\Delta\boldsymbol{\beta})$ can be minimized by solving a linear system: $\Delta\hat{\boldsymbol{\beta}} = -(\nabla_{\boldsymbol{\beta}}^2\ell(\boldsymbol{\beta}))^{-1}\nabla_{\boldsymbol{\beta}}\ell(\boldsymbol{\beta})$. To reveal the computational nuances more explicitly, we use an intermediate variable $\boldsymbol{\eta}$ with $\boldsymbol{\eta} = \boldsymbol{X}\boldsymbol{\beta}$. Then we can rewrite the approximation function $f(\Delta\boldsymbol{\beta})$ as:

$$f(\Delta\boldsymbol{\beta}) = \ell(\boldsymbol{\beta} + \Delta\boldsymbol{\beta}) \approx \ell(\boldsymbol{\eta}) + \nabla_{\boldsymbol{\eta}}\ell(\boldsymbol{\eta})^T \boldsymbol{X}\Delta\boldsymbol{\beta} + \frac{1}{2}\Delta\boldsymbol{\beta}^T \boldsymbol{X}^T \nabla_{\boldsymbol{\eta}}^2\ell(\boldsymbol{\eta})\boldsymbol{X}\Delta\boldsymbol{\beta}. \tag{6}$$

At each iteration, calculating the Hessian matrix $\nabla_{\boldsymbol{\eta}}^2\ell(\boldsymbol{\eta})$ requires $O(n^2)$ complexity. Past methods on the CPH model have resorted to various approximation strategies by replacing $\nabla_{\boldsymbol{\eta}}^2\ell(\boldsymbol{\eta})$ with $H(\boldsymbol{\eta})$ to reduce the computational complexity:

**1. Exact Newton** $\quad H(\boldsymbol{\eta}) = \nabla_{\boldsymbol{\eta}}^2\ell(\boldsymbol{\eta}) \qquad\qquad\qquad\qquad\qquad$ *# no approximation*

**2. Quasi Newton** $\quad H(\boldsymbol{\eta})_{ij} = \begin{cases} [\nabla_{\boldsymbol{\eta}}^2\ell(\boldsymbol{\eta})]_{ii} & \text{if } i = j \\ 0 & \text{otherwise} \end{cases} \quad$ *# ignore off-diagonal terms*

**3. Proximal Newton** $\quad H(\boldsymbol{\eta}) = \text{diag}(\nabla_{\boldsymbol{\eta}}\ell(\boldsymbol{\eta}) + \boldsymbol{\delta}), \qquad$ *# diagonal upper bound on $\nabla_{\boldsymbol{\eta}}^2\ell(\boldsymbol{\eta})$*

where $\text{diag}(\cdot)$ constructs a matrix with its diagonal equal to the input vector and other entries equal to 0. There are two major problems with the above approaches. One common problem is that the these Newton-type methods inherently have trouble converging beyond the local region of minimizers without backtrack line search [53]. We provide a concrete example to demonstrate this issue in the experiment section. Ideally, we want to avoid backtracking because this increases the running time. In contrast, our methods do not have this flaw and guarantee global convergence.

The other problem is that when the above approaches do converge to the optimal solutions, none of them can converge with high precision fast enough (in a practical sense). The exact Newton's method [22] has a local quadratic convergence rate, but each iteration can take a long time. Quasi Newton [62] and proximal Newton [51] [1] methods are computationally much cheaper to evaluate per iteration, but they make less progress toward the optimal solution. In the next section, we show that, by exploiting hidden mathematical structure, we can obtain the *best of both worlds*: cheap evaluation per iteration and fast convergence with respect to the number of iterations.

## 3 Methodology

### 3.1 New Formulas for First, Second, and Third Order Partial Derivatives

As we have mentioned, the reason for the diagonal approximations of $\nabla_{\boldsymbol{\eta}}^2\ell(\boldsymbol{\eta})$ is to reduce the complexity of the mathematics and associated high computational cost. Here, we take a completely different approach from past methods. First, we avoid making any approximations and embrace the full Hessian matrix. Second, we bypass the intermediate step of calculating the Hessian in the sample space $\boldsymbol{\eta}$ and focus on the Hessian in the feature space $\boldsymbol{\beta}$. The involved mathematics may already sound complicated, but we do not stop here. We apply these two ideas not only to the second order partial derivatives but the third order partial derivatives as well. Although this seems like a burdensome task, we show that the end result is very elegant and has an intuitive interpretation. Out of complexity comes simplicity. We summarize the relevant results in the first following theorem:

---

[1]See the skglm tutorial at `https://contrib.scikit-learn.org/skglm/tutorials/cox_datafit.html#maths-cox-datafit`.

**Theorem 3.1.** *For the CPH loss function defined in Equation* (4)*, the first, second, and third order partial derivatives with respect to coordinate l are:*

*1st order partial derivative:*

$$\frac{\partial \ell(\boldsymbol{\beta})}{\partial \beta_l} = \sum_{i=1}^{n} \delta_i \left( \sum_{k \in R_i} \frac{e^{\eta_k}}{\sum_{j \in R_i} e^{\eta_j}} X_{kl} \right) - \sum_{i=1}^{n} \delta_i X_{il}. \tag{7}$$

*2nd order partial derivative:*

$$\frac{\partial^2 \ell(\boldsymbol{\beta})}{\partial \beta_l^2} = \sum_{i=1}^{n} \delta_i \left[ \sum_{k \in R_i} \frac{e^{\eta_k}}{\sum_{j \in R_i} e^{\eta_j}} X_{kl}^2 - \left( \sum_{k \in R_i} \frac{e^{\eta_k}}{\sum_{j \in R_i} e^{\eta_j}} X_{kl} \right)^2 \right]. \tag{8}$$

*3rd order partial derivative:*

$$\frac{\partial^3 \ell(\boldsymbol{\beta})}{\partial \beta_l^3} = \sum_{i=1}^{n} \delta_i \left[ \sum_{k \in R_i} \frac{e^{\eta_k}}{\sum_{j \in R_i} e^{\eta_j}} X_{kl}^3 + 2 \left( \sum_{k \in R_i} \frac{e^{\eta_k}}{\sum_{j \in R_i} e^{\eta_j}} X_{kl} \right)^3 \right.$$
$$\left. -3 \left( \sum_{k \in R_i} \frac{e^{\eta_k}}{\sum_{j \in R_i} e^{\eta_j}} X_{kl}^2 \right) \left( \sum_{k \in R_i} \frac{e^{\eta_k}}{\sum_{j \in R_i} e^{\eta_j}} X_{kl} \right) \right]. \tag{9}$$

The proof can be found in Appendix A. The first, second, and third order partial derivatives all have a probabilistic interpretation. Notice that for any $i$, the coefficients in front of $X_{kl}$, $X_{kl}^2$, and $X_{kl}^3$ are nonnegative and sum up to 1, *i.e.*, $e^{\eta_k}/(\sum_{j \in R_i} e^{\eta_j}) \geq 0$ and $\sum_{k \in R_i} [e^{\eta_k}/(\sum_{j \in R_i} e^{\eta_j})] = 1$. Then, we can regard these coefficients as a discrete probability distribution. Thus, for Equation (8), the term inside $[\cdot]$ resembles the variance or second order central moment formula: $\mathbb{E}[X^2] - (\mathbb{E}[X])^2 = E[(X - \mathbb{E}[X])^2]$. For Equation (9), the term inside $[\cdot]$ resembles the skewness or third order central moment formula: $\mathbb{E}[X^3] + 2(\mathbb{E}[X])^3 - 3\mathbb{E}[X^2]\mathbb{E}[X] = \mathbb{E}[(X - \mathbb{E}[X])^3]$.

One may wonder whether for higher orders (order $r \geq 4$), the relationship between the $r$-th order partial derivative and $r$-th central moment still preserve. The answer is no and this can be easily deduced from the following lemma. The proof can be found in Appendix A.

**Lemma 3.2.** *Let us define $C_r$ to be the $r$-th central moment with*

$$C_r := \sum_{k \in R_i} \frac{e^{\eta_k}}{\sum_{j \in R_i} e^{\eta_j}} \left( X_{kl} - \sum_{k_1 \in R_i} \frac{e^{\eta_{k_1}}}{\sum_{j \in R_i} e^{\eta_j}} X_{k_1 l} \right)^r. \tag{10}$$

*Then we can calculate the partial derivative of $C_r$ with respect to $\beta_l$ as:*

$$\frac{\partial}{\partial \beta_l} \left( C_r \right) = C_{r+1} - r \cdot C_2 \cdot C_{r-1}. \tag{11}$$

From Lemma 3.2, we can see why the connection to central moment does not work for higher order partial derivatives. If $r = 2$, the second term in Equation (11) disappears, *i.e.*, $C_{r-1} = C_1 = 0$. Therefore, we get $\partial C_2 / \partial \beta_l = C_3$. However, for $r \geq 3$, $C_{r-1}$ in general is not zero, so we cannot extrapolate this pattern to higher order partial derivatives.

Theorem 3.1 forms the basis upon which we build everything else. These results are not only mathematically interesting but also have significant implications for computation, which we elaborate in the next two sections.

## 3.2 Time Complexity of First and Second Order Partial Derivative Calculation

From the connections to the second and third central moment, we have the following corollary regarding the time complexity of calculating the first, second, and third order derivatives:

**Corollary 3.3.** *For the CPH model, the time complexities to calculate $\frac{\partial \ell(\boldsymbol{\beta})}{\partial \beta_l}$ and $\frac{\partial^2 \ell(\boldsymbol{\beta})}{\partial \beta_l^2}$ are $O(n)$ .*

This is a surprising result, especially for the second order partial derivatives. The intermediate Hessian, $\nabla_{\boldsymbol{\eta}}^2 \ell(\boldsymbol{\eta})$, takes $O(n^2)$ to compute, so we would expect the second order partial derivative, $\frac{\partial^2 \ell(\boldsymbol{\beta})}{\partial \beta_j^2} = \boldsymbol{e}_j^T \boldsymbol{X}^T \nabla_{\boldsymbol{\eta}}^2 \ell(\boldsymbol{\eta}) \boldsymbol{X} \boldsymbol{e}_j$, would take $O(n^2)$ to compute as well. Yet, the time complexity is just $O(n)$. We use the first order partial derivative formula, Equation (7), as an example to explain why this happens. We ignore the second term $\sum_{i=1}^n \delta_i X_{il}$ because it is just a constant. The first term in Equation (7) can be rewritten as:

$$\sum_{i=1}^n \delta_i \left( \sum_{k_1 \in R_i} \frac{e^{\eta_{k_1}}}{\sum_{j \in R_i} e^{\eta_j}} X_{k_1 l} \right) = \sum_{i=1}^n \delta_i \left( \frac{\sum_{k_1 \in R_i} e^{\eta_{k_1}} X_{k_1 l}}{\sum_{j \in R_i} e^{\eta_j}} \right). \tag{12}$$

Let us focus on the numerator inside the parenthesis for now. For the entire sequence ($i = 1, 2, ..., n$) of numerator terms, we can obtain all of them together at the cost of $O(n)$ by performing reverse cumulative summation. The same is true when we obtain the entire sequence of denominators. Once we have all numerators and denominators, calculating the entire sequence of ratios inside the parenthesis also costs $O(n)$. Finally, multiplying each ratio with $\delta_i$ and summing up all these products costs $O(n)$ as well. Therefore, the computational cost to calculate the first order partial derivative is $O(n)$. We can apply the same idea to the second order partial derivative formula, Equation 8, to show that the computational complexity is also $O(n)$. Note that the reverse cumulative summation trick has already been explored in [62] for calculating the diagonal of $\nabla_{\boldsymbol{\eta}}^2 \ell(\boldsymbol{\eta})$ in the sample space $\boldsymbol{\eta}$, but this trick has not been used to calculate the partial derivatives in the feature space $\boldsymbol{\beta}$.

We will later see how this $O(n)$ time complexity allows us to design a second order optimization method whose evaluation cost per iteration is just as cheap as a first order optimization method. Before we discuss that, let us continue and discuss another computational implication of Theorem 3.1.

## 3.3 Lipschitz-Continuity Property of First and Second Order Partial Derivatives

The connection to the central moment calculation allows us to conclude that the first and second order partial derivatives are Lipschitz-continuous. Moreover, we can calculate these Lipschitz constants explicitly. Recall that for a univariate function $f(x)$, we say that the function is Lipschitz-continuous [5] if there exists $L \geq 0$ such that for any two points in the domain of $f$, i.e., $x, y \in \mathcal{D}(f)$, we have $|f(x) - f(y)| \leq L|x - y|$. The value $L$ is called the Lipschitz constant for function $f(\cdot)$. If the function is continuously differentiable, the previous definition is equivalent to the condition where the first order derivative is bounded, $|f'(x)| \leq L$ for any $x \in \mathbb{R}$ [5].

Not only can we say that the first and second order partial derivatives are Lipschitz-continuous, but we can also calculate the Lipschitz constants explicitly. We summarize the results in the theorem below. The proof can be found in Appendix A

**Theorem 3.4.** *For the second order partial derivatives in Equation* (8)*, its absolute values are bounded by the following formula:*

$$0 \leq \frac{\partial^2 \ell(\boldsymbol{\eta})}{\partial \beta_l^2} \leq \frac{1}{4} \sum_{i=1}^n \delta_i \left( \max_{k_1 \in R_i} X_{k_1 l} - \min_{k_1 \in R_i} X_{k_1 l} \right)^2 \tag{13}$$

*For the third order partial derivatives in Equation* (9)*, its absolute values are bounded by the following formula:*

$$\left| \frac{\partial^3 \ell(\boldsymbol{\eta})}{\partial \beta_l^3} \right| \leq \frac{1}{6\sqrt{3}} \sum_{i=1}^n \delta_i \left| \max_{k_1 \in R_1} X_{k_1 l} - \min_{k_1 \in R} X_{k_1 l} \right|^3 \tag{14}$$

The availability of these Lipschitz constants suggests that we might construct surrogate functions.

## 3.4 Quadratic and Cubic Surrogate Functions

We now have all the tools at hand to attack the original optimization problem. For a univariate convex $f(x)$, if we have access to $L_2$, the Lipschitz constant for its first order derivative, then we can construct the following quadratic surrogate function $g_x(\cdot)$ [53]:

$$f(x + \Delta x) \leq f(x) + f'(x)\Delta x + \frac{1}{2} L_2 \Delta x^2 =: g_x(\Delta x). \tag{15}$$

If we have access to $L_3$, the Lipschitz constant for its second order derivative, then we can construct the following cubic surrogate function $h_x(\cdot)$ [53]:

$$f(x + \Delta x) \leq f(x) + f'(x)\Delta x + \frac{1}{2}f''(x)\Delta x^2 + \frac{1}{6}L_3|\Delta x|^3 =: h_x(\Delta x) \tag{16}$$

A nice thing about these surrogate functions is that their minimizers can be computed analytically:

$$\underset{\Delta x}{\operatorname{argmin}}\, g_x(\Delta x) = -\frac{1}{L_2}f'(x) \tag{17}$$

$$\underset{\Delta x}{\operatorname{argmin}}\, h_x(\Delta x) = \operatorname{sgn}(f'(x)) \cdot \frac{f''(x) - \sqrt{(f''(x))^2 + 2L_3|f'(x)|}}{L_3}, \tag{18}$$

where the function $\operatorname{sgn}(\cdot)$ extracts the sign ($+$ or $-$) of the input. The analytical solution to the quadratic surrogate function is well known, but the analytical solution to this cubic surrogate function has not been well studied. We provide a derivation for Equation (18) in Appendix A.

Since these surrogate functions are convex and are upper bounds of the original functions, minimizing them will lead to a decrease of the original function $f(x)$ as well. This explains why our methods ensure monotonic decrease in loss and guarantee global convergence. The final algorithms are very easy to understand and can be thought of as coordinate descent-type methods. We anticipate these core ideas can be applied to solve a wide range of problems related to the CPH model. In the next subsection, we showcase two problems our algorithms can tackle.

## 3.5 Applications to Regularized and Constrained Problems

**Regularized Problem** The first problem is the regularized CPH problem whose penalty terms are separable. The penalties that qualify for this category include LASSO [64], ElasticNet [72], SCAD [15], MCP [68], etc. For the $\ell_1$-regularized problems, we can in fact find analytical solutions [2].

For the quadratic surrogate function, solving the $\ell_1$-regularized problem in Equation (15) is equivalent to solving the following optimization problem (with $a = f'(x)$, $b = L_2$, and $c = x$),

$$\Delta\hat{x} = \underset{\Delta x}{\operatorname{argmin}}\, a\Delta x + \frac{1}{2}b\Delta x^2 + \lambda_1|c + \Delta x|. \tag{19}$$

The solution for the above problem is

$$\Delta\hat{x} = \begin{cases} -(a - \lambda_1)/b & \text{if} \quad bc - a < -\lambda_1 \\ -(a + \lambda_1)/b & \text{if} \quad bc - a > \lambda_1 \\ -c & \text{otherwise.} \end{cases} \tag{20}$$

For the cubic surrogate function, solving the $\ell_1$-regularized problem of Equation (16) is equivalent to solving the following optimization problem (with $a = f'(x)$, $b = f''(x)$, $c = L_3$, and $d = x$):

$$\Delta\hat{x} = \underset{\Delta x}{\operatorname{argmin}}\, a\Delta x + \frac{1}{2}b\Delta x^2 + \frac{1}{6}c|x|^3 + \lambda_1|d + \Delta x|, \tag{21}$$

whose solution is:

$$\Delta\hat{x} = \begin{cases} \operatorname{sgn}(d)\left(-b + \sqrt{b^2 - 2c(\operatorname{sgn}(d)a + \lambda_1)}\right)/c & \text{if} \quad \operatorname{sgn}(d)a + \lambda_1 \leq 0 \\ \operatorname{sgn}(d)\left(b + \sqrt{b^2 + 2c(\operatorname{sgn}(d)a - \lambda_1)}\right)/c & \text{if} \quad \operatorname{sgn}(d)(a - bd) - \frac{1}{2}cd^2 > \lambda_1 \\ \operatorname{sgn}(d)\left(b + \sqrt{b^2 + 2c(\operatorname{sgn}(d)a + \lambda_1)}\right)/c & \text{if} \quad \operatorname{sgn}(d)(a - bd) - \frac{1}{2}cd^2 < -\lambda_1 \\ -d & \text{otherwise.} \end{cases} \tag{22}$$

Equation (20) is well known in a slightly different format. Equation (22) has not been well studied in the past. We provide derivations for both in Appendix A.

---

[2]For the ElasticNet problem where the penalty is $\lambda_1\|\cdot\|_1 + \lambda_2\|\cdot\|_2^2$, we can also easily obtain analytical solutions. The trick is to absorb the first and second order derivatives into the coefficients of the surrogate functions and only have a $\lambda_1\|\cdot\|$ penalty.

**Constrained Problem** The second problem is the cardinality-constrained CPH problem. Recently, the beam search framework (a combination of the beam search method [66] from natural language processing and generalized orthogonal matching pursuit [13]) has shown promise in finding near optimal solutions for a class of $l_0$-constrained nonconvex problems, including sparse ridge regression [47] and sparse logistic regression [48].

Similar to the generalized orthogonal matching pursuit algorithm, we expand our support (starting from an empty set) by adding one feature at a time until the cardinality is satisfied. However, instead of selecting features based on partial derivatives, we select features based on which coefficient, if optimized, can result in the largest decrease of the loss function. After the feature is added into the support, we fine-tune all nonzero coefficients in the support. Additionally, during each support expansion step, we select multiple feature candidate instead of the best one, similar to the core idea in beam search. We use our coordinate descent methods to solve the feature selection step and the coefficient fine-tuning step.

Although the beam search framework has already been proposed for other cardinality-constrained problems, it cannot be applied directly to the CPH model without our coordinate descent methods to select features, especially in the highly correlated settings.

## 4 Experiments

We test the effectiveness of our optimization methods on both synthetic and real-world datasets. We run experiments for both regularized and constrained problems mentioned in Section 3.5. Our main objectives are: 1) When minimizing the same objective functions, how fast can our methods converge to the optimal solutions when compared with all existing optimization methods for the CPH model? 2) When coupled with the beam search framework, how well can our methods help with variable selection when compared with the state-of-the-arts methods, especially for challenging scenarios where features are highly correlated?

### 4.1 Accessing How Fast Our Methods Converge to Optimal Solutions

We compare our methods (one based on the quadratic surrogate function and the other based on the cubic surrogate function) with the existing optimization methods outlined in Section 2: exact Newton method, the quasi Newton method, and the proximal Newton method. We run on both $\ell_2$-regularized CPH problems and $\ell_1 + \ell_2$-regularized CPH problems. The choices of these regularizations are: $\lambda_2 = \{0, 1\}$ and $\lambda_2 = \{1, 5\}$. The coefficients are all initialized to be $0$. In the main paper, we show results on the *Flchain* dataset in Figure 1. More results on other datasets can be found in Appendix D. During each iteration, the baseline methods [62, 51] optimize all coefficients at once, whereas our methods optimize coefficients sequentially with respect to the original loss function. To assess the per-iteration convergence rate, we plot the CPH loss against the number of iterations. To assess the practical running speed, we plot the CPH loss against the overall time elapsed (wall clock). From the left two plots of loss vs. number of iterations, we see that the *Newton-type baselines sometimes have losses that blow up or increase* during the initial phase of optimization. This is a common problem of Newton's method. Our methods are the only ones with monotonically decreasing loss curves. This is the main reason why only our method can be used for the beam search framework in the variable selection experiments. From the right two plots of loss vs. overall time elapsed, we can see that *our methods are significantly faster than the baselines*. This is due to the fact that both our first and second order partial derivatives are very cheap to compute (with time complexity $O(n)$), as we have explained in Section 3.2.

### 4.2 Accessing How Well Our Methods Perform Variable Selection

We compare our method with both Cox-based methods and other model classes. For Cox-based models, we run on both synthetic datasets and real-world datasets. For other model classes, we only run on the real-world datasets. To assess how well different methods select important variables, features are highly correlated in all datasets. Synthetic datasets are generated with high correlation level, $\rho = 0.9$. For each continuous feature on the real-world datasets, we perform binary thresholding for preprocessing [49] to obtain many one-hot encoded binary features. This preprocessing step result in highly correlated features on which it is challenging to perform variable selection. We use the

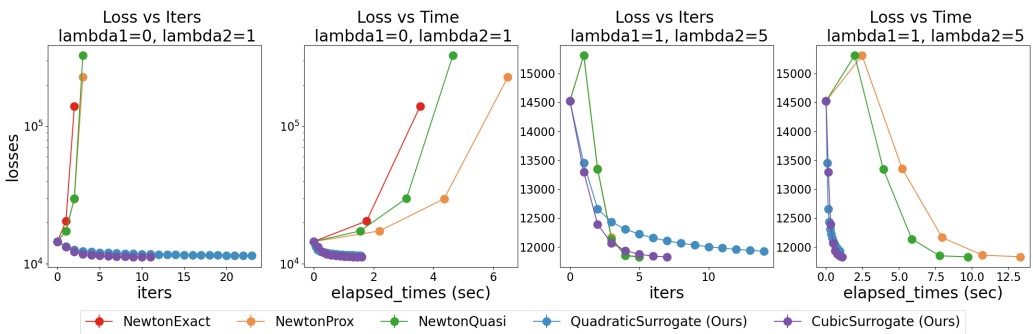

Figure 1: Efficiency experiments on the first fold of the Flchain dataset. a) The left two plots are on the $\ell_2$-regularized problem with $\lambda_2 = 1$. For all Newton-type methods, the losses blow up when regularization is weak. In contrast, our methods (quadratic and cubic surrogates) ensure monotonic decrease of losses. b) The right two plots are on the $\ell_1 + \ell_2$–regularized problem with $\lambda_1 = 1$ and $\lambda_2 = 5$. The exact Newton method cannot be directly applied, so we compare only with quasi Newton [62] and proximal Newton [51] methods, which have losses that increase at the beginning. Our methods are significantly faster than both baselines. Because the evaluation cost per iteration is very cheap for our methods, we are significantly faster in terms of wall clock time (see the difference between the third and fourth plots). See Appendix D for results on other datasets.

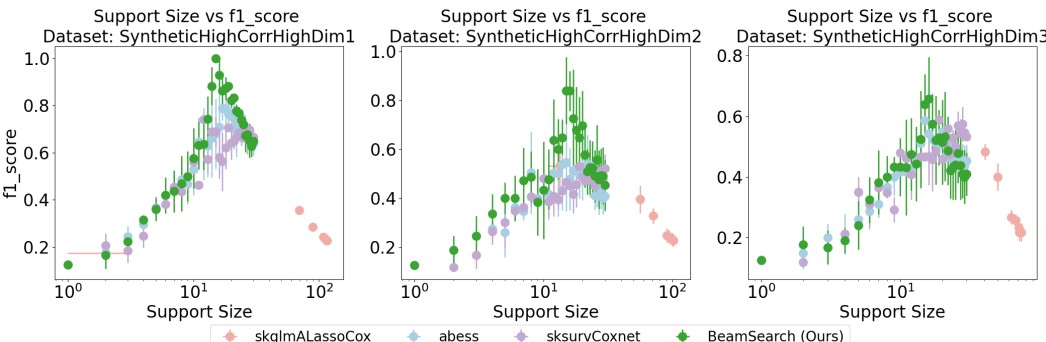

Figure 2: Variable selection on synthetic datasets with high correlation (correlation level $\rho = 0.9$). From left to right, the sample sizes are 1200, 1000, and 800, respectively. The F1 score (the higher the better) is closely related to the support recovery rate. On the left two plots, we can see our method recovers the true variables significantly better than other methods ($100\%$ recovery rate on the left plot; true support size is 15). As the sample size decreases, the F1 score decreases for all methods.

following metrics to evaluate our solution qualities: CPH loss, CIndex, and IBS. On the synthetic datasets where we know the true coefficients, we also calculate the F1 score. We perform 5-fold cross validation and report the mean and standard deviation of different metrics on both the training and test sets. For details about the experimental setup, please see Appendix C. For Cox-based methods, we compare our method with Coxnet, Abess, and Adaptive Lasso.

Results on the synthetic datasets are shown in Figure 2. We plot support size vs F1 score. The F1 score is closely related to the support recovery rate. Our method performs significantly better than the baselines. In particular, on the leftmost plot with 1200 samples, *our method is the only one to achieve 100% recovery rate*; the true support size is 15 and we recover all 15 features with a model of size 15. Results for the Employee Attrition dataset are shown in Figure 3. We plot support size vs. CIndex and support size vs. IBS. Similar to the trend on the synthetic datasets, *our method performs significantly better than the baselines in terms of both metrics*. Lastly, we compare our method with other model classes on the Dialysis dataset. The results are shown in Figure 4. We plot support size vs. CIndex and support size vs. IBS. The results indicate that other model classes are prone to overfitting on the training sets. Our method achieves the best accuracy-sparsity tradeoff. *We are able to obtain solutions with the smallest number of coefficients without losing predictive performance.*

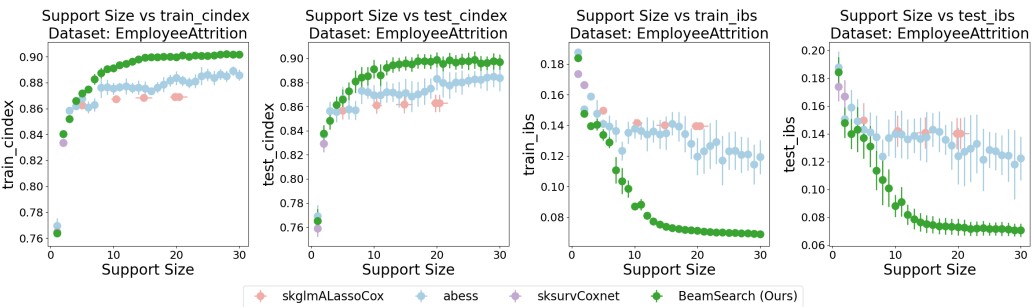

Figure 3: Variable selection on the Employee Attrition dataset. We show support size vs. CIndex (left two plots, the higher the better) and support size vs. IBS score (right two plots, the lower the better). We compare our method with Cox-based sparse learning methods. For both metrics, our method is significantly better than other baselines.

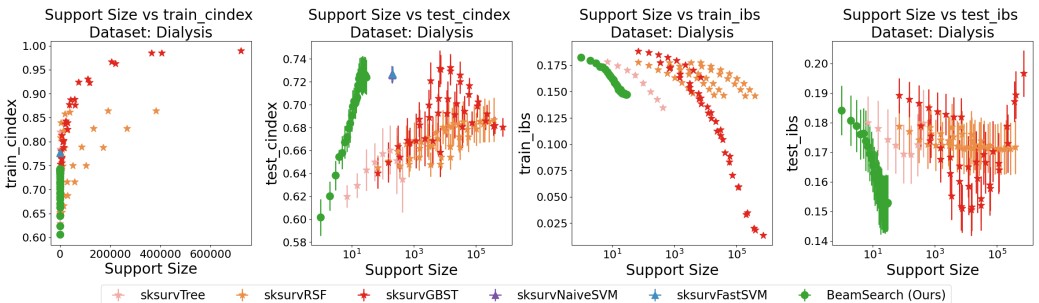

Figure 4: Variable selection on the Dialysis dataset. We show support size vs. CIndex (left two plots, the higher the better) and support size vs. IBS score (right two plots, the lower the better). We compare our method with other model classes. For both metrics, our method obtains solutions that are significantly sparser than other model classes without losing accuracy on the test sets. Other model classes are prone to overfitting on the training sets.

All these results demonstrate the superior sparse learning capability of our method. For more results, with all baselines on all datasets, please see Appendix D.

**Limitations of FastSurvival** Our work focuses on efficient training and effective variable selection of the CPH model. Other model classes, such as trees, random forests, and neural networks, have their own unique merits in capturing complex patterns when the linear (or in our case, additive) model assumption is not satisfied. Another limitation is using the CPH model itself, since its assumptions do not always hold. Handling this question is out of scope for this work.

## 5 Conclusion and Future Outlook

We presented new optimization methods to train the Cox proportional hazards (CPH) model by constructing and minimizing either a quadratic or a cubic surrogate function. We achieve computational efficiency by exploiting the hidden mathematical structures discovered for the CPH model. Our algorithms are able to train the model significantly faster than previous approaches while avoiding the issue of loss explosion. Furthermore, when applied to the variable selection problem, our method can produce solutions with much fewer parameters while maintaining predictive performance. There are many possible extensions to build upon this work. On the optimization side, it will be interesting to see whether we can derive analytical solutions for other types of regularizers mentioned in Section 3.5. On the theoretical side, questions remain whether higher order partial derivatives are Lipschitz-continuous and how to compute these Lipschitz constants. On the application side, we can apply our method to solve the CPH models with time-varying features [16], stratifications [40], and feature interactions [45].

## Code Availability

Implementations of FastSurvival discussed in this paper are available at `https://github.com/jiachangliu/FastSurvival`.

## Acknowledgements

The authors gratefully acknowledge funding support from the National Institutes of Health under 5R01-DA054994 and the Department of Energy under DE-SC0021358.

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

# Appendix to FastSurvival: Hidden Computational Blessings in Training Cox Proportional Hazards Models

## Table of Contents

# A Derivations and Proofs

## A.1 First, Second, and Third Order Partial Derivatives

The derivation for the first order partial derivative is shown in Section A.1.1. The derivation for the second order partial derivative is shown in Section A.1.2. The derivation for the third order partial derivative is shown in Section A.1.3.

### A.1.1 First Order Partial Derivative

We want to show that

$$\frac{\partial \ell(\boldsymbol{\beta})}{\partial \beta_l} = \sum_{i=1}^{n} \delta_i \left( \sum_{k \in R_i} \frac{e^{\eta_k}}{\sum_{j \in R_i} e^{\eta_j}} X_{kl} \right) - \sum_{i=1}^{n} \delta_i X_{il}.$$

$$\frac{\partial \ell(\boldsymbol{\beta})}{\partial \beta_l} = \frac{\partial \ell(\boldsymbol{\eta})}{\partial \beta_l}$$

$$= \sum_{k_1=1}^{n} \frac{\partial \ell(\boldsymbol{\eta})}{\partial \eta_{k_1}} \frac{\partial \eta_{k_1}}{\partial \beta_l} \qquad \textcolor{green}{\text{\# apply chain rule from calculus}}$$

$$= \sum_{k_1=1}^{n} \frac{\partial}{\partial \eta_{k_1}} \left\{ \sum_{i=1}^{n} \delta_i \left[ \log(\sum_{j \in R_i} e^{\eta_j}) - \eta_i \right] \right\} \frac{\partial \eta_{k_1}}{\partial \beta_l} \qquad \textcolor{green}{\text{\# plug in the formula for } \ell(\boldsymbol{\eta})}$$

$$= \sum_{i=1}^{n} \delta_i \left\{ \sum_{k_1=1}^{n} \frac{\partial}{\partial \eta_{k_1}} \left[ \log(\sum_{j \in R_i} e^{\eta_j}) - \eta_i \right] \frac{\partial \eta_{k_1}}{\partial \beta_l} \right\}$$
$$\textcolor{green}{\text{\# exchange the summation orders; sum over } k_1 \text{ first and sum over } i \text{ later}}$$

$$= \sum_{i=1}^{n} \delta_i \left\{ \sum_{k_1=1}^{n} \frac{\partial}{\partial \eta_{k_1}} \left[ \log(\sum_{j \in R_i} e^{\eta_j}) - \eta_i \right] X_{k_1 l} \right\}$$
$$\textcolor{green}{\text{\# evaluate the partial derivative } \frac{\partial \eta_{k_1}}{\partial \beta_l} = X_{k_1 l}}$$

$$= \sum_{i=1}^{n} \delta_i \left\{ \sum_{k_1=1}^{n} \left[ \frac{\partial}{\partial \eta_{k_1}} (\log(\sum_{j \in R_i} e^{\eta_j})) - \frac{\partial}{\partial \eta_{k_1}} (\eta_i) \right] X_{k_1 l} \right\} \qquad \textcolor{green}{\text{\# distribute } \frac{\partial}{\partial \eta_{k_1}} (\cdot) \text{ inside } [\cdot]}$$

$$= \sum_{i=1}^{n} \delta_i \left[ \sum_{k_1=1}^{n} \left( \frac{1}{\sum_{j \in R_i} e^{\eta_j}} e^{\eta_{k_1}} \mathbb{1}_{y_{k_1} \geq y_i} - \mathbb{1}_{k_1=i} \right) X_{k_1 l} \right]$$
$$\textcolor{green}{\text{\# } \frac{\partial}{\partial \eta_{k_1}} (\log(\sum_{j \in R_i} e^{\eta_j})) = \frac{1}{\sum_{j \in R_i} e^{\eta_j}} e^{\eta_{k_1}} \mathbb{1}_{y_{k_1} \geq y_i} \text{ and } \frac{\partial}{\partial \eta_{k_1}} (\eta_i) = \mathbb{1}_{k_1=i}}$$

$$= \sum_{i=1}^{n} \delta_i \left[ \sum_{k_1=1}^{n} \left( \frac{1}{\sum_{j \in R_i} e^{\eta_j}} e^{\eta_{k_1}} \mathbb{1}_{y_{k_1} \geq y_i} X_{k_1 l} - \mathbb{1}_{k_1=i} X_{k_1 l} \right) \right] \qquad \textcolor{green}{\text{\# distribute } X_{k_1 l} \text{ inside } [\cdot]}$$

$$= \sum_{i=1}^{n} \delta_i \left( \sum_{k_1=1}^{n} \frac{1}{\sum_{j \in R_i} e^{\eta_j}} e^{\eta_{k_1}} \mathbb{1}_{y_{k_1} \geq y_i} X_{k_1 l} - \sum_{k_1=1}^{n} \mathbb{1}_{k_1=i} X_{k_1 l} \right)$$
$$\textcolor{green}{\text{\# distribute } \sum_{k_1=1}^{n} \text{ inside } (\cdot)}$$

$$= \sum_{i=1}^{n} \delta_i \left( \sum_{k_1=1}^{n} \mathbb{1}_{y_{k_1} \geq y_i} \frac{1}{\sum_{j \in R_i} e^{\eta_j}} e^{\eta_{k_1}} X_{k_1 l} - \sum_{k_1=1}^{n} \mathbb{1}_{k_1=i} X_{k_1 l} \right)$$
$$\textcolor{green}{\text{\# move } \mathbb{1}_{y_{k_1} \geq y_i} \text{ to the front}}$$

$$= \sum_{i=1}^{n} \delta_i \left( \sum_{k_1 \in R_i} \frac{1}{\sum_{j \in R_i} e^{\eta_j}} e^{\eta_{k_1}} X_{k_1 l} - X_{il} \right)$$
$$\textcolor{green}{\text{\# evaluate the two summations over } k_1, \text{ i.e., } \sum_{k_1=1}^{n}; \text{ the first summation can be simplified because } R_i \text{ is a set of indices whose time is greater than or equal to } y_i}$$

$$= \sum_{i=1}^{n} \delta_i \Big( \sum_{k_1 \in R_i} \frac{1}{\sum_{j \in R_i} e^{\eta_j}} e^{\eta_{k_1}} X_{k_1 l} \Big) - \sum_{i=1}^{n} \delta_i X_{il} \qquad \textcolor{green}{\textit{\# distribute } \sum_{i=1}^{n} \delta_i \textit{ inside } (\cdot)}$$

$$= \sum_{i=1}^{n} \delta_i \Big( \sum_{k_1 \in R_i} \frac{e^{\eta_{k_1}}}{\sum_{j \in R_i} e^{\eta_j}} X_{k_1 l} \Big) - \sum_{i=1}^{n} \delta_i X_{il} \qquad \textcolor{green}{\textit{\# move } e^{\eta_{k_1}} \textit{ to the numerator}}$$

$$= \sum_{i=1}^{n} \delta_i \Big( \sum_{k \in R_i} \frac{e^{\eta_k}}{\sum_{j \in R_i} e^{\eta_j}} X_{kl} \Big) - \sum_{i=1}^{n} \delta_i X_{il} \qquad \textcolor{green}{\textit{\# change notation } k_1 \textit{ to } k}$$

### A.1.2  Second Order Partial Derivative

We want to show that

$$\frac{\partial^2 \ell(\boldsymbol{\beta})}{\partial \beta_l^2} = \sum_{i=1}^{n} \delta_i \left[ \sum_{k \in R_i} \frac{e^{\eta_k}}{\sum_{j \in R_i} e^{\eta_j}} X_{kl}^2 - \Big( \sum_{k \in R_i} \frac{e^{\eta_k}}{\sum_{j \in R_i} e^{\eta_j}} X_{kl} \Big)^2 \right].$$

$$\frac{\partial^2 \ell(\boldsymbol{\beta})}{\partial \beta_l^2} = \frac{\partial^2 \ell(\boldsymbol{\eta})}{\partial \beta_l^2}$$

$$= \sum_{k_2=1}^{n} \frac{\partial}{\partial \eta_{k_2}} \Big( \frac{\partial \ell(\boldsymbol{\eta})}{\partial \beta_l} \Big) \frac{\partial \eta_{k_2}}{\partial \beta_l} \qquad \textcolor{green}{\textit{\# Apply chain rule from calculus}}$$

$$= \sum_{k_2=1}^{n} \frac{\partial}{\partial \eta_{k_2}} \Big[ \sum_{i=1}^{n} \delta_i \Big( \sum_{k \in R_i} \frac{e^{\eta_k}}{\sum_{j \in R_i} e^{\eta_j}} X_{kl} \Big) - \sum_{i=1}^{n} \delta_i X_{il} \Big] \frac{\partial \eta_{k_2}}{\partial \beta_l}$$

<p style="text-align:center">*# plug in $\frac{\partial \ell(\boldsymbol{\eta})}{\partial \beta_l}$ from the end result for the first order partial derivative above*</p>

$$= \sum_{i=1}^{n} \delta_i \Big[ \sum_{k_2=1}^{n} \frac{\partial}{\partial \eta_{k_2}} \Big( \sum_{k \in R_i} \frac{e^{\eta_k}}{\sum_{j \in R_i} e^{\eta_j}} X_{kl} \Big) \frac{\partial \eta_{k_2}}{\partial \beta_l} \Big] - \sum_{k_2=1}^{n} \frac{\partial}{\partial \eta_{k_2}} \Big( \sum_{i=1}^{n} \delta_i X_{il} \Big) \frac{\partial \eta_{k_2}}{\partial \beta_l}$$

<p>*# distribute $\sum_{k_2=1}^{n} \frac{\partial}{\partial \eta_{k_2}}(\cdot) \frac{\partial \eta_{k_2}}{\partial \beta_l}$ inside each term inside $[\cdot]$; for the first term, also exchange the summation orders by summing over $i$ first and $k_2$ later*</p>

$$= \sum_{i=1}^{n} \delta_i \Big[ \sum_{k_2=1}^{n} \frac{\partial}{\partial \eta_{k_2}} \Big( \sum_{k \in R_i} \frac{e^{\eta_k}}{\sum_{j \in R_i} e^{\eta_j}} X_{kl} \Big) \frac{\partial \eta_{k_2}}{\partial \beta_l} \Big]$$

<p style="text-align:center">*# second term is $0$ because the input to $\frac{\partial}{\partial \eta_{k_2}}(\cdot)$ is a constant*</p>

$$= \sum_{i=1}^{n} \delta_i \Big\{ \sum_{k_2=1}^{n} \Big[ \sum_{k \in R_i} \frac{\partial}{\partial \eta_{k_2}} \Big( \frac{e^{\eta_k}}{\sum_{j \in R_i} e^{\eta_j}} \Big) X_{kl} \Big] \frac{\partial \eta_{k_2}}{\partial \beta_l} \Big\}$$

<p style="text-align:center">*# move $\frac{\partial}{\partial \eta_{k_2}}$ inside the summation of $k$*</p>

$$= \sum_{i=1}^{n} \delta_i \Big[ \sum_{k_2=1}^{n} \Big( \sum_{k \in R_i} \frac{\big( \sum_{j \in R_i} e^{\eta_j} \big) e^{\eta_k} \mathbb{1}_{k=k_2} - e^{\eta_k} e^{\eta_{k_2}} \mathbb{1}_{y_{k_2} \geq y_i}}{\big( \sum_{j \in R_i} e^{\eta_j} \big)^2} X_{kl} \Big) \frac{\partial \eta_{k_2}}{\partial \beta_l} \Big]$$

<p style="text-align:center">*# evaluate the partial derivative using quotient rule from calculus*</p>

$$= \sum_{i=1}^{n} \delta_i \Big[ \sum_{k_2=1}^{n} \Big( \sum_{k \in R_i} \frac{\big( \sum_{j \in R_i} e^{\eta_j} \big) e^{\eta_k} \mathbb{1}_{k=k_2} - e^{\eta_k} e^{\eta_{k_2}} \mathbb{1}_{y_{k_2} \geq y_i}}{\big( \sum_{j \in R_i} e^{\eta_j} \big)^2} X_{kl} \Big) X_{k_2 l} \Big]$$

<p style="text-align:right">*# $\frac{\partial \eta_{k_2}}{\partial \beta_l} = X_{k_2 l}$*</p>

$$= \sum_{i=1}^{n} \delta_i \Big[ \sum_{k_2=1}^{n} \Big( \sum_{k \in R_i} \frac{\big( \sum_{j \in R_i} e^{\eta_j} \big) e^{\eta_k} \mathbb{1}_{k=k_2} - e^{\eta_k} e^{\eta_{k_2}} \mathbb{1}_{y_{k_2} \geq y_i}}{\big( \sum_{j \in R_i} e^{\eta_j} \big)^2} X_{kl} X_{k_2 l} \Big) \Big]$$

<p style="text-align:center">*# move $X_{k_2 l}$ inside $\sum_{k \in R_i}$*</p>

$$= \sum_{i=1}^{n} \delta_i \Big[ \sum_{k_2=1}^{n} \sum_{k \in R_i} \Big( \frac{e^{\eta_k} \mathbb{1}_{k=k_2}}{\sum_{j \in R_i} e^{\eta_j}} - \frac{e^{\eta_k} e^{\eta_{k_2}} \mathbb{1}_{y_{k_2} \geq y_i}}{\big( \sum_{j \in R_i} e^{\eta_j} \big)^2} \Big) X_{kl} X_{k_2 l} \Big]$$

<p style="text-align:center">*# divide the fraction into two parts*</p>

<p style="text-align:center">16</p>

$$= \sum_{i=1}^{n} \delta_i \Big[ \sum_{k_2=1}^{n} \sum_{k \in R_i} \Big( \frac{e^{\eta_k} \mathbb{1}_{k=k_2}}{\sum_{j \in R_i} e^{\eta_j}} X_{kl} X_{k_2 l} - \frac{e^{\eta_k} e^{\eta_{k_2}} \mathbb{1}_{y_{k_2} \geq y_i}}{\left( \sum_{j \in R_i} e^{\eta_j} \right)^2} X_{kl} X_{k_2 l} \Big) \Big]$$
# distribute $X_{kl} X_{k_2 l}$ inside $(\cdot)$

$$= \sum_{i=1}^{n} \delta_i \Big[ \sum_{k \in R_i} \sum_{k_2=1}^{n} \Big( \frac{e^{\eta_k} \mathbb{1}_{k=k_2}}{\sum_{j \in R_i} e^{\eta_j}} X_{kl} X_{k_2 l} - \frac{e^{\eta_k} e^{\eta_{k_2}} \mathbb{1}_{y_{k_2} \geq y_i}}{\left( \sum_{j \in R_i} e^{\eta_j} \right)^2} X_{kl} X_{k_2 l} \Big) \Big]$$
# exchange summation orders; sum over $k_2$ and then $k$

$$= \sum_{i=1}^{n} \delta_i \Big[ \sum_{k \in R_i} \Big( \sum_{k_2=1}^{n} \frac{e^{\eta_k} \mathbb{1}_{k=k_2}}{\sum_{j \in R_i} e^{\eta_j}} X_{kl} X_{k_2 l} - \sum_{k_2=1}^{n} \frac{e^{\eta_k} e^{\eta_{k_2}} \mathbb{1}_{y_{k_2} \geq y_i}}{\left( \sum_{j \in R_i} e^{\eta_j} \right)^2} X_{kl} X_{k_2 l} \Big) \Big]$$
# distribute $\sum_{k_2=1}^{n}$ inside $(\cdot)$

$$= \sum_{i=1}^{n} \delta_i \Big[ \sum_{k \in R_i} \Big( \frac{e^{\eta_k}}{\sum_{j \in R_i} e^{\eta_j}} X_{kl}^2 - \sum_{k_2 \in R_i} \frac{e^{\eta_k} e^{\eta_{k_2}}}{\left( \sum_{j \in R_i} e^{\eta_j} \right)^2} X_{kl} X_{k_2 l} \Big) \Big]$$
*for the first term, only $k_2 = k$ is left because of the indicator $\mathbb{1}_{k=k_2}$; for the second # term, the expression can be simplified because $R_i$ is a set of indices whose time is greater than or equal to $y_i$*

$$= \sum_{i=1}^{n} \delta_i \Big( \sum_{k \in R_i} \frac{e^{\eta_k}}{\sum_{j \in R_i} e^{\eta_j}} X_{kl}^2 - \sum_{k \in R_i} \sum_{k_2 \in R_i} \frac{e^{\eta_k} e^{\eta_{k_2}}}{\left( \sum_{j \in R_i} e^{\eta_j} \right)^2} X_{kl} X_{k_2 l} \Big)$$
# distribute $\sum_{k \in R_i}$ inside $(\cdot)$

$$= \sum_{i=1}^{n} \delta_i \Big[ \sum_{k \in R_i} \frac{e^{\eta_k}}{\sum_{j \in R_i} e^{\eta_j}} X_{kl}^2 - \Big( \sum_{k \in R_i} \frac{e^{\eta_k}}{\sum_{j \in R_i} e^{\eta_j}} X_{kl} \Big) \Big( \sum_{k_2 \in R_i} \frac{e^{\eta_{k_2}}}{\sum_{j \in R_i} e^{\eta_j}} X_{k_2 l} \Big) \Big]$$
# the double sum of the second term can be rewritten as product of two sums

$$= \sum_{i=1}^{n} \delta_i \Big[ \sum_{k \in R_i} \frac{e^{\eta_k}}{\sum_{j \in R_i} e^{\eta_j}} X_{kl}^2 - \Big( \sum_{k \in R_i} \frac{e^{\eta_k}}{\sum_{j \in R_i} e^{\eta_j}} X_{kl} \Big)^2 \Big]$$
*# the two terms of the product are exactly the same and can be simplified because $k$ and $k_2$ are independent*

### A.1.3 Third Order Partial Derivative

We want to show that

$$\frac{\partial^3 \ell(\boldsymbol{\beta})}{\partial \beta_l^3} = \sum_{i=1}^{n} \delta_i \Bigg[ \sum_{k \in R_i} \frac{e^{\eta_k}}{\sum_{j \in R_i} e^{\eta_j}} X_{kl}^3 + 2 \Bigg( \sum_{k \in R_i} \frac{e^{\eta_k}}{\sum_{j \in R_i} e^{\eta_j}} X_{kl} \Bigg)^3$$
$$- 3 \Bigg( \sum_{k \in R_i} \frac{e^{\eta_k}}{\sum_{j \in R_i} e^{\eta_j}} X_{kl}^2 \Bigg) \Bigg( \sum_{k \in R_i} \frac{e^{\eta_k}}{\sum_{j \in R_i} e^{\eta_j}} X_{kl} \Bigg) \Bigg] .$$

$$\frac{\partial^3 \ell(\boldsymbol{\beta})}{\partial \beta_l^3} = \frac{\partial^3 \ell(\boldsymbol{\eta})}{\partial \beta_l^3}$$

$$= \sum_{k_3=1}^{n} \frac{\partial}{\partial \eta_{k_3}} \Big( \frac{\partial^2 \ell(\boldsymbol{\eta})}{\partial \beta_l^2} \Big) \frac{\partial \eta_{k_3}}{\partial \beta_l} \qquad \qquad \text{\color{green}# Apply chain rule from calculus}$$

$$= \sum_{k_3=1}^{n} \frac{\partial}{\partial \eta_{k_3}} \Big\{ \sum_{i=1}^{n} \delta_i \Big[ \sum_{k \in R_i} \frac{e^{\eta_k}}{\sum_{j \in R_i} e^{\eta_j}} X_{kl}^2 - \Big( \sum_{k \in R_i} \frac{e^{\eta_k}}{\sum_{j \in R_i} e^{\eta_j}} X_{kl} \Big)^2 \Big] \Big\} \frac{\partial \eta_{k_3}}{\partial \beta_l}$$
# plug in $\frac{\partial \ell(\boldsymbol{\eta})}{\partial \beta_l}$ from the end result for the second order partial derivative above

$$= \sum_{i=1}^{n} \delta_i \Big\{ \sum_{k_3=1}^{n} \frac{\partial}{\partial \eta_{k_3}} \Big[ \sum_{k \in R_i} \frac{e^{\eta_k}}{\sum_{j \in R_i} e^{\eta_j}} X_{kl}^2 - \Big( \sum_{k \in R_i} \frac{e^{\eta_k}}{\sum_{j \in R_i} e^{\eta_j}} X_{kl} \Big)^2 \Big] \frac{\partial \eta_{k_3}}{\partial \beta_l} \Big\}$$
*# exchange summation orders; sum over $k_3$ and then $i$ by moving $\frac{\partial}{\partial \eta_{k_3}}(\cdot) \frac{\partial \eta_{k_3}}{\partial \beta_l}$ inside into the inner summation*

$$= \sum_{i=1}^{n} \delta_i \Big\{ \sum_{k_3=1}^{n} \frac{\partial}{\partial \eta_{k_3}} \Big[ \sum_{k \in R_i} \frac{e^{\eta_k}}{\sum_{j \in R_i} e^{\eta_j}} X_{kl}^2 - \Big( \sum_{k \in R_i} \frac{e^{\eta_k}}{\sum_{j \in R_i} e^{\eta_j}} X_{kl} \Big)^2 \Big] X_{k_3 l} \Big\}$$

# $\frac{\partial \eta_{k_3}}{\partial \beta_l} = X_{k_3 l}$

$$= \sum_{i=1}^{n} \delta_i \Big\{ \sum_{k_3=1}^{n} \Big\{ \frac{\partial}{\partial \eta_{k_3}} \Big( \sum_{k \in R_i} \frac{e^{\eta_k}}{\sum_{j \in R_i} e^{\eta_j}} X_{kl}^2 \Big) - \frac{\partial}{\partial \eta_{k_3}} \Big[ \Big( \sum_{k \in R_i} \frac{e^{\eta_k}}{\sum_{j \in R_i} e^{\eta_j}} X_{kl} \Big)^2 \Big] \Big\} X_{k_3 l} \Big\}$$

# distribute $\frac{\partial}{\partial \eta_{k_3}}$ inside $[\cdot]$

$$= \sum_{i=1}^{n} \delta_i \Big\{ \sum_{k_3=1}^{n} \Big\{ \sum_{k \in R_i} \frac{\partial}{\partial \eta_{k_3}} \Big( \frac{e^{\eta_k}}{\sum_{j \in R_i} e^{\eta_j}} \Big) X_{kl}^2 - \frac{\partial}{\partial \eta_{k_3}} \Big[ \Big( \sum_{k \in R_i} \frac{e^{\eta_k}}{\sum_{j \in R_i} e^{\eta_j}} X_{kl} \Big)^2 \Big] \Big\} X_{k_3 l} \Big\}$$

# for the first term, move $\frac{\partial}{\partial \eta_{k_3}}(\cdot)$ inside the summation

$$= \sum_{i=1}^{n} \delta_i \Big\{ \sum_{k_3=1}^{n} \Big[ \sum_{k \in R_i} \frac{\partial}{\partial \eta_{k_3}} \Big( \frac{e^{\eta_k}}{\sum_{j \in R_i} e^{\eta_j}} \Big) X_{kl}^2$$
$$- 2 \Big( \sum_{k \in R_i} \frac{e^{\eta_k}}{\sum_{j \in R_i} e^{\eta_j}} X_{kl} \Big) \frac{\partial}{\partial \eta_{k_3}} \Big( \sum_{k \in R_i} \frac{e^{\eta_k}}{\sum_{j \in R_i} e^{\eta_j}} X_{kl} \Big) \Big] X_{k_3 l} \Big\}$$

# apply product rule from calculus to the second term

$$= \sum_{i=1}^{n} \delta_i \Big\{ \sum_{k_3=1}^{n} \Big\{ \sum_{k \in R_i} \frac{\partial}{\partial \eta_{k_3}} \Big( \frac{e^{\eta_k}}{\sum_{j \in R_i} e^{\eta_j}} \Big) X_{kl}^2$$
$$- 2 \Big( \sum_{k \in R_i} \frac{e^{\eta_k}}{\sum_{j \in R_i} e^{\eta_j}} X_{kl} \Big) \Big[ \sum_{k \in R_i} \frac{\partial}{\partial \eta_{k_3}} \Big( \frac{e^{\eta_k}}{\sum_{j \in R_i} e^{\eta_j}} \Big) X_{kl} \Big] \Big\} X_{k_3 l} \Big\}$$

# for the second term, move $\frac{\partial}{\partial \eta_{k_3}}(\cdot)$ inside the summation

$$= \sum_{i=1}^{n} \delta_i \Big\{ \sum_{k_3=1}^{n} \Big\{ \sum_{k \in R_i} \frac{\partial}{\partial \eta_{k_3}} \Big( \frac{e^{\eta_k}}{\sum_{j \in R_i} e^{\eta_j}} \Big) X_{kl}^2 X_{k_3 l}$$
$$- 2 \Big( \sum_{k \in R_i} \frac{e^{\eta_k}}{\sum_{j \in R_i} e^{\eta_j}} X_{kl} X_{k_3 l} \Big) \Big[ \sum_{k \in R_i} \frac{\partial}{\partial \eta_{k_3}} \Big( \frac{e^{\eta_k}}{\sum_{j \in R_i} e^{\eta_j}} \Big) X_{kl} \Big] \Big\} \Big\}$$

# distribute $X_{k_3 l}$ inside $\{\cdot\}$

$$= \sum_{i=1}^{n} \delta_i \Big\{ \sum_{k_3=1}^{n} \Big\{ \sum_{k \in R_i} \Big( \frac{e^{\eta_k}}{\sum_{j \in R_i} e^{\eta_j}} \mathbb{1}_{k=k_3} - \frac{e^{\eta_k} e^{\eta_{k_3}}}{\big( \sum_{j \in R_i} e^{\eta_j} \big)^2} \mathbb{1}_{y_{k_3} \geq y_i} \Big) X_{kl}^2 X_{k_3 l}$$
$$- 2 \Big( \sum_{k \in R_i} \frac{e^{\eta_k}}{\sum_{j \in R_i} e^{\eta_j}} X_{kl} \Big) \Big[ \sum_{k \in R_i} \Big( \frac{e^{\eta_k}}{\sum_{j \in R_i} e^{\eta_j}} \mathbb{1}_{k=k_3} - \frac{e^{\eta_k} e^{\eta_{k_3}}}{\big( \sum_{j \in R_i} e^{\eta_j} \big)^2} \mathbb{1}_{y_{k_3} \geq y_i} \Big) X_{kl} \Big] X_{k_3 l} \Big\} \Big\}$$

# $\frac{\partial}{\partial \eta_{k_3}} \Big( \frac{e^{\eta_k}}{\sum_{j \in R_i} e^{\eta_j}} \Big) = \frac{e^{\eta_k}}{\sum_{j \in R_i} e^{\eta_j}} \mathbb{1}_{k=k_3} - \frac{e^{\eta_k} e^{\eta_{k_3}}}{\big( \sum_{j \in R_i} e^{\eta_j} \big)^2} \mathbb{1}_{y_{k_3} \geq y_i}$

$$= \sum_{i=1}^{n} \delta_i \Big\{ \sum_{k_3=1}^{n} \Big[ \sum_{k \in R_i} \Big( \frac{e^{\eta_k}}{\sum_{j \in R_i} e^{\eta_j}} \mathbb{1}_{k=k_3} - \frac{e^{\eta_k} e^{\eta_{k_3}}}{\big( \sum_{j \in R_i} e^{\eta_j} \big)^2} \mathbb{1}_{y_{k_3} \geq y_i} \Big) X_{kl}^2 X_{k_3 l}$$
$$- 2 \Big( \sum_{k \in R_i} \frac{e^{\eta_k}}{\sum_{j \in R_i} e^{\eta_j}} X_{kl} \Big) \Big( \sum_{k \in R_i} \frac{e^{\eta_k}}{\sum_{j \in R_i} e^{\eta_j}} \mathbb{1}_{k=k_3} X_{kl} - \sum_{k \in R_i} \frac{e^{\eta_k} e^{\eta_{k_3}}}{\big( \sum_{j \in R_i} e^{\eta_j} \big)^2} \mathbb{1}_{y_{k_3} \geq y_i} X_{kl} \Big) X_{k_3 l} \Big] \Big\}$$

# for the last term, distribute $\sum_{k \in R_i} (\cdot) X_{kl}$ inside $[\cdot]$

$$= \sum_{i=1}^{n} \Big\{ \delta_i \sum_{k_3=1}^{n} \Big[ \sum_{k \in R_i} \frac{e^{\eta_k}}{\sum_{j \in R_i} e^{\eta_j}} \mathbb{1}_{k=k_3} X_{kl}^2 X_{k_3 l} - \sum_{k \in R_i} \frac{e^{\eta_k} e^{\eta_{k_3}}}{\big( \sum_{j \in R_i} e^{\eta_j} \big)^2} \mathbb{1}_{y_{k_3} \geq y_i} X_{kl}^2 X_{k_3 l}$$
$$- 2 \Big( \sum_{k \in R_i} \frac{e^{\eta_k}}{\sum_{j \in R_i} e^{\eta_j}} X_{kl} \Big) \Big( \sum_{k \in R_i} \frac{e^{\eta_k}}{\sum_{j \in R_i} e^{\eta_j}} \mathbb{1}_{k=k_3} X_{kl} - \sum_{k \in R_i} \frac{e^{\eta_k} e^{\eta_{k_3}}}{\big( \sum_{j \in R_i} e^{\eta_j} \big)^2} \mathbb{1}_{y_{k_3} \geq y_i} X_{kl} \Big) X_{k_3 l} \Big] \Big\}$$

# for the first term, distribute $\sum_{k \in R_i} (\cdot) X_{kl}^2 X_{k_3 l}$ inside $(\cdot)$

$$= \sum_{i=1}^{n} \delta_i \Big\{ \sum_{k_3=1}^{n} \Big[ \sum_{k \in R_i} \frac{e^{\eta_k}}{\sum_{j \in R_i} e^{\eta_j}} \mathbb{1}_{k=k_3} X_{kl}^2 X_{k_3 l} - \sum_{k \in R_i} \frac{e^{\eta_k} e^{\eta_{k_3}}}{\left( \sum_{j \in R_i} e^{\eta_j} \right)^2} \mathbb{1}_{y_{k_3} \geq y_i} X_{kl}^2 X_{k_3 l} $$

$$- 2 \Big( \sum_{k \in R_i} \frac{e^{\eta_k}}{\sum_{j \in R_i} e^{\eta_j}} X_{kl} \Big) \Big( \sum_{k \in R_i} \frac{e^{\eta_k}}{\sum_{j \in R_i} e^{\eta_j}} X_{kl} \mathbb{1}_{k=k_3} X_{k_3 l} \Big) $$

$$+ 2 \Big( \sum_{k \in R_i} \frac{e^{\eta_k}}{\sum_{j \in R_i} e^{\eta_j}} X_{kl} \Big)^2 \Big( \frac{e^{\eta_{k_3}}}{\sum_{j \in R_i} e^{\eta_j}} \Big) \mathbb{1}_{y_{k_3} \geq y_i} X_{k_3 l} \Big] \Big\} $$

# *for the third term, distribute* $2 (\sum_{k \in R_i} (\frac{e^{\eta_k}}{\sum_{j \in R_i} e^{\eta_j}}))(\cdot) X_{k_3 l}$ *inside* $(\cdot)$

$$= \sum_{i=1}^{n} \delta_i \Big\{ \Big( \sum_{k \in R_i} \frac{e^{\eta_k}}{\sum_{j \in R_i} e^{\eta_j}} \Big) X_{kl}^2 \Big( \sum_{k_3=1}^{n} \mathbb{1}_{k=k_3} X_{k_3 l} \Big) $$

$$- \Big( \sum_{k \in R_i} \frac{e^{\eta_k}}{\sum_{j \in R_i} e^{\eta_j}} X_{kl}^2 \Big) \Big( \sum_{k_3=1}^{n} \mathbb{1}_{y_{k_3} \geq y_i} X_{k_3 l} \frac{e^{\eta_{k_3}}}{\sum_{j \in R_i} e^{\eta_j}} \Big) $$

$$- 2 \Big( \sum_{k \in R_i} \frac{e^{\eta_k}}{\sum_{j \in R_i} e^{\eta_j}} X_{kl} \Big) \Big[ \sum_{k \in R_i} \frac{e^{\eta_k}}{\sum_{j \in R_i} e^{\eta_j}} X_{kl} \Big( \sum_{k_3=1}^{n} \mathbb{1}_{k=k_3} X_{k_3 l} \Big) \Big] $$

$$+ 2 \Big( \sum_{k \in R_i} \frac{e^{\eta_k}}{\sum_{j \in R_i} e^{\eta_j}} X_{kl} \Big)^2 \Big[ \sum_{k_3=1}^{n} \Big( \frac{e^{\eta_{k_3}}}{\sum_{j \in R_i} e^{\eta_j}} \Big) \mathbb{1}_{y_{k_3} \geq y_i} X_{k_3 l} \Big] \Big\} $$

# *exchange summation orders; distribute* $\sum_{k_3=1}^{n}$ *into each of the four terms inside* $[\cdot]$

$$= \sum_{i=1}^{n} \delta_i \Big[ \Big( \sum_{k \in R_i} \frac{e^{\eta_k}}{\sum_{j \in R_i} e^{\eta_j}} \Big) X_{kl}^3 $$

$$- \Big( \sum_{k \in R_i} \frac{e^{\eta_k}}{\sum_{j \in R_i} e^{\eta_j}} X_{kl}^2 \Big) \Big( \sum_{k_3 \in R_i} X_{k_3 l} \frac{e^{\eta_{k_3}}}{\sum_{j \in R_i} e^{\eta_j}} \Big) $$

$$- 2 \Big( \sum_{k \in R_i} \frac{e^{\eta_k}}{\sum_{j \in R_i} e^{\eta_j}} X_{kl} \Big) \Big( \sum_{k \in R_i} \frac{e^{\eta_k}}{\sum_{j \in R_i} e^{\eta_j}} X_{kl}^2 \Big) $$

$$+ 2 \Big( \sum_{k \in R_i} \frac{e^{\eta_k}}{\sum_{j \in R_i} e^{\eta_j}} X_{kl} \Big)^2 \Big( \sum_{k_3 \in R_i} \frac{e^{\eta_{k_3}}}{\sum_{j \in R_i} e^{\eta_j}} X_{k_3 l} \Big) \Big] $$

# *simplify the summation over* $k_3$

$$= \sum_{i=1}^{n} \delta_i \Big[ \Big( \sum_{k \in R_i} \frac{e^{\eta_k}}{\sum_{j \in R_i} e^{\eta_j}} \Big) X_{kl}^3 $$

$$- \Big( \sum_{k \in R_i} \frac{e^{\eta_k}}{\sum_{j \in R_i} e^{\eta_j}} X_{kl}^2 \Big) \Big( \sum_{k \in R_i} X_{kl} \frac{e^{\eta_k}}{\sum_{j \in R_i} e^{\eta_j}} \Big) $$

$$- 2 \Big( \sum_{k \in R_i} \frac{e^{\eta_k}}{\sum_{j \in R_i} e^{\eta_j}} X_{kl} \Big) \Big( \sum_{k \in R_i} \frac{e^{\eta_k}}{\sum_{j \in R_i} e^{\eta_j}} X_{kl}^2 \Big) $$

$$+ 2 \Big( \sum_{k \in R_i} \frac{e^{\eta_k}}{\sum_{j \in R_i} e^{\eta_j}} X_{kl} \Big)^2 \Big( \sum_{k \in R_i} \frac{e^{\eta_k}}{\sum_{j \in R_i} e^{\eta_j}} X_{kl} \Big) \Big] $$

# *change notation by replacing* $k_3$ *with* $k$

$$= \sum_{i=1}^{n} \delta_i \Big[ \Big( \sum_{k \in R_i} \frac{e^{\eta_k}}{\sum_{j \in R_i} e^{\eta_j}} \Big) X_{kl}^3 $$

$$- 3 \Big( \sum_{k \in R_i} \frac{e^{\eta_k}}{\sum_{j \in R_i} e^{\eta_j}} X_{kl}^2 \Big) \Big( \sum_{k \in R_i} X_{kl} \frac{e^{\eta_k}}{\sum_{j \in R_i} e^{\eta_j}} \Big) $$

$$+ 2 \Big( \sum_{k \in R_i} \frac{e^{\eta_k}}{\sum_{j \in R_i} e^{\eta_j}} X_{kl} \Big)^3 \Big] $$

# *simplify all relevant terms*

## A.2 Partial Derivative of $r$-th Central Moment

*Proof.* Recall that the $r$-th central moment $C_r$ is defined as:

$$C_r := \sum_{k \in R_i} \frac{e^{\eta_k}}{\sum_{j \in R_i} e^{\eta_j}} \left( X_{kl} - \sum_{k_1 \in R_i} \frac{e^{\eta_{k_1}}}{\sum_{j \in R_i} e^{\eta_j}} X_{k_1 l} \right)^r.$$

We need to show that

$$\frac{\partial}{\partial \beta_l} \left( C_r \right) = C_{r+1} - r \cdot C_2 \cdot C_{r-1}.$$

$$\frac{\partial}{\partial \beta_l} \left( C_r \right)$$

$$= \frac{\partial}{\partial \beta_l} \left[ \sum_{k \in R_i} \frac{e^{\eta_k}}{\sum_{j \in R_i} e^{\eta_j}} \left( X_{kl} - \sum_{k_1 \in R_i} \frac{e^{\eta_{k_1}}}{\sum_{j \in R_i} e^{\eta_j}} X_{k_1 l} \right)^r \right]$$

$$= \sum_{k \in R_i} \frac{\partial}{\partial \beta_l} \left[ \frac{e^{\eta_k}}{\sum_{j \in R_i} e^{\eta_j}} \left( X_{kl} - \sum_{k_1 \in R_i} \frac{e^{\eta_{k_1}}}{\sum_{j \in R_i} e^{\eta_j}} X_{k_1 l} \right)^r \right]$$

# *move the partial differentail operator $\frac{\partial}{\partial \beta_l}(\cdot)$ inside $\sum_{k \in R_i}$*

$$= \sum_{k \in R_i} \left\{ \frac{\partial}{\partial \beta_l} \left( \frac{e^{\eta_k}}{\sum_{j \in R_i} e^{\eta_j}} \right) \left( X_{kl} - \sum_{k_1 \in R_i} \frac{e^{\eta_{k_1}}}{\sum_{j \in R_i} e^{\eta_j}} X_{k_1 l} \right)^r \right.$$

$$\left. + \left( \frac{e^{\eta_k}}{\sum_{j \in R_i} e^{\eta_j}} \right) \frac{\partial}{\partial \beta_l} \left[ \left( X_{kl} - \sum_{k_1 \in R_i} \frac{e^{\eta_{k_1}}}{\sum_{j \in R_i} e^{\eta_j}} X_{k_1 l} \right)^r \right] \right\}$$

# *apply the product rule from calculus*

$$= \sum_{k \in R_i} \left[ \frac{\partial}{\partial \beta_l} \left( \frac{e^{\eta_k}}{\sum_{j \in R_i} e^{\eta_j}} \right) \left( X_{kl} - \sum_{k_1 \in R_i} \frac{e^{\eta_{k_1}}}{\sum_{j \in R_i} e^{\eta_j}} X_{k_1 l} \right)^r \right.$$

$$\left. + \left( \frac{e^{\eta_k}}{\sum_{j \in R_i} e^{\eta_j}} \right) r \left( X_{kl} - \sum_{k_1 \in R_i} \frac{e^{\eta_{k_1}}}{\sum_{j \in R_i} e^{\eta_j}} X_{k_1 l} \right)^{r-1} \frac{\partial}{\partial \beta_l} \left( X_{kl} - \sum_{k_1 \in R_i} \frac{e^{\eta_{k_1}}}{\sum_{j \in R_i} e^{\eta_j}} X_{k_1 l} \right) \right]$$

# *apply the chain rule from calculus to the second term $\frac{d}{dx} y^r = (r-1) y^{r-1} \frac{dy}{dx}$*

$$= \sum_{k \in R_i} \left\{ \frac{\partial}{\partial \beta_l} \left( \frac{e^{\eta_k}}{\sum_{j \in R_i} e^{\eta_j}} \right) \left( X_{kl} - \sum_{k_1 \in R_i} \frac{e^{\eta_{k_1}}}{\sum_{j \in R_i} e^{\eta_j}} X_{k_1 l} \right)^r \right.$$

$$\left. - \left( \frac{e^{\eta_k}}{\sum_{j \in R_i} e^{\eta_j}} \right) r \left( X_{kl} - \sum_{k_1 \in R_i} \frac{e^{\eta_{k_1}}}{\sum_{j \in R_i} e^{\eta_j}} X_{k_1 l} \right)^{r-1} \left[ \sum_{k_1 \in R_i} \frac{\partial}{\partial \beta_l} \left( \frac{e^{\eta_{k_1}}}{\sum_{j \in R_i} e^{\eta_j}} \right) X_{k_1 l} \right] \right\}$$

# *move the partial differential operator $\frac{\partial}{\partial \beta_l}$ inside $sum_{k_1 \in R_i}$; move the negative sign $-$ in the second term to the front*

$$= \sum_{k \in R_i} \left[ \frac{\partial}{\partial \beta_l} \left( \frac{e^{\eta_k}}{\sum_{j \in R_i} e^{\eta_j}} \right) \left( X_{kl} - \sum_{k_1 \in R_i} \frac{e^{\eta_{k_1}}}{\sum_{j \in R_i} e^{\eta_j}} X_{k_1 l} \right)^r \right]$$

$$- \sum_{k \in R_i} \left\{ \left( \frac{e^{\eta_k}}{\sum_{j \in R_i} e^{\eta_j}} \right) r \left( X_{kl} - \sum_{k_1 \in R_i} \frac{e^{\eta_{k_1}}}{\sum_{j \in R_i} e^{\eta_j}} X_{k_1 l} \right)^{r-1} \left[ \sum_{k_1 \in R_i} \frac{\partial}{\partial \beta_l} \left( \frac{e^{\eta_{k_1}}}{\sum_{j \in R_i} e^{\eta_j}} \right) X_{k_1 l} \right] \right\}$$

# *distribute $\sum_{k \in R_i}$ into the two summation terms*

$$= \sum_{k \in R_i} \left[ \frac{\partial}{\partial \beta_l} \left( \frac{e^{\eta_k}}{\sum_{j \in R_i} e^{\eta_j}} \right) \left( X_{kl} - \sum_{k_1 \in R_i} \frac{e^{\eta_{k_1}}}{\sum_{j \in R_i} e^{\eta_j}} X_{k_1 l} \right)^r \right]$$

$$- r \left[ \sum_{k_1 \in R_i} \frac{\partial}{\partial \beta_l} \left( \frac{e^{\eta_{k_1}}}{\sum_{j \in R_i} e^{\eta_j}} \right) X_{k_1 l} \right] \left[ \sum_{k \in R_i} \frac{e^{\eta_k}}{\sum_{j \in R_i} e^{\eta_j}} \left( X_{kl} - \sum_{k_1 \in R_i} \frac{e^{\eta_{k_1}}}{\sum_{j \in R_i} e^{\eta_j}} X_{k_1 l} \right)^{r-1} \right]$$

# *move the two values in the second term outside of $\sum_{k \in R_i}$ because they are independent of $k$*

$$= \sum_{k \in R_i} \left[ \frac{\partial}{\partial \beta_l} \left( \frac{e^{\eta_k}}{\sum_{j \in R_i} e^{\eta_j}} \right) \left( X_{kl} - \sum_{k_1 \in R_i} \frac{e^{\eta_{k_1}}}{\sum_{j \in R_i} e^{\eta_j}} X_{k_1 l} \right)^{r} \right]$$

$$- r \left[ \sum_{k_1 \in R_i} \frac{\partial}{\partial \beta_l} \left( \frac{e^{\eta_{k_1}}}{\sum_{j \in R_i} e^{\eta_j}} \right) X_{k_1 l} \right] C_{r-1}$$

# *simplify by replacing the last value in the second term with $C_{r-1}$ because the central moment definition*

Let us focus on the solution for the subproblem $\frac{\partial}{\partial \beta_l} \left[ e^{\eta_k} / \left( \sum_{j \in R_i} e^{\eta_j} \right) \right]$.

$$\frac{\partial}{\partial \beta_l} \left( \frac{e^{\eta_k}}{\sum_{j \in R_i} e^{\eta_j}} \right)$$

$$= \sum_{k_2=1}^{n} \frac{\partial}{\partial \eta_{k_2}} \left( \frac{e^{\eta_k}}{\sum_{j \in R_i} e^{\eta_j}} \right) \frac{\partial \eta_{k_2}}{\beta_l} \qquad \text{\color{green}\# \textit{apply the chain rule from calculus}}$$

$$= \sum_{k_2=1}^{n} \left[ \frac{1}{\sum_{j \in R_i} e^{\eta_j}} \frac{\partial}{\partial \eta_{k_2}} \left( e^{\eta_k} \right) - \frac{e^{\eta_k}}{\left( \sum_{j \in R_i} e^{\eta_j} \right)^2} \frac{\partial}{\partial \eta_{k_2}} \left( \sum_{j \in R_i} e^{\eta_j} \right) \right] \frac{\partial \eta_{k_2}}{\beta_l}$$

# *apply the quotient rule from calculus*

$$= \sum_{k_2=1}^{n} \left[ \frac{e^{\eta_k}}{\sum_{j \in R_i} e^{\eta_j}} \mathbb{1}_{k=k_2} - \frac{e^{\eta_k} e^{\eta_{k_2}}}{\left( \sum_{j \in R_i} e^{\eta_j} \right)^2} \mathbb{1}_{t_{k_2} \geq t_i} \right] X_{k_2 l} \qquad \text{\color{green}\# \textit{calculate the partial derivative}}$$

$$= \sum_{k_2=1}^{n} \frac{e^{\eta_k}}{\sum_{j \in R_i} e^{\eta_j}} \mathbb{1}_{k=k_2} X_{k_2 l} - \sum_{k_2=1}^{n} \frac{e^{\eta_k} e^{\eta_{k_2}}}{\left( \sum_{j \in R_i} e^{\eta_j} \right)^2} \mathbb{1}_{t_{k_2} \geq t_i} X_{k_2 l}$$

# *distribute $\sum_{k_2=1}^{n} (\cdot) X_{k_2 l}$ into the two terms inside $[\cdot]$*

$$= \frac{e^{\eta_k}}{\sum_{j \in R_i} e^{\eta_j}} X_{kl} - \sum_{k_2 \in R_i} \frac{e^{\eta_k} e^{\eta_{k_2}}}{\left( \sum_{j \in R_i} e^{\eta_j} \right)^2} X_{k_2 l} \qquad \text{\color{green}\# \textit{evaluate the two summations}}$$

$$= \frac{e^{\eta_k}}{\sum_{j \in R_i} e^{\eta_j}} \left( X_{kl} - \sum_{k_2 \in R_i} \frac{e^{\eta_{k_2}}}{\sum_{j \in R_i} e^{\eta_j}} X_{k_2 l} \right) \qquad \text{\color{green}\# \textit{evaluate the two summations}}$$

Let us now plug this result back into the original problem:

$$\frac{\partial}{\partial \beta_l}(C_r)$$

$$= \sum_{k \in R_i} \left[ \frac{\partial}{\partial \beta_l} \left( \frac{e^{\eta_k}}{\sum_{j \in R_i} e^{\eta_j}} \right) \left( X_{kl} - \sum_{k_1 \in R_i} \frac{e^{\eta_{k_1}}}{\sum_{j \in R_i} e^{\eta_j}} X_{k_1 l} \right)^r \right]$$

$$- r \left[ \sum_{k_1 \in R_i} \frac{\partial}{\partial \beta_l} \left( \frac{e^{\eta_{k_1}}}{\sum_{j \in R_i} e^{\eta_j}} \right) X_{k_1 l} \right] C_{r-1}$$

*# pick up from where we left for the original problem*

$$= \sum_{k \in R_i} \left[ \frac{e^{\eta_k}}{\sum_{j \in R_i} e^{\eta_j}} \left( X_{kl} - \sum_{k_2 \in R_i} \frac{e^{\eta_{k_2}}}{\sum_{j \in R_i} e^{\eta_j}} X_{k_2 l} \right) \left( X_{kl} - \sum_{k_1 \in R_i} \frac{e^{\eta_{k_1}}}{\sum_{j \in R_i} e^{\eta_j}} X_{k_1 l} \right)^r \right]$$

$$- r \left[ \sum_{k_1 \in R_i} \frac{e^{\eta_{k_1}}}{\sum_{j \in R_i} e^{\eta_j}} \left( X_{k_1 l} - \sum_{k_2 \in R_i} \frac{e^{\eta_{k_2}}}{\sum_{j \in R_i} e^{\eta_j}} X_{k_2 l} \right) X_{k_1 l} \right] C_{r-1}$$

*# plug in the solution to the subproblem*

$$= \sum_{k \in R_i} \left[ \frac{e^{\eta_k}}{\sum_{j \in R_i} e^{\eta_j}} \left( X_{kl} - \sum_{k_1 \in R_i} \frac{e^{\eta_{k_1}}}{\sum_{j \in R_i} e^{\eta_j}} X_{k_1 l} \right)^{r+1} \right]$$

$$- r \left[ \sum_{k_1 \in R_i} \frac{e^{\eta_{k_1}}}{\sum_{j \in R_i} e^{\eta_j}} \left( X_{k_1 l} - \sum_{k_2 \in R_i} \frac{e^{\eta_{k_2}}}{\sum_{j \in R_i} e^{\eta_j}} X_{k_2 l} \right) X_{k_1 l} \right] C_{r-1}$$

*# for the first term, change $k_2$ into $k_1$ because both are dummy variables that are independent from each other*

$$= \sum_{k \in R_i} \left[ \frac{e^{\eta_k}}{\sum_{j \in R_i} e^{\eta_j}} \left( X_{kl} - \sum_{k_1 \in R_i} \frac{e^{\eta_{k_1}}}{\sum_{j \in R_i} e^{\eta_j}} X_{k_1 l} \right)^{r+1} \right]$$

$$- r \left[ \sum_{k_1 \in R_i} \frac{e^{\eta_{k_1}}}{\sum_{j \in R_i} e^{\eta_j}} \left( X_{k_1 l}^2 - \sum_{k_2 \in R_i} \frac{e^{\eta_{k_2}}}{\sum_{j \in R_i} e^{\eta_j}} X_{k_1 l} X_{k_2 l} \right) \right] C_{r-1}$$

*# for the second term, move $X_{k_1 l}$ inside $(\cdot)$*

$$= \sum_{k \in R_i} \left[ \frac{e^{\eta_k}}{\sum_{j \in R_i} e^{\eta_j}} \left( X_{kl} - \sum_{k_1 \in R_i} \frac{e^{\eta_{k_1}}}{\sum_{j \in R_i} e^{\eta_j}} X_{k_1 l} \right)^{r+1} \right]$$

$$- r \left( \sum_{k_1 \in R_i} \frac{e^{\eta_{k_1}}}{\sum_{j \in R_i} e^{\eta_j}} X_{k_1 l}^2 - \sum_{k_1 \in R_i} \sum_{k_2 \in R_i} \frac{e^{\eta_{k_1}} e^{\eta_{k_2}}}{\left( \sum_{j \in R_i} e^{\eta_j} \right)^2} X_{k_1 l} X_{k_2 l} \right) C_{r-1}$$

*# for the second term, distribute $\sum_{k_1 \in R_i} \frac{e^{\eta_{k_1}}}{\sum_{j \in R_i}}$ into the two terms inside $(\cdot)$*

$$= \sum_{k \in R_i} \left[ \frac{e^{\eta_k}}{\sum_{j \in R_i} e^{\eta_j}} \left( X_{kl} - \sum_{k_1 \in R_i} \frac{e^{\eta_{k_1}}}{\sum_{j \in R_i} e^{\eta_j}} X_{k_1 l} \right)^{r+1} \right]$$

$$- r \left[ \sum_{k_1 \in R_i} \frac{e^{\eta_{k_1}}}{\sum_{j \in R_i} e^{\eta_j}} X_{k_1 l}^2 - \left( \sum_{k_1 \in R_i} \frac{e^{\eta_{k_1}}}{\sum_{j \in R_i} e^{\eta_j}} X_{k_1 l} \right)^2 \right] C_{r-1}$$

*# in the second term, $k_1$ and $k_2$ are independent dummy variables, so we can turn a double sum of products into a products of sums; we can further simplify this into a square of a sum because the two terms equal to the same value*

$$= C_{r+1} - r \cdot C_2 \cdot C_{r-1} \qquad \text{*# simplify by using the definition of central moment*}$$

$$\square$$

## A.3 First and Second Order Partial Derivatives Are Lipschitz-Continuous

*Proof.* To show that the first and second order partial derivatives are Lipschitz-continuous, we need to show that the second and third order partial derivatives are bounded, respectively.

**First order partial derivative is Lipschitz-continuous**   To show that the first order partial derivative with respect to each coordinate is Lipschitz, we need to show that the second order partial derivative with respect to each coordinate is bounded. Recall that the second order partial derivative with respect to the $l$-th coordinate can be expressed as:

$$\frac{\partial^2 \ell(\boldsymbol{\beta})}{\partial \beta_l^2} = \sum_{i=1}^n \delta_i \Big[ \sum_{k \in R_i} \frac{e^{\eta_k}}{\sum_{j \in R_i} e^{\eta_j}} X_{kl}^2 - \Big( \sum_{k \in R_i} \frac{e^{\eta_k}}{\sum_{j \in R_i} e^{\eta_j}} X_{kl} \Big)^2 \Big]$$

It suffices to show that each term inside the bracket is bounded.

If we interpret the expression probabilistically, then the coefficients $\frac{e^{\eta_k}}{\sum_{j \in R_i} e^{\eta_j}}$ in front of $X_{kl}^2$ and $X_{kl}$ can be thought of as the probability of a particular distribution because all terms are greater than or equal to $0$ and sum up to $1$.

For notational convenience, let us use $\boldsymbol{a}$ to denote the probability of this specific distribution with $a_k = \frac{e^{\eta_k}}{\sum_{j \in R_i} e^{\eta_j}}$. Then we can rewrite each term inside $[\cdot]$ as:

$$\sum_{k \in R_i} \frac{e^{\eta_k}}{\sum_{j \in R_i} e^{\eta_j}} X_{kl}^2 - \Big( \sum_{k \in R_i} \frac{e^{\eta_k}}{\sum_{j \in R_i} e^{\eta_j}} X_{kl} \Big)^2 = \sum_{k \in R_i} a_k X_{kl}^2 - \Big( \sum_{k \in R_i} a_k X_{kl} \Big)^2$$

The right-hand side is nothing but the variance of $\{X_{kl}\}_{k \in R_i}$ with respect to the distribution $\{a_k\}_{k \in R_i}$. Since the variance is always non-negative, we have

$$\sum_{k \in R_i} a_k X_{kl}^2 - \Big( \sum_{k \in R_i} a_k X_{kl} \Big)^2 \geq 0$$

Let us now denote $a := \min_{k \in R_i} X_{kl}$ as the minimum of this given set, $b := \max_{k \in R_i} X_{kl}$ as the maximum of this given set, $Z$ as a random variable with values restricted to $[a, b]$. We are going to show that

$$\sum_{k \in R_i} a_k X_{kl}^2 - \Big( \sum_{k \in R_i} a_k X_{kl} \Big)^2 \leq \max_Z \big[ \mathbb{E}[Z^2] - (\mathbb{E}[Z])^2 \big]. \tag{23}$$

We achieve this through two steps.

First, suppose the random variable can only take finite $|R_i|$ number of values $\{Z_1, Z_2, ..., Z_{|R_i|}\}$ with probability $\{p_1, p_2, ..., p_{|R_i|}\}$, where $|R_i|$ is the cardinality of the set $R_i$. Then, we have

$$\sum_{k \in R_i} a_k X_{kl}^2 - \Big( \sum_{k \in R_i} a_k X_{kl} \Big)^2 \leq \max_{Z, \boldsymbol{p}} \big[ \mathbb{E}_{\boldsymbol{p}}[Z^2] - (\mathbb{E}_{\boldsymbol{p}}[Z])^2 \big], \tag{24}$$

where the expectation is taken with respect to the distribution $\boldsymbol{p}$. The above inequality holds because the left-hand side is a specific instance of the expression inside $[\cdot]$, so taking $\max(\cdot)$ produces the inequality above.

Next, if we drop the assumption that the random variable $Z$ can only take $|R_i|$ number of values, then the maximum variance we can achieve is no smaller than before. Mathematically, this means that

$$\max_{Z, \boldsymbol{p}} \big[ \mathbb{E}_{\boldsymbol{p}}[Z^2] - (\mathbb{E}_{\boldsymbol{p}}[Z])^2 \big] \leq \max_Z \big[ \mathbb{E}[Z^2] - (\mathbb{E}[Z])^2 \big]. \tag{25}$$

Combining Inequality (24) and Inequality (25), we arrive at Inequality (23). Lastly, note that the Popoviciu Inequality [58] tells us that for a bounded random variable restricted to $[a, b]$, the maximum variance it can achieve is $\frac{(a-b)^2}{4}$, *i.e.*,

$$\max_Z \big[ \mathbb{E}[Z^2] - (\mathbb{E}[Z])^2 \big] \leq \frac{(b-a)^2}{4}.$$

This allows us to conclude that

$$\sum_{k \in R_i} a_k X_{kl}^2 - \Big( \sum_{k \in R_i} a_k X_{kl} \Big)^2 \leq \frac{(b-a)^2}{4} = \frac{(\max_{k \in R_i} X_{kl} - \min_{k \in R_i} X_{kl})^2}{4}.$$

Plugging in this inequality into the expression of the second order partial derivative, we can show that the second order partial derivative is bounded:

$$0 \le \frac{\partial^2 \ell(\boldsymbol{\beta})}{\partial \beta_l^2} = \sum_{i=1}^n \delta_i \Big[ \sum_{k_1 \in R_i} \frac{e^{\eta_{k_1}}}{\sum_{j \in R_i} e^{\eta_j}} X_{k_1 l}^2 - \Big( \sum_{k_1 \in R_i} \frac{e^{\eta_{k_1}}}{\sum_{j \in R_i} e^{\eta_j}} X_{k_1 l} \Big)^2 \Big]$$

$$\le \frac{1}{4} \sum_{i=1}^n \delta_i (\max_{k \in R_i} X_{kl} - \min_{k \in R_i} X_{kl})^2$$

**Second order partial derivative is Lipschitz-continuous**   This proof is similar to the first part above. We need to show that the third order partial derivative with respect to each coordinate is bounded. Recall that the third order partial derivative can be expressed as:

$$\frac{\partial^3 \ell(\boldsymbol{\beta})}{\partial \beta_l^3} = \sum_{i=1}^n \delta_i \Big[ \sum_{k \in R_i} \frac{e^{\eta_k}}{\sum_{j \in R_i} e^{\eta_j}} X_{kl}^3 + 2 \Big( \sum_{k \in R_i} \frac{e^{\eta_k}}{\sum_{j \in R_i} e^{\eta_j}} X_{kl} \Big)^3$$

$$- 3 \Big( \sum_{k \in R_i} \frac{e^{\eta_k}}{\sum_{j \in R_i} e^{\eta_j}} X_{kl}^2 \Big) \Big( \sum_{k \in R_i} \frac{e^{\eta_k}}{\sum_{j \in R_i} e^{\eta_j}} X_{kl} \Big) \Big]$$

As we have done in the first part, if we use $\boldsymbol{a}$ to denote the probability of this specific distribution with $a_k = \frac{e^{\eta_k}}{\sum_{j \in R_i} e^{\eta_j}}$, we can rewrite each term inside $[\cdot]$ as

$$\sum_{k \in R_i} \frac{e^{\eta_k}}{\sum_{j \in R_i} e^{\eta_j}} X_{kl}^3 + 2 \Big( \sum_{k \in R_i} \frac{e^{\eta_k}}{\sum_{j \in R_i} e^{\eta_j}} X_{kl} \Big)^3 - 3 \Big( \sum_{k \in R_i} \frac{e^{\eta_k}}{\sum_{j \in R_i} e^{\eta_j}} X_{kl}^2 \Big) \Big( \sum_{k \in R_i} \frac{e^{\eta_k}}{\sum_{j \in R_i} e^{\eta_j}} X_{kl} \Big)$$

$$= \sum_{k \in R_i} a_k X_{kl}^3 + 2 \Big( \sum_{k \in R_i} a_k X_{kl} \Big)^3 - 3 \Big( \sum_{k \in R_i} a_k X_{kl}^2 \Big) \Big( \sum_{k \in R_i} a_k X_{kl} \Big)$$

Similarly, as in the first part, let us now denote $a := \min_{k \in R_i} X_{kl}$ as the minimum of this given set, $b := \max_{k \in R_i} X_{kl}$ as the maximum of this given set, $Z$ as a random variable with values restricted to $[a, b]$. Using the exact same logic, we have

$$\Big| \sum_{k \in R_i} a_k X_{kl}^3 + 2 \Big( \sum_{k \in R_i} a_k X_{kl} \Big)^3 - 3 \Big( \sum_{k \in R_i} a_k X_{kl}^2 \Big) \Big( \sum_{k \in R_i} a_k X_{kl} \Big) \Big|$$

$$\le \max_Z |\mathbb{E}[Z^3] + 2\mathbb{E}[Z]^3 - 3\mathbb{E}[Z^2]\mathbb{E}[Z]|||$$

$$= \max_Z |\mathbb{E}[(Z - \mathbb{E}[Z])^3]|$$

The expression inside $|\cdot|$ on the right-hand side is known as the third central moment (skewedness) in statistics. Fortunately, we can derive an explicit formula for the maximum of the absolute third central moment of a bounded variable.

According to [61], we have the following inequality involving the second and third central moment:

$$\mathbb{E}[(Z - \mathbb{E}[Z])^2] + \Big( \frac{\mathbb{E}[(Z - \mathbb{E}[Z])^3]}{2\mathbb{E}[(Z - \mathbb{E}[Z])^2]} \Big)^2 \le \frac{1}{4}(b - a)^2$$

From this, we can derive an upper bound on the third central moment:

$$|\mathbb{E}[(Z - \mathbb{E}[Z])^3]| \le 2\mathbb{E}[(Z - \mathbb{E}[Z])^2] \sqrt{\frac{1}{4}(b - a)^2 - \mathbb{E}[(Z - \mathbb{E}[Z])^2]}$$

For notational convenience, let us denote $V := \mathbb{E}[(Z - \mathbb{E}[Z])^2]$, then the right-hand side above can be expressed as a function of V:

$$f(V) := 2V \sqrt{\frac{1}{4}(b - a)^2 - V} = \sqrt{(b - a)^2 V^2 - 4V^3}$$

Because $V$ is the variance, $V \in [0, \frac{1}{4}(b - a)^2]$. Additionally, because $\sqrt{\cdot}$ is monotonically increasing, the maximum is achieved either at the points where the first order derivative of $(b - a)^2 V^2 - 4V^3$

with respect to $V$ is 0 or at the boundary, 0 and $\frac{1}{4}(b-a)^2$. Let us calculate the points where the first order derivative is 0:

$$\frac{d}{dV}\left[(b-a)^2 V^2 - 4V^3\right] = 2(b-a)^2 V - 12V^2 = 0 \iff V = \frac{(b-a)^2}{6} \quad \text{or} \quad V = 0$$

To obtain the maximum value achievable, we calculate the values at points $V = 0$, $V = \frac{(b-a)^2}{6}$, and $V = \frac{1}{4}(b-a)^2$ and pick the maximum value afterwards:

$$f(0) = 2 \times 0 \sqrt{\frac{1}{4}(b-a)^2 - 0} = 0$$

$$f(\frac{1}{6}(b-a)^2) = 2 \times \frac{1}{6}(b-a)^2 \sqrt{\frac{1}{4}(b-a)^2 - \frac{1}{6}(b-a)^2} = \frac{1}{6\sqrt{3}}|b-a|^3$$

$$f(\frac{1}{4}(b-a)^2) = 2 \times \frac{1}{4}(b-a)^2 \sqrt{\frac{1}{4}(b-a)^2 - \frac{1}{4}(b-a)^2} = 0$$

Therefore, the maximum value achievable is $\frac{1}{6\sqrt{3}}|b-a|^3$.

We now show that upper bound on the absolute value of the third central moment is actually tight by providing with a concrete example. For a random variable Z, let $\mathbb{P}[Z = a] = \frac{1}{4}$, $\mathbb{P}[Z = b] = \frac{1}{4}$, and $\mathbb{P}[Z = \frac{a+b}{2}] = \frac{1}{2}$. We can verify that $\mathbb{E}[(Z - \mathbb{E}[Z])^3] = \frac{1}{6\sqrt{3}}|b-a|^3$, thus proving that this upper bound is indeed tight.

This helps us to arrive at the following inequality:

$$|\sum_{k \in R_i} a_k X_{kl}^3 + 2(\sum_{k \in R_i} a_k X_{kl})^3 - 3(\sum_{k \in R_i} a_k X_{kl}^2)(\sum_{k \in R_i} a_k X_{kl})| \leq \frac{1}{6\sqrt{3}}|\max_{k \in R_i} X_{kl} - \min_{k \in R_i} X_{kl}|^3$$

Therefore, for the upper bound of the third order partial derivative, we have the following explicit formula:

$$|\frac{\partial^3 \ell(\boldsymbol{\beta})}{\partial \beta_l^3}| \leq \frac{1}{6\sqrt{3}} \sum_{i=1}^n \delta_i |\max_{k \in R_i} X_{kl} - \min_{k \in R_i} X_{kl}|^3$$

$\square$

### A.4 Analytical Solution to the Cubic Surrogate Problem

Let $f(x)$ be a convex function whose first, second, and third derivatives all exist. Let $h_x(\Delta x) := f(x) + f'(x)\Delta x + \frac{1}{2}f''(x)\Delta x^2 + \frac{1}{6}L_3|\Delta x|^3$ be the cubic surrogate function [53] of $f(x)$, where $L_3 > 0$ is the Lipschitz-constant of the second derivative $f''(x)$. Then, the minimum of this surrogate function is achieved at the following point:

$$\Delta\hat{x} = \underset{\Delta x}{\operatorname{argmin}}\, h_x(\Delta x) = \operatorname{sgn}(f'(x)) \cdot \frac{f''(x) - \sqrt{(f''(x))^2 + 2L_3|f'(x)|}}{L_3} \tag{26}$$

*Proof.* We discuss three cases: $f'(x) > 0$, $f'(x) < 0$, and $f'(x) = 0$.

**Case 1** $f'(x) > 0$.

If $f'(x) > 0$, then $\Delta\hat{x} < 0$. For the sake of contradiction, suppose $\Delta\hat{x} > 0$, which means $h_x(\Delta x)$ achieves its minimum at some point with $\Delta x > 0$. However, we arrive at a contradiction because

$$h_x(0) = f(x) < f(x) + f'(x)\Delta x + \frac{1}{2}f''(x)\Delta x^2 + \frac{1}{6}L_3|\Delta x|^3 = h_x(\Delta x), \quad \text{for } \Delta x > 0.$$

Therefore, the minimum is achieved either at $\Delta x = 0$ or $\Delta x < 0$. However, since $\Delta x^2$ and $|\Delta x|^3$ grow slower than $|\Delta x|$ when $\Delta x$ is close to 0, there exists some $\Delta x < 0$ such that $f'(x)\Delta x + \frac{1}{2}f''(x)\Delta x^2 + \frac{1}{6}L_3|\Delta x|^3 < 0$. Thus, $h_x(0)$ cannot be the minimum value, and we are left with the

minimum value achieved at some $\Delta x < 0$. If $\Delta x < 0$, $h_x(\Delta x) = f(x) + f'(x)\Delta x + \frac{1}{2}f''(x)\Delta x^2 - \frac{1}{6}L_3(\Delta x)^3$. Note that the second order derivative of $h_x(\Delta x)$ is greater than or equal to 0 since

$$\frac{d^2}{d\Delta x^2}h_x(\Delta x) = f''(x) - L_3\Delta x \geq 0.$$

# $f''(x) \geq 0$ because $f(x)$ is convex, $0 \leq |f'''(x)| \leq L_3$, and $\Delta x < 0$

Therefore, $h_x(\Delta x)$ is a convex function with respect to $\Delta x$ when $\Delta x < 0$, and its minimum value is achieved when the first order derivative is 0. When the derivative with respect to $\Delta x$ is 0, we have

$$f'(x) + f''(x)\Delta x - \frac{1}{2}(\Delta x)^2 = 0 \iff \Delta x = \frac{f''(x) \pm \sqrt{(f''(x))^2 + 2L_3 f'(x)}}{L_3}.$$

Since $f'(x) > 0$, only one root $\frac{f''(x) - \sqrt{(f''(x))^2 + 2L_3 f'(x)}}{L_3}$ satisfying the condition $\Delta x < 0$. Thus, when $f'(x) < 0$, we have

$$\Delta\hat{x} = \frac{f''(x) - \sqrt{(f''(x))^2 + 2L_3 f'(x)}}{L_3}$$

**Case 2** $f'(x) < 0$.

If $f'(x) < 0$, we have $\Delta\hat{x} > 0$ using the same logic as above. If $\Delta x > 0$, $h_x(\Delta x) = f(x) + f'(x)\Delta x + \frac{1}{2}f''(x)\Delta x^2 + \frac{1}{6}L_3(\Delta x)^3$. Note that the second order derivative of $h_x(\Delta x)$ is also greater than or equal to 0 since

$$\frac{d^2}{d\Delta x^2}h_x(\Delta x) = f''(x) + L_3\Delta x \geq 0.$$

# $f''(x) \geq 0$ because $f(x)$ is convex, $L_3 > 0$, and $\Delta x > 0$

Therefore, $h_x(\Delta x)$ is a convex function with respect to $\Delta x$ when $\Delta x > 0$, and its minimum value is achieved when the first derivative is 0. When the derivative with respect to $\Delta x$ is 0, we have

$$f'(x) + f''(x)\Delta x + \frac{1}{2}(\Delta x)^2 = 0 \iff \Delta x = \frac{-f''(x) \pm \sqrt{(f''(x))^2 - 2L_3 f'(x)}}{L_3}.$$

Since $f'(x) < 0$, only one root $\frac{-f''(x) + \sqrt{(f''(x))^2 - 2L_3 f'(x)}}{L_3}$ satisfying the condition $\Delta x > 0$. Thus, when $f'(x) < 0$, we have

$$\Delta\hat{x} = \frac{-f''(x) + \sqrt{(f''(x))^2 - 2L_3 f'(x)}}{L_3}$$

**Case 3** $f'(x) = 0$.

When $f'(x) = 0$, the minimum value of $h_x(\Delta x)$ is achieved at $\Delta x = 0$.

The explicit formulas for the three cases above can be unified into one succinct formula below:

$$\Delta\hat{x} = \text{sgn}(f'(x)) \cdot \frac{f''(x) - \sqrt{(f''(x))^2 + 2L_3|f'(x)|}}{L_3}$$

$\square$

### A.5 Analytical Solution to the $\ell_1$-regularized Quadratic and Cubic Surrogate Problems

$\ell_1$**-regularized quadratic surrogate problem**  We have the following $\ell_1$-regularized quadratic surrogate problem:

$$\Delta\hat{x} = \underset{\Delta x}{\arg\min}\, a\Delta x + \frac{1}{2}b\Delta x^2 + \lambda_1|c + \Delta x|.$$

The solution for the above problem is

$$\Delta\hat{x} = \begin{cases} -(a - \lambda_1)/b & \text{if} \quad bc - a < -\lambda_1 \\ -(a + \lambda_1)/b & \text{if} \quad bc - a > \lambda_1 \\ -c & \text{otherwise} \end{cases}.$$

*Proof.* Since the function $a\Delta x + \frac{1}{2}b\Delta x^2 + \lambda_1|c + \Delta x|$ is convex, the condition for this function to achieve the minimum value is for its differential to include 0. The differential of this function is:

$$\partial_{\Delta x}\left(a\Delta x + \frac{1}{2}b\Delta x^2 + \lambda_1|c + \Delta x|\right) = \begin{cases} a + b\Delta x + \lambda_1 & \text{if} \quad \Delta x > -c \\ a + b\Delta x - \lambda_1 & \text{if} \quad \Delta x < -c \\ a + b\Delta x + [-\lambda_1, \lambda_1] & \text{if} \quad \Delta x = -c \end{cases}$$

1. For the first condition, if the differential contains 0, we have

$$a + b\Delta x + \lambda_1 = 0 \quad \Rightarrow \quad \Delta x = -\frac{a + \lambda_1}{b}$$

However, because we require $\Delta x > -c$, we have

$$-\frac{a + \lambda_1}{b} > -c \Rightarrow bc - a > \lambda_1$$

2. For the second condition, if the differential contains 0, we have

$$a + b\Delta x - \lambda_1 = 0 \quad \Rightarrow \quad \Delta x = -\frac{a - \lambda_1}{b}$$

However, because we require $\Delta x < -c$, we have

$$-\frac{a - \lambda_1}{b} < -c \Rightarrow bc - a < -\lambda_1$$

3. For the third condition, if the differential contains 0, we have

$$0 \in a + b\Delta x + [-\lambda_1, \lambda_1] \quad \Rightarrow a + b\Delta x - \lambda_1 \leq 0 \leq a + b\Delta x + \lambda_1$$

However, because we require $\Delta x = -c$, we have

$$-\lambda_1 \leq bc - a \leq \lambda_1$$

$\square$

**$\ell_1$-regularized cubic surrogate problem**  We have the following $\ell_1$-regularized cubic surrogate problem:

$$\Delta\hat{x} = \operatorname*{argmin}_{\Delta x} a\Delta x + \frac{1}{2}b\Delta x^2 + \frac{1}{6}c|x|^3 + \lambda_1|d + \Delta x|. \tag{27}$$

The solution to the above problem is

$$\Delta\hat{x} = \begin{cases} \operatorname{sgn}(d)\left(-b + \sqrt{b^2 - 2c(\operatorname{sgn}(d)a + \lambda_1)}\right)/c & \text{if} \quad \operatorname{sgn}(d)a + \lambda_1 \leq 0 \\ \operatorname{sgn}(d)\left(b + \sqrt{b^2 + 2c(\operatorname{sgn}(d)a - \lambda_1)}\right)/c & \text{if} \quad \operatorname{sgn}(d)(a - bd) - \frac{1}{2}cd^2 > \lambda_1 \\ \operatorname{sgn}(d)\left(b + \sqrt{b^2 + 2c(\operatorname{sgn}(d)a + \lambda_1)}\right)/c & \text{if} \quad \operatorname{sgn}(d)(a - bd) - \frac{1}{2}cd^2 < -\lambda_1 \\ -d & \text{otherwise} \end{cases}.$$

*Proof.* Like the first part, since the function $a\Delta x + \frac{1}{2}b\Delta x^2 + \frac{1}{6}c|x|^3 + \lambda_1|d + \Delta x|$ is convex, the condition for this function to achieve the minimum value is for its differential to include 0. We discuss the differential of this function in two cases: $d \geq 0$ and $d < 0$.

- When $d \geq 0$, the differential of this function is:

$$\partial_{\Delta x}\left(a\Delta x + \frac{1}{2}b\Delta x^2 + \frac{1}{6}c|x|^3 + \lambda_1|d + \Delta x|\right)$$

$$= \begin{cases} a + b\Delta x + \frac{1}{2}c\Delta x^2 + \lambda_1 & \text{if} \quad \Delta x > 0 \\ a + b\Delta x + \frac{1}{2}c[-\Delta x^2, \Delta x^2] + \lambda_1 & \text{if} \quad \Delta x = 0 \\ a + b\Delta x - \frac{1}{2}c\Delta x^2 + \lambda_1 & \text{if} \quad -d < \Delta x < 0 \\ a + b\Delta x - \frac{1}{2}c\Delta x^2 + [-\lambda_1, \lambda_1] & \text{if} \quad \Delta x = -d \\ a + b\Delta x - \frac{1}{2}c\Delta x^2 - \lambda_1 & \text{if} \quad \Delta x < -d \end{cases}$$

We discuss these 5 cases one by one.

1. For the first condition, if the differential contains 0, we have
$$a + b\Delta\hat{x} + \frac{1}{2}c\Delta\hat{x}^2 + \lambda_1 = 0 \Rightarrow \Delta\hat{x} = \frac{-b \pm \sqrt{b^2 - 2c(a + \lambda_1)}}{c}$$
However, because we require $\Delta x > 0$, we have
$$\lambda_1 = -(a + b\Delta x + \frac{1}{2}c\Delta x^2) < -a$$
This means that we can only have one root because the other root violates $\Delta x > 0$:
$$\Delta\hat{x} = \frac{-b + \sqrt{b^2 - 2c(a + \lambda_1)}}{c}$$

2. For the second condition, if the differential contains 0, we have
$$0 \in a + b\Delta\hat{x} + \frac{1}{2}c[-\Delta\hat{x}^2, \Delta\hat{x}^2] + \lambda_1$$
$$\Rightarrow a + b\Delta\hat{x} - \frac{1}{2}c\Delta\hat{x}^2 + \lambda_1 \leq 0 \leq a + b\Delta\hat{x} + \frac{1}{2}c\Delta\hat{x}^2 + \lambda_1$$
However, because we require $\Delta x = 0$, we have
$$a + \lambda_1 = 0$$

3. For the third condition, if the differential contains 0, we have
$$a + b\Delta\hat{x} - \frac{1}{2}c\Delta\hat{x}^2 + \lambda_1 = 0 \Rightarrow \Delta\hat{x} = \frac{-b \pm \sqrt{b^2 + 2c(a + \lambda_1)}}{-c}$$
However, because we require $-d < \Delta x < 0$, we have
$$\lambda_1 = -\left(a + b\Delta\hat{x} - \frac{1}{2}c\Delta\hat{x}^2\right) > -a \Rightarrow a + \lambda_1 > 0$$
This means that we can only have one root because the other root violates the condition $-d < \Delta x < 0$:
$$\Delta\hat{x} = \frac{b - \sqrt{b^2 + 2c(a + \lambda_1)}}{c}$$
Moreover, since the root is between $-d$ and 0, and the coefficients in front of $\Delta x$, $-\frac{1}{2}c$, is negative, we have $a - bd - \frac{1}{2}cd^2 + \lambda_1 < 0$.

4. For the fourth condition, if the differential contains 0, we have
$$0 \in a + b\Delta\hat{x} - \frac{1}{2}c\Delta\hat{x}^2 + [-\lambda_1, \lambda_1]$$
$$\Rightarrow a + b\Delta\hat{x} - \frac{1}{2}c\Delta\hat{x}^2 - \lambda_1 \leq 0 \leq a + b\Delta\hat{x} - \frac{1}{2}c\Delta\hat{x}^2 + \lambda_1$$
However, because we require $\Delta x = -d$, we have
$$a - bd - \frac{1}{2}cd^2 - \lambda_1 \leq 0 \leq a - bd - \frac{1}{2}cd^2 - \lambda_1 \Rightarrow |a - bd - \frac{1}{2}cd^2| \leq 0$$

5. For the fifth condition, if the differential contains 0, we have
$$a + b\Delta\hat{x} - \frac{1}{2}c\Delta\hat{x}^2 - \lambda_1 = 0 \Rightarrow \Delta\hat{x} = \frac{-b \pm \sqrt{b^2 + 2c(a - \lambda_1)}}{-c}$$
However, because we require $\Delta x < -d$, we have
$$\lambda_1 = a + b\Delta\hat{x} - \frac{1}{2}c\Delta\hat{x}^2 < a \Rightarrow a - \lambda_1 > 0$$
This means that we can only have one root because the other root violates the condition $\Delta x < -d \leq 0$:
$$\Delta\hat{x} = \frac{b - \sqrt{b^2 + 2c(a - \lambda_1)}}{c}$$
Moreover, because the left root is less than $-d$ and the coefficient in front $\Delta x^2$, $-\frac{1}{2}c$, is negative, we have $a - bd - \frac{1}{2}cd^2 - \lambda_1 > 0$.

- When $d < 0$, the differential of this function is

$$\partial_{\Delta x} \left( a\Delta x + \frac{1}{2}b\Delta x^2 + \frac{1}{6}c|x|^3 + \lambda_1 |d + \Delta x| \right)$$

$$= \begin{cases} a + b\Delta x + \frac{1}{2}c\Delta x^2 + \lambda_1 & \text{if} \quad \Delta x > -d \\ a + b\Delta x + \frac{1}{2}c\Delta x^2 + [-\lambda_1, \lambda_1] & \text{if} \quad \Delta x = -d \\ a + b\Delta x + \frac{1}{2}c\Delta x^2 - \lambda_1 & \text{if} \quad 0 < \Delta x < -d \\ a + b\Delta x + \frac{1}{2}c[-\Delta x^2, \Delta x^2] - \lambda_1 & \text{if} \quad \Delta x = 0 \\ a + b\Delta x - \frac{1}{2}c\Delta x^2 - \lambda_1 & \text{if} \quad \Delta x < 0 \end{cases}$$

Similar to the previous part when $d \geq 0$, we discuss the 5 cases one by one but omit the details because the logic and the reasoning process are exactly the same:

1. For the first condition, if the differential contains 0, we have

$$\Delta\hat{x} = \frac{-b + \sqrt{b^2 - 2c(a + \lambda_1)}}{c} \quad \text{and} \quad a - bd + \frac{1}{2}cd^2 + \lambda_1 < 0$$

2. For the second condition, if the differential contains 0, we have

$$\Delta\hat{x} = -d \quad \text{and} \quad |a - bd + \frac{1}{2}cd^2| \leq \lambda_1$$

3. For the third condition, if the differential contains 0, we have

$$\Delta\hat{x} = \frac{-b + \sqrt{b^2 - 2c(a - \lambda_1)}}{c}$$

4. For the fourth condition, if the differential contains 0, we have

$$\Delta\hat{x} = 0 \quad \text{and} \quad a - \lambda_1 = 0$$

5. For the fifth condition, if the differential contains 0, we have

$$\Delta\hat{x} = \frac{b - \sqrt{b^2 + 2c(a - \lambda_1)}}{c} \quad \text{and} \quad a - \lambda_1 > 0$$

We can combine the two situations where $d \geq 0$ and $d < 0$ and obtain a unified formula:

$$\Delta\hat{x} = \begin{cases} \text{sgn}(d)\left(-b + \sqrt{b^2 - 2c(\text{sgn}(d)a + \lambda_1)}\right)/c & \text{if} \quad \text{sgn}(d)a + \lambda_1 < 0 \\ 0 & \text{if} \quad \text{sgn}(d)a + \lambda_1 = 0 \\ \text{sgn}(d)\left(b + \sqrt{b^2 + 2c(\text{sgn}(d)a - \lambda_1)}\right)/c & \text{if} \quad \text{sgn}(d)(a - bd) - \frac{1}{2}cd^2 > \lambda_1 \\ \text{sgn}(d)\left(b + \sqrt{b^2 + 2c(\text{sgn}(d)a + \lambda_1)}\right)/c & \text{if} \quad \text{sgn}(d)(a - bd) - \frac{1}{2}cd^2 < -\lambda_1 \\ -d & \text{otherwise} \end{cases}.$$

The first and second equations above can be further unified into just one equation $\text{sgn}(d)\left(-b + \sqrt{b^2 - 2c(\text{sgn}(d)a + \lambda_1)}\right)/c$ if $\text{sgn}(d)a + \lambda_1 \leq 0$, so we finally have

$$\Delta\hat{x} = \begin{cases} \text{sgn}(d)\left(-b + \sqrt{b^2 - 2c(\text{sgn}(d)a + \lambda_1)}\right)/c & \text{if} \quad \text{sgn}(d)a + \lambda_1 \leq 0 \\ \text{sgn}(d)\left(b + \sqrt{b^2 + 2c(\text{sgn}(d)a - \lambda_1)}\right)/c & \text{if} \quad \text{sgn}(d)(a - bd) - \frac{1}{2}cd^2 > \lambda_1 \\ \text{sgn}(d)\left(b + \sqrt{b^2 + 2c(\text{sgn}(d)a + \lambda_1)}\right)/c & \text{if} \quad \text{sgn}(d)(a - bd) - \frac{1}{2}cd^2 < -\lambda_1 \\ -d & \text{otherwise} \end{cases}.$$

$\square$

# B  Related Work

**Optimization for CPH**   One way to train the CPH model is through gradient descent [63]. However, because the CPH loss is complex, it is difficult to pick the right step sizes, so gradient descent tends to be slow when we want to obtain solutions with high precision. To speed up the training process, people have used the Newton method [25, 22, 23]. The drawback of this approach is that it is computationally intensive to calculate the full Hessian matrix. Moreover, the vanilla Newton method cannot be applied to solve the $\ell_1$-regularized problem. To alleviate this problem, quasi Newton [62] and proximal Newton [51] methods have been proposed. However, as we have shown in our experiments, these Newton methods have the flaw of training loss blow up due to vanishing second order derivatives. A generic binary search method [41] has also been proposed, but the algorithm has been shown to be slower than the quasi Newton method. In contrast to all these approaches, our method is computationally efficient, can easily handle different regularizers, and guarantees that the loss decreases monotonically.

**Modern First and Second-order Optimization Methods**   Our work is greatly inspired by the modern developments for convex optimization [52], but the general principle cannot be rigidly applied. For first order methods, as we mentioned about gradient descent in the last paragraph, it is difficult to choose the right stepsize for fast convergence. Instead of performing gradient descent, our method performs coordinate descent, which has been shown to be effective in training other statistical models [9, 18, 19, 20, 34, 55]. We give an explicit formula, by leveraging the Popoviciu's inequality on variances [58], to calculate the Lipschitz constant at each coordinate. We also design a second order method (still under the coordinate descent framework) based on the cubic-regularization of the Newton method [53]. To calculate the Lipschizt constant, we make the connections to the third central moment and Bhatia–Davis's inequality [61]. Moreover, we are able to exploit the mathematical structures of the CPH model to compute the second order partial derivatives at the computational complexity of $O(n)$, making the evaluation per iteration of our second order method as fast as that of our first order method.

**Variable Selection and Interpretability for the CPH Model**   If we can find sparse solutions whose predictive performances are as good as dense solutions, we can better interpret which features play important roles. A popular way to select important variables for the CPH model is adding an $\ell_1$ regularization term [64], commonly known as the LASSO method. We can also use the ElasticNet method, which adds an $\ell_1 + \ell_2$ regularization term [62]. Another way is to apply the Adaptive LASSO [69], which repeatedly use the absolute values of coefficients obtained from the previous iteration as weights of $\ell_1$ regularizations for parameters in the current iteration. These above approaches all use convex regularizers and have difficulty obtaining high-quality solutions when the support size is small. The reason is that these convex regularizers penalize the magnitude of the coefficients while promoting sparsity. To avoid this issue, recently, solving the $\ell_0$-constrained problem [49, 29, 4] has attracted lots of attention and shown to produce much sparser models without losing accuracy. For the $\ell_0$-constrained CPH problem, ABESS [71] has proposed to use a hybrid method of greedy selection and feature swapping to solve the problem heuristically. However, as we have shown in our experiments, this method cannot handle highly correlated features. Our method also solves the $\ell_0$-constrained CPH problem but uses the beam search framework [66, 48, 47]. Although the beam search framework has already existed, this frameowork cannot be applied to the CPH model without our coordinate descent algorithm. We need to use coordinate descent for support expansion as well as coefficient finetuning, in which other Newton-type methods all have issues with losses potentially blowing up.

**Other Model Classes for Survival Analysis**   In addition to the CPH model, there are some other model classes that can be applied to analyze time-to-event data. One model class is the survival tree models [70, 3, 35]. Survival trees have the advantage of capturing non-linear interactions between features. However, when sparse models are desired, the accuracy of sparse trees is compromised by the fact that all samples in the same leaf node share the same predictions. One way to overcome this issue is to construct ensembles of trees using random forest or boosting techniques [36, 32, 33]. Another model class for survival analysis is based on neural networks [39, 7, 6, 59, 21]. However, for all these other model classes mentioned, they are not very interpretable due to large parameter space. The CPH model, which is the focus of our work, provides both interpretability and good accuracy. For applications involving high stakes decisions, it is desirable to produce models that are as sparse

as possible without losing accuracy. In this work and especially the variable selection experiments, we push the limit of sparsity-accuracy tradeoff curve for this model class.

## C Experimental Setup Details

### C.1 Computing Platforms

All experiments were run on the Intel(R) Xeon(R) CPU E5-2680 v3 Processor, 2.50GHz. We set the memory limit to be 100GB.

### C.2 Datasets, Baselines, and Licenses

We have a summary of datasets for experiments in Table 1.

| Dataset | Samples | Origin Features | Encoded Binary Features |
|---|---|---|---|
| Flchain | 7874 | 39 | 333 |
| Kickstarter1 | 4175 | 54 | 2144 |
| Dialysis | 6805 | 7 | 207 |
| EmployeeAttrition | 14999 | 17 | 272 |
| SyntheticHighCorrHighDim1 | 1200 | 1200 | N/A |
| SyntheticHighCorrHighDim2 | 900 | 900 | N/A |
| SyntheticHighCorrHighDim3 | 600 | 600 | N/A |

Table 1: Datasets Summary.

**Synthetic Data Generation Process** The synthetic data used in the paper is generated according to the following process, similar to [71]:

Firstly, from a Gaussian distribution $\mathcal{N}(\mathbf{0}, \Sigma)$ where the first entry is the mean and the second entry is the covariance matrix with size $p \times p$, we sample features:

$$\boldsymbol{x}_i \sim \mathcal{N}(\mathbf{0}, \Sigma). \tag{28}$$

The covariance matrix is defined as $\Sigma_{jl} = \rho^{|j-l|}$, where $\rho \in (0, 1]$ is the correlation parameter. When $\rho$ is large, the features in $\boldsymbol{x}_i$ are more correlated. We create a k-sparse coefficient vector $\boldsymbol{\beta}^* \in \mathbb{R}^p$. The entries of $\boldsymbol{\beta}^*$ are either 1 or 0. If $j \bmod (p/k) = 0$, then $\beta_j^* = 1$; otherwise, $\beta_j^* = 0$.

Secondly, we generate the death time $t_i$ according to the following equation:

$$t_i = \left( -\frac{\log V_i}{e^{\boldsymbol{x}_i^T \boldsymbol{\beta}^*}} \right)^s, \tag{29}$$

where $V_i \sim U(0, 1)$ (samples are drawn from a uniform distribution on the interval $[0, 1]$) and $s$ is a hyperparameter. In our experiments, we set $s = 0.1$.

Lastly, we generate the censoring time, the censoring indicator, and change the death time to observation time. We sample the censoring time from a uniform distribution: $C_i \sim U(0, 1)$. If the death time is bigger than the censoring time, we have the indicator equal to 1; otherwise, we have the indicator equal to 0. Specifically, we have:

$$\delta_i = \mathbb{1}_{t_i > C_i} \tag{30}$$

Afterwards, we change the death time to observation time, taking into consideration of censoring:

$$t_i = \min(t_i, C_i). \tag{31}$$

We form a triplet $(\boldsymbol{x}_i, t_i, \delta_i)$ and return this triplet as one sample.

**Real-world survival data:**

- **Flchain**: Use of nonclonal serum immunoglobulin free light chains to predict overall survival in the general population [10]. The event is death.

- **Kickstarter1**: Data from a popular crowdfunding platform, used to predict project success [46]. We used the version from `https://dmkd.cs.vt.edu/projects/survival/data/`.

- **Dialysis**: Data from a survival study of dialysis patients, which aims to assess quality of renal replacement therapy at dialysis centers in Rio de Janeiro, Brazil [60].

- **EmployeeAttrition**: The task of predicting when an IBM employee will quit. The event is an IBM employee's leaving [30].

**Licenses**    We list the licenses of the software packages used in this paper:

- **Abess**: The license of this package is GPL-3.

- **skglm**: The license of this package is BSD-3.

- **Scikit-survival (SkSurv)**: The license of this package is GPL-3.

- **Flchain**: We use the dataset from the Scikit-Survival [56] package. The GitHub link to this dataset is `https://github.com/sebp/scikit-survival/tree/master/sksurv/datasets/data`. The license of this package is GPL-3.

- **Kickstarter1**: We use the dataset from the Virginia Tech. The link to this dataset is `https://dmkd.cs.vt.edu/projects/survival/data/`. There is no license associated with this dataset. This means we cannot modify any part of the dataset, which we have obeyed while doing experiments on this dataset.

- **Dialysis**: We use the dataset from the SurvSet [12] package. The GitHub link to this dataset is `https://raw.githubusercontent.com/ErikinBC/SurvSet/main/SurvSet/_datagen/output/Dialysis.csv`. The license of this package is GPL-3.

- **EmployeeAttrition**: We use the dataset from the PySurvival [17] package. The GitHub link to this dataset is `https://github.com/square/pysurvival/blob/master/pysurvival/datasets/employee_attrition.csv`. The license of this package is Apache-2.

**Baselines**    We compared our method against various survival models:

- **Abess:** Adaptive Best-Subset Selection (ABESS) algorithm [71] for Cox proportional hazards model. We used the Cox model in abess python package Version 0.4.6.

- **SkglmALassoCox**: Cox model with the adaptive Lasso regularization [69]. We used the implementation from skglm [2, 51].

- **SksurvCoxnet**: Cox's proportional hazard's model with elastic net penalty [62]. We used the implementation from Scikit-survival (SkSurv): scikit-survival version-0.20.0 (`https://scikit-survival.readthedocs.io/en/stable/index.html`).

- **SksurvTree**: A greedy decision tree model using log-rank splitting rule [43]. We used the implementation from sksurv.

- **SksurvRSF**: Random survival forest [37] algorithm. We used the implementation from Sksurv.

- **SksurvGBST**: Gradient-boosted Cox proportional hazards loss with regression trees as base learner. In each stage, a regression tree is fit on the negative gradient of the loss function. We used the implementation from Sksurv.

- **SksurvNaiveSVM:** Naive version of linear Survival Support Vector Machine [65]. We used the implementation from Sksurv.

- **SksurvFastSVM**: Efficient Training of linear Survival Support Vector Machine [57]. We used the implementation from Sksurv.

**Evaluation Metrics**

1. **CIndex**: The full name of this metric score is Harrell's Concordance Indices [26]. It is used to evaluate the discrimination ability of a survival model. It assesses how well the model ranks observations based on their predicted risk of experiencing an event (e.g., death, disease recurrence) over time. The higher the CIndex score, the better the model.

2. **IBS**: The Integrated Brier score was proposed by [24] to evaluate survival models across all possible time threshold. The IBS score takes the Brier score a step further by integrating it across all possible time points within the follow-up period of interest. This provides a single score summarizing the model's performance over the entire time range. The lower the IBS score, the better the model.

3. **F1-score, Precision, Recall**: Suppose the true coefficients are $\boldsymbol{\beta}^*$ and the estimated coefficients are $\hat{\boldsymbol{\beta}}$. Then the precision score can be calculated as $P = |\text{supp}(\boldsymbol{\beta}^*) \cup \text{supp}(\hat{\boldsymbol{\beta}})|/|\text{supp}(\hat{\boldsymbol{\beta}})|$, where $\text{supp}(\cdot)$ extracts the support (indices whose coefficients are nonzero) of the input vector. The recall score can be calculated as $R = |\text{supp}(\boldsymbol{\beta}^*) \cup \text{supp}(\hat{\boldsymbol{\beta}})|/|\text{supp}(\boldsymbol{\beta}^*)|$. We calculate the F1 score as $\text{F1} = 2PR/(P + R)$.

### C.3 Details about Variable Selection Experiments

**Collection and Setup:** We ran 5-fold cross-validation (random seed 0) on the following datasets: Dialysis, Flchain, Kickstarter1, EmployeeAttrition, SyntheticHighCorrHighDim1, SyntheticHighCorrHighDim2, SyntheticHighCorrHighDim3. In order to create highly correlated features, we encoded continuous features into binarized features, by considering 1000 quantiles for each continuous column. For each dataset, we ran algorithms with different configurations and evaluated fitted models with metrics described in Appendix C.2:

- **Abess:** We ran this algorithm with 30 different configurations: support size, $k$, ranging from 1 to 30, forcing the number of non-zero coefficients in the Cox model to be exact $k$. We set $primary\_model\_fit\_max\_iter$ to be 20, $approximate\_Newton$ to be False. All other parameters were set to the default.

- **SksurvCoxnet:** We ran this algorithm with 30 different configurations: support size, $k$, ranging from 1 to 30, forcing the number of non-zero coefficients in the Cox model to be exact $k$. We set $l1\_ratio$ to be 1.0, $alpha\_min\_ratio$ to be 0.01. All other parameters were set to the default.

- **SkglmALassoCox**: We ran this algorithm with 9 different L1 regularization penalty parameters (alpha): 0.01, 0.05, 0.1, 0.5, 1, 5, 10, 50, 100. All other parameters were set to the default.

- **SksurvTree**: We ran this algorithm with 8 different configurations: max depth limit, $d$, ranging from 2 to 9, and a corresponding maximum leaf limit $2^d$. The random state was set to 2024 and all other parameters were set to the default.

- **SksurvRSF, SksurvGBST**: We ran this algorithm with $8 \times 5$ configurations: max depth limit, $d$, ranging from 2 to 9, and 5 different total numbers of estimators (10, 50, 100, 500, 100). The random state was set to 2024 and all other parameters were set to the default.

- **SksurvNaiveSVM, SksurvFastSVM**: We ran this algorithm with 9 different $\ell_2$ regularization penalty parameters (alpha): 0.01, 0.05, 0.1, 0.5, 1, 5, 10, 50, 100. All other parameters were set to the default.

- **SksurvCoxPHBeamSearch (our method)**: We ran this algorithm with 30 different configurations: support size, $k$, ranging from 1 to 30, forcing the number of non-zero coefficients in the Cox model to be exact $k$.

**Recording experimental results:** For each method with specific configuration, we have a set of up to 5 fitted models on each dataset. Some metrics may be unavailable:

- Precision, recall, and f1-score are not available on real-world data as we do not know the true coeffcients.

- The losses on the training and testing folds of cox models are not applied to non-Cox models.

- SksurvNaiveSVM and SksurvFastSVM were not able to provide IBS and AUC.

- Some methods' training time exceeded our 3-hour time limit (We noticed that sksurvNaiveSVM often timed out).

For Cox and SVM models, we recorded the number of non-zero coefficients as the support size. For tree based models we recorded the number of nodes as the support size. We plotted the standard deviation of support size and various metric scores as corresponding error bars.

# D    Additional Results

## D.1    Optimization on $\ell_1$ and $\ell_1 + \ell_2$-regularized Problems

### D.1.1    Results on Flchain

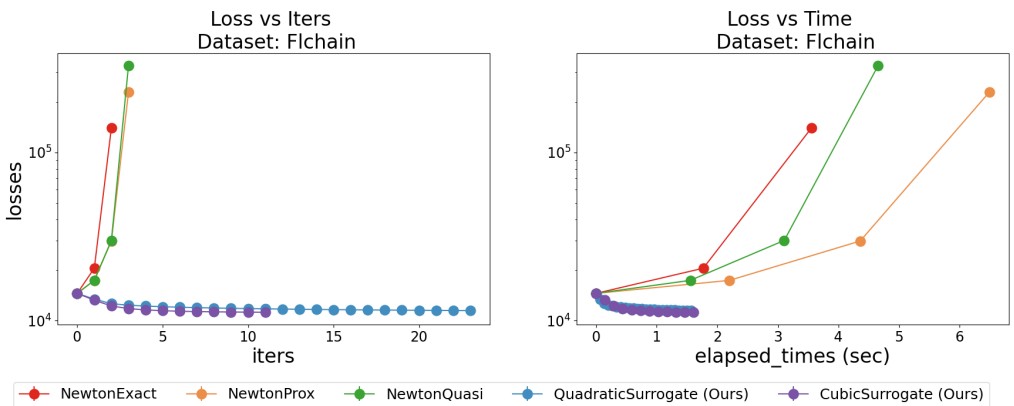

Figure 5: Optimization on the Flchain dataset with $\lambda_1 = 0$ and $\lambda_2 = 1.0$. The baselines (exact Newton, quasi Newton, and proximal Newton) all have the losses blow up. In contrast, our methods based on the quadratic and cubic surrogate functions have the losses monotonically decreasing.

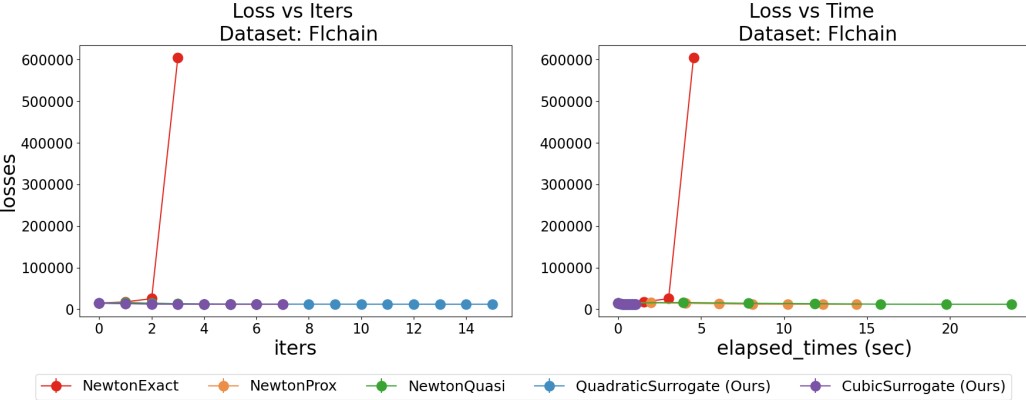

Figure 6: Optimization on the Flchain dataset with $\lambda_1 = 0$ and $\lambda_2 = 5.0$. The baseline, exact Newton, has the losses blow up despite a stronger $\ell_2$ regularization. The other two baselines, quasi Newton and proximal Newton, do not have this issue when $\ell_2$ increases but are significantly slower than our methods.

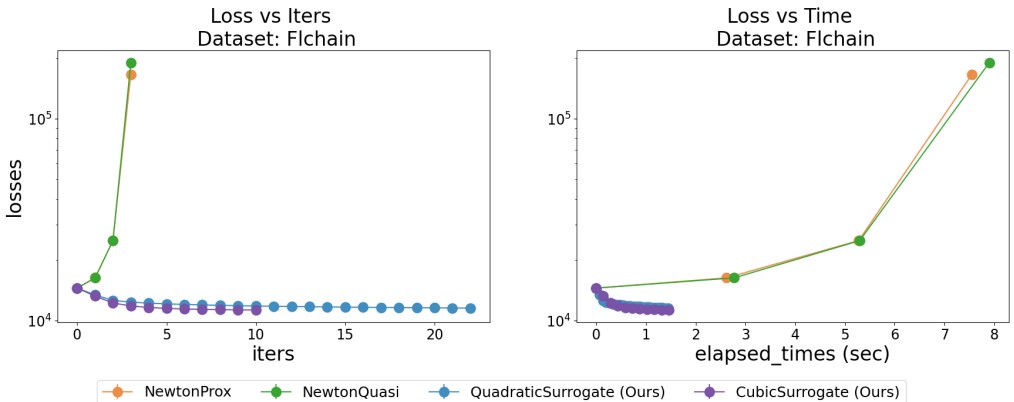

Figure 7: Optimization on the Flchain dataset with $\lambda_1 = 1.0$ and $\lambda_2 = 1.0$. The exact Newton method can be applied to solve the $\ell_1$-regularized problems, so we only compare with quasi Newton and proximal Newton. These two baselines both have the losses blow up when the $\ell_2$ regularization is weak. In contrast, our methods based on the quadratic and cubic surrogate functions have losses that monotonically decrease.

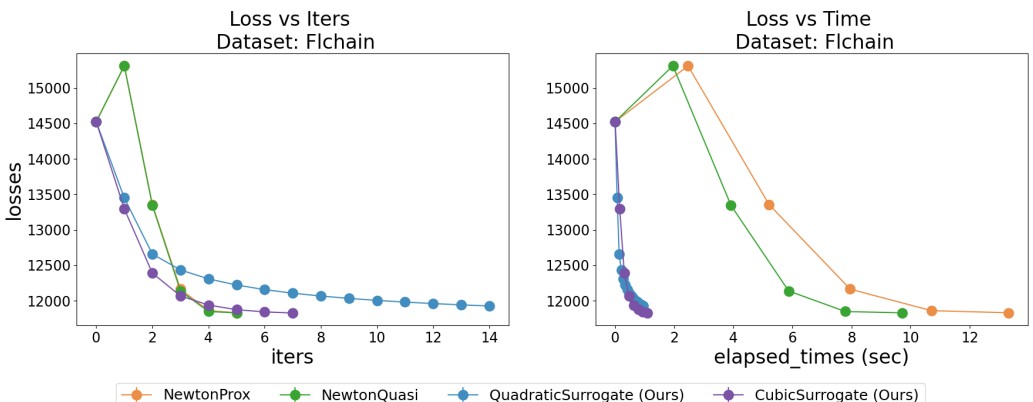

Figure 8: Optimization on the Flchain dataset with $\lambda_1 = 1.0$ and $\lambda_2 = 5.0$. The exact Newton method can be applied to solve the $\ell_1$-regularized problems, so we only compare with quasi Newton and proximal Newton. Stronger $\ell_2$ regularization helps these two baselines avoid the losses going into infinity. However, these two baselines are still significantly slower than our methods.

### D.1.2 Results on Employee Attrition

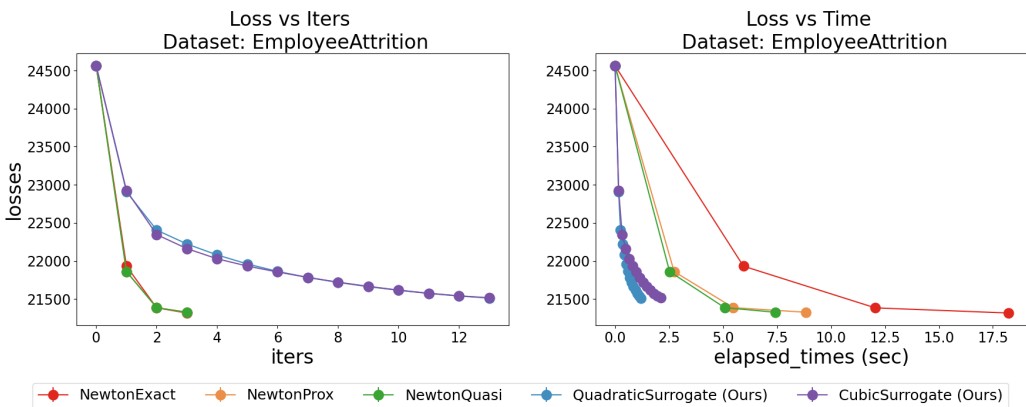

Figure 9: Optimization on the Employee Attrition dataset with $\lambda_1 = 0$ and $\lambda_2 = 1.0$. Although our methods make less progress toward the minimum loss per iteration (left plot), we are significantly faster than other methods in terms of elapsed time (wall clock) due to cheap evaluation cost per iteration. *For ease of figure reading, we only give a partial plot with a few iterations. When the number of iterations is large, our methods achieve better losses than the baseline methods.*

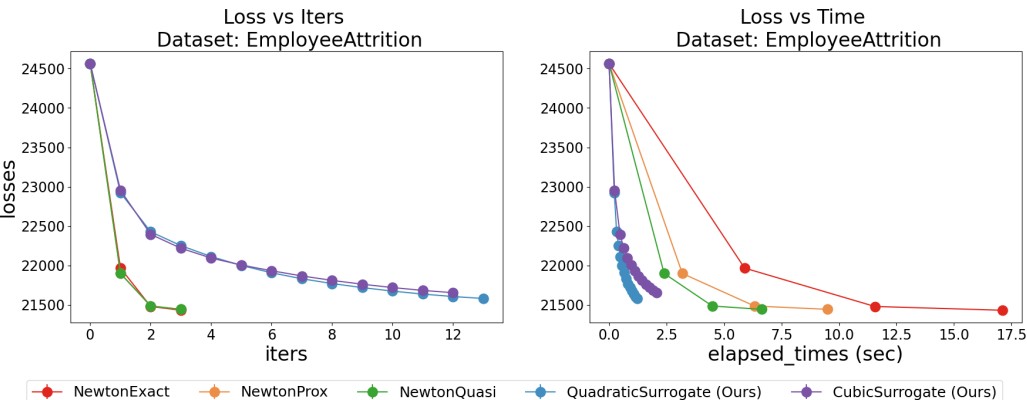

Figure 10: Optimization on the Employee Attrition dataset with $\lambda_1 = 0$ and $\lambda_2 = 5.0$. Although our methods make less progress toward the minimum loss per iteration (left plot), we are significantly faster than other methods in terms of elapsed time (wall clock) due to cheap evaluation cost per iteration. *For ease of figure reading, we only give a partial plot with a few iterations. When the number of iterations is large, our methods achieve better losses than the baseline methods.*

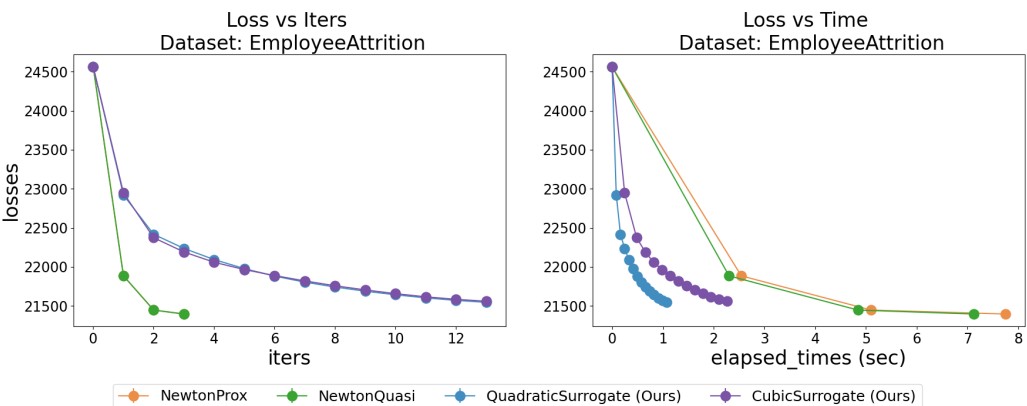

Figure 11: Optimization on the Employee Attrition dataset with $\lambda_1 = 1.0$ and $\lambda_2 = 1.0$. The exact Newton method can be applied to solve the $\ell_1$-regularized problems, so we only compare with quasi Newton and proximal Newton. Although our methods make less progress toward the minimum loss per iteration (left plot), we are significantly faster than other methods in terms of elapsed time (wall clock) due to cheap evaluation cost per iteration. *For ease of figure reading, we only give a partial plot with a few iterations. When the number of iterations is large, our methods achieve better losses than the baseline methods.*

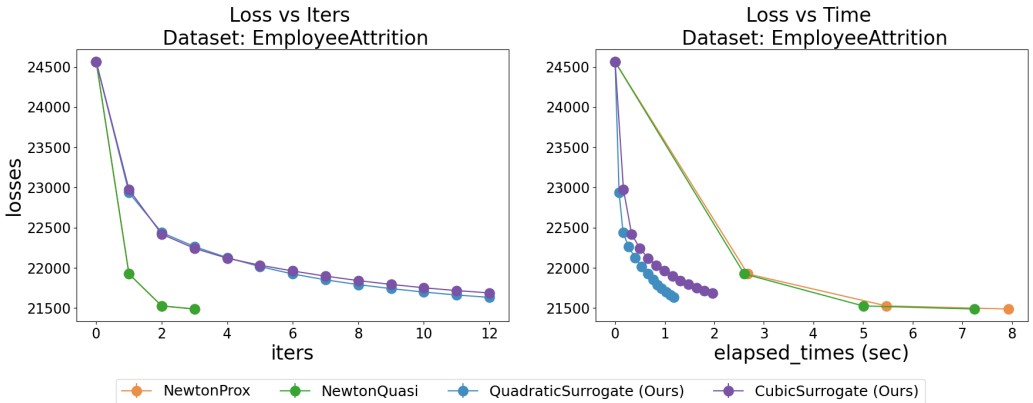

Figure 12: Optimization on the Employee Attrition dataset with $\lambda_1 = 1.0$ and $\lambda_2 = 5.0$. The exact Newton method can be applied to solve the $\ell_1$-regularized problems, so we only compare with quasi Newton and proximal Newton. Although our methods make less progress toward the minimum loss per iteration (left plot), we are significantly faster than other methods in terms of elapsed time (wall clock) due to cheap evaluation cost per iteration. *For ease of figure reading, we only give a partial plot with a few iterations. When the number of iterations is large, our methods achieve better losses than the baseline methods.*

### D.1.3 Results on Kickstarter1

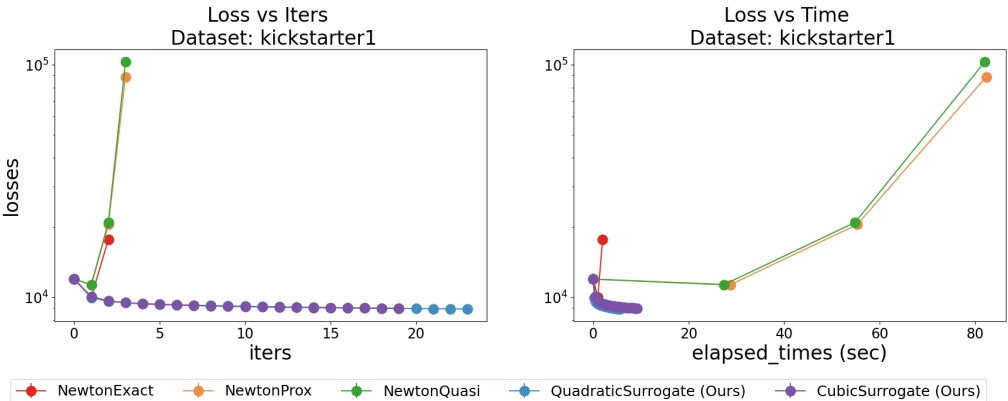

Figure 13: Optimization on the Kickstarter1 dataset with $\lambda_1 = 0$ and $\lambda_2 = 1.0$. The baselines (exact Newton, quasi Newton, and proximal Newton) all have losses that blow up. In contrast, our methods based on the quadratic and cubic surrogate functions have monotonically decreasing losses.

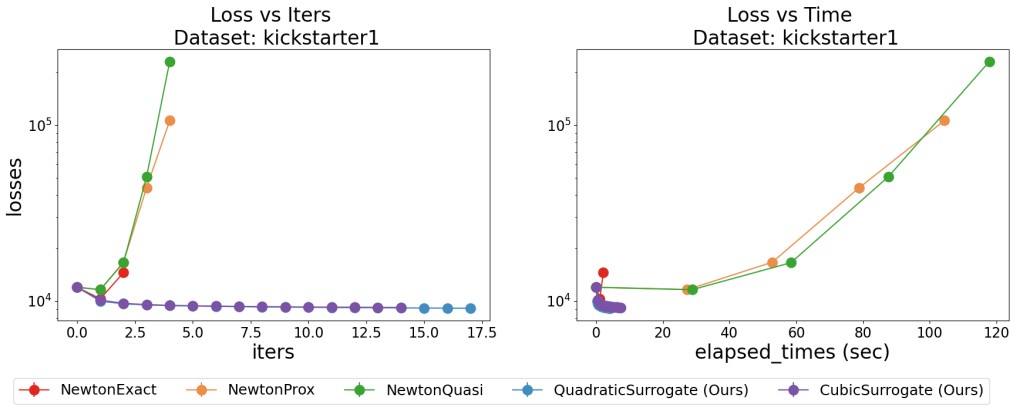

Figure 14: Optimization on the Kickstarter1 dataset with $\lambda_1 = 0$ and $\lambda_2 = 5.0$. The baselines (exact Newton, quasi Newton, and proximal Newton) all have the losses blow up. In contrast, our methods based on the quadratic and cubic surrogate functions have monotonically decreasing losses.

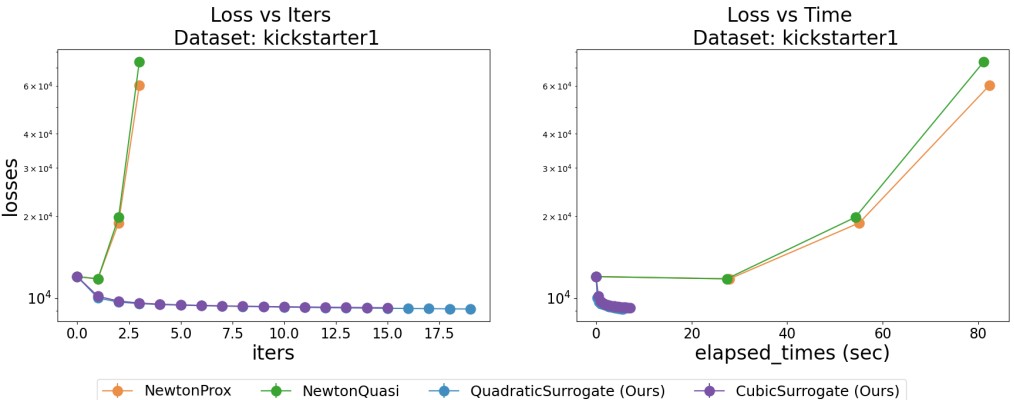

Figure 15: Optimization on the Kickstarter1 dataset with $\lambda_1 = 1.0$ and $\lambda_2 = 1.0$. The exact Newton method can be applied to solve the $\ell_1$-regularized problems, so we only compare with quasi Newton and proximal Newton. These two baselines both have the losses blow up when both the $\ell_1$ and $\ell_2$ regularizations are weak. In contrast, our methods based on the quadratic and cubic surrogate functions have monotonically decreasing losses.

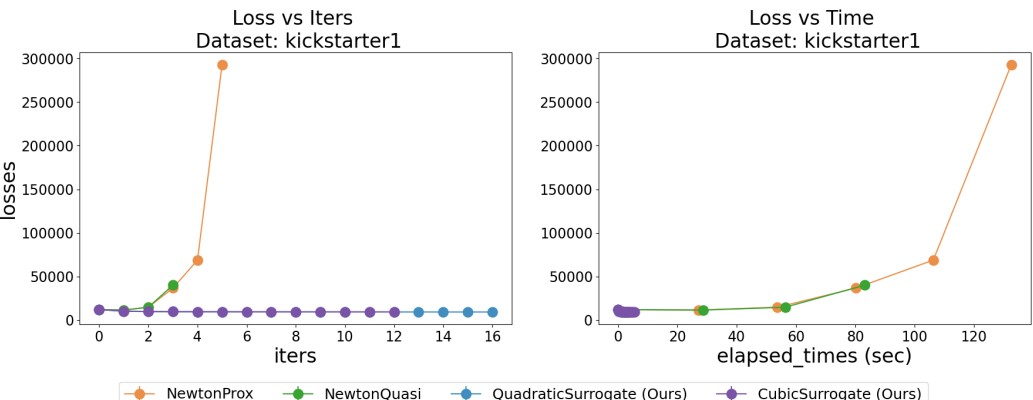

Figure 16: Optimization on the Kickstarter1 dataset with $\lambda_1 = 1.0$ and $\lambda_2 = 5.0$. The exact Newton method can be applied to solve the $\ell_1$-regularized problems, so we only compare with quasi Newton and proximal Newton. These two baselines both have the losses blow up even when we have a stronger $\ell_2$ regularization. In contrast, our methods based on the quadratic and cubic surrogate functions have monotonically decreasing losses.

### D.1.4 Results on Dialysis

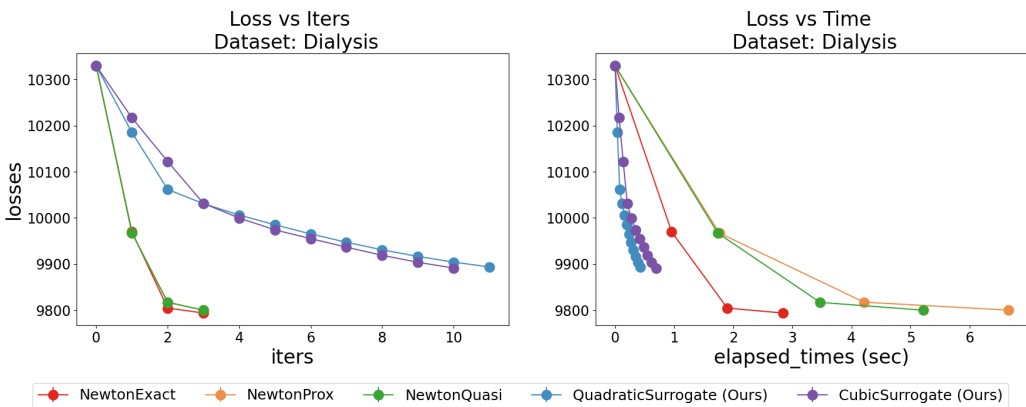

Figure 17: Optimization on the Dialysis dataset with $\lambda_1 = 0$ and $\lambda_2 = 1.0$. Although our methods make less progress toward the minimum loss per iteration (left plot), we are significantly faster than other methods in terms of elapsed time (wall clock) due to cheap evaluation cost per iteration. *For ease of figure reading, we only give a partial plot with a few iterations. When the number of iterations is large, our methods achieve better losses than the baseline methods.*

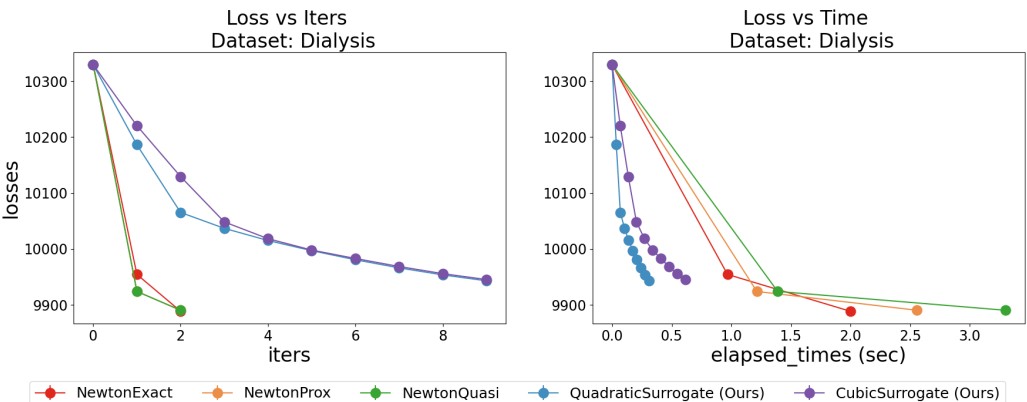

Figure 18: Optimization on the Dialysis dataset with $\lambda_1 = 0$ and $\lambda_2 = 5.0$. Although our methods make less progress toward the minimum loss per iteration (left plot), we are significantly faster than other methods in terms of elapsed time (wall clock) due to cheap evaluation cost per iteration. *For ease of figure reading, we only give a partial plot with a few iterations. When the number of iterations is large, our methods achieve better losses than the baseline methods.*

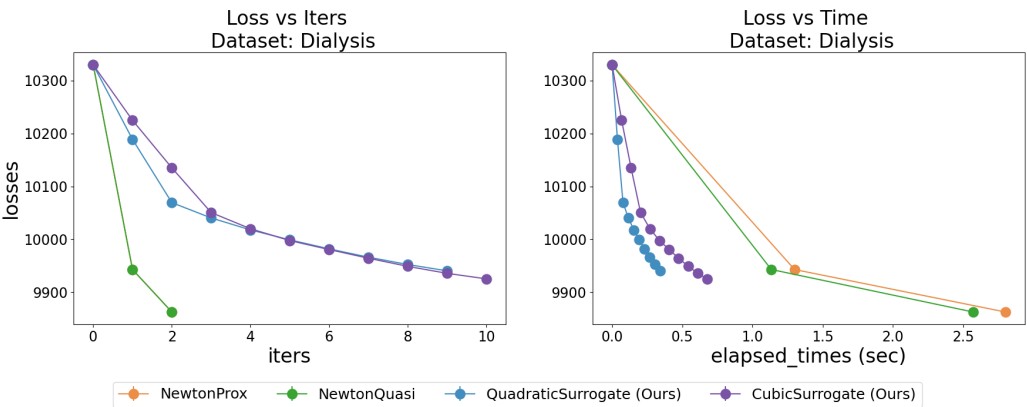

Figure 19: Optimization on the Dialysis dataset with $\lambda_1 = 1.0$ and $\lambda_2 = 1.0$. The exact Newton method can be applied to solve the $\ell_1$-regularized problems, so we only compare with quasi Newton and proximal Newton. Although our methods make less progress toward the minimum loss per iteration (left plot), we are significantly faster than other methods in terms of elapsed time (wall clock) due to cheap evaluation cost per iteration. *For ease of figure reading, we only give a partial plot with a few iterations. When the number of iterations is large, our methods achieve better losses than the baseline methods.*

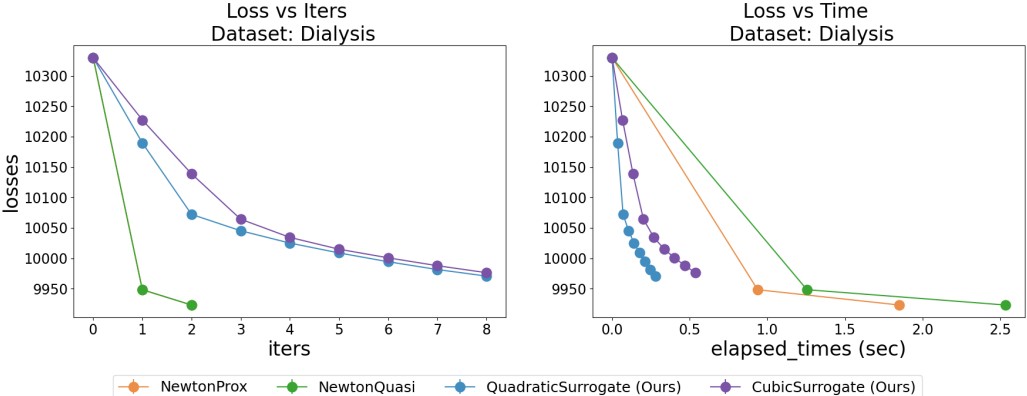

Figure 20: Optimization on the Dialysis dataset with $\lambda_1 = 1.0$ and $\lambda_2 = 5.0$. The exact Newton method can be applied to solve the $\ell_1$-regularized problems, so we only compare with quasi Newton and proximal Newton. Although our methods make less progress toward the minimum loss per iteration (left plot), we are significantly faster than other methods in terms of elapsed time (wall clock) due to cheap evaluation cost per iteration. *For ease of figure reading, we only give a partial plot with a few iterations. When the number of iterations is large, our methods achieve better losses than the baseline methods.*

## D.2 Variable Selection for the CPH Model

### D.2.1 Results on Dialysis

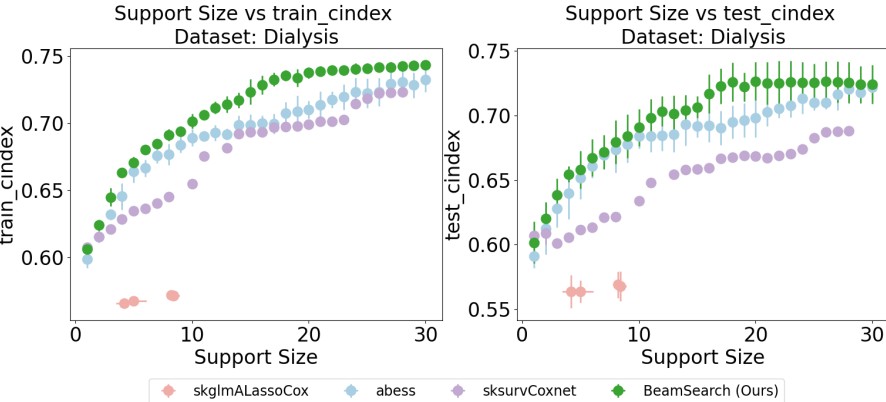

Figure 21: 5-fold Cross-validation on *Dialysis* dataset. Comparision with other cox models, metric: CIndex

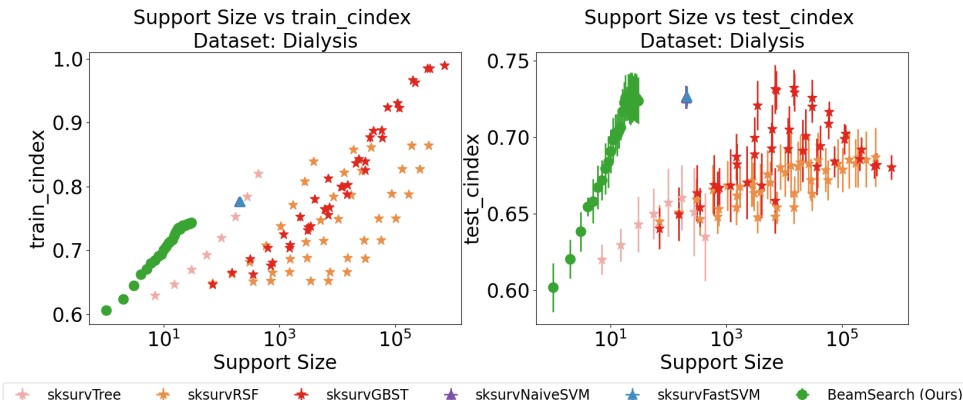

Figure 22: 5-fold Cross-validation on *Dialysis* dataset. Comparision with non-cox models, metric: CIndex

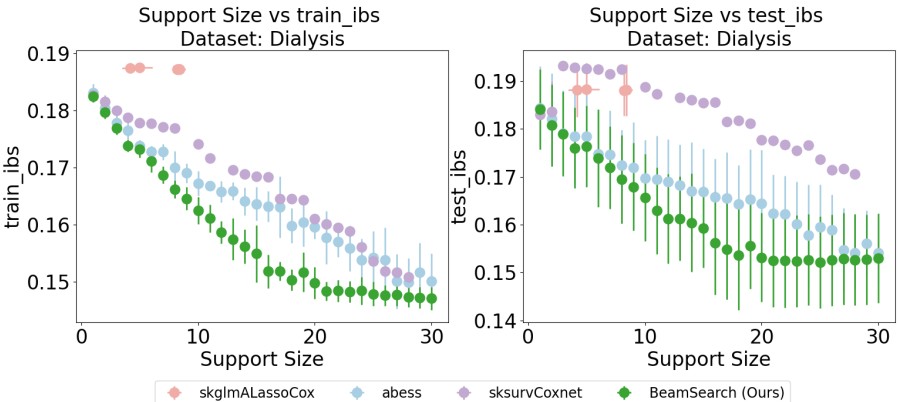

Figure 23: 5-fold Cross-validation on *Dialysis* dataset. Comparision with other cox models, metric: IBS

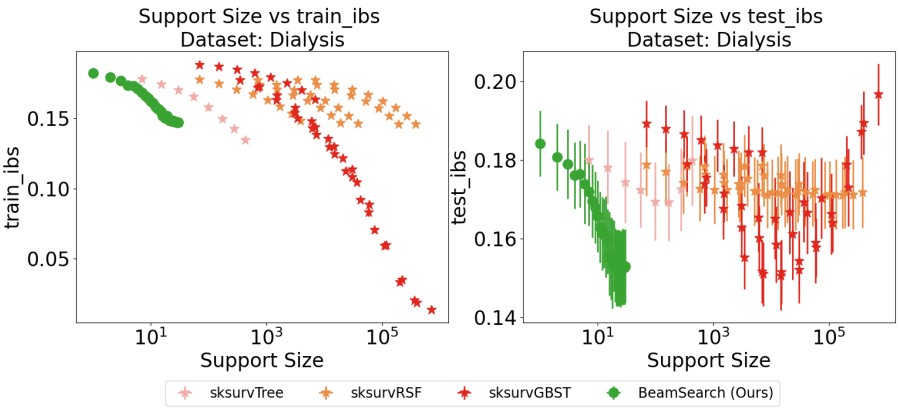

Figure 24: 5-fold Cross-validation on *Dialysis* dataset. Comparision with non-cox models, metric: IBS

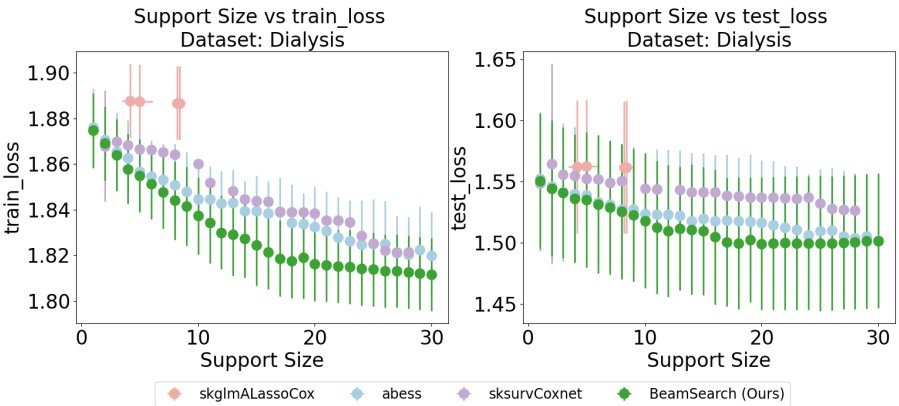

Figure 25: 5-fold Cross-validation on *Dialysis* dataset. Comparision with other cox models, metric: CPH Loss

### D.2.2 Results on EmployeeAttrition

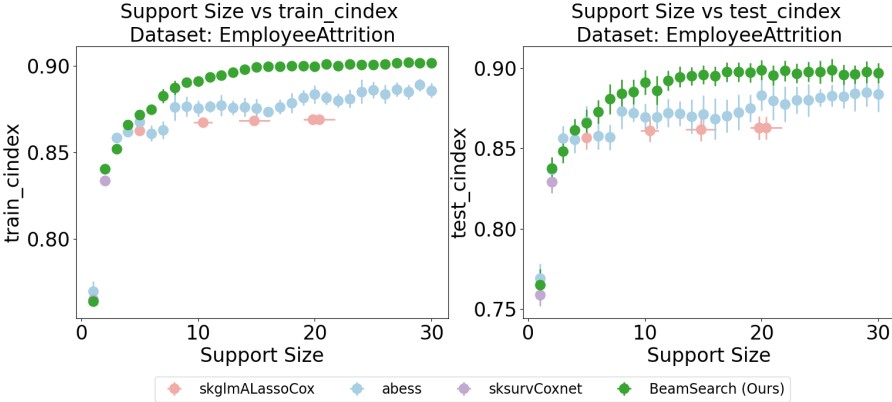

Figure 26: 5-fold Cross-validation on *EmployeeAttrition* dataset. Comparision with other cox models, metric: CIndex

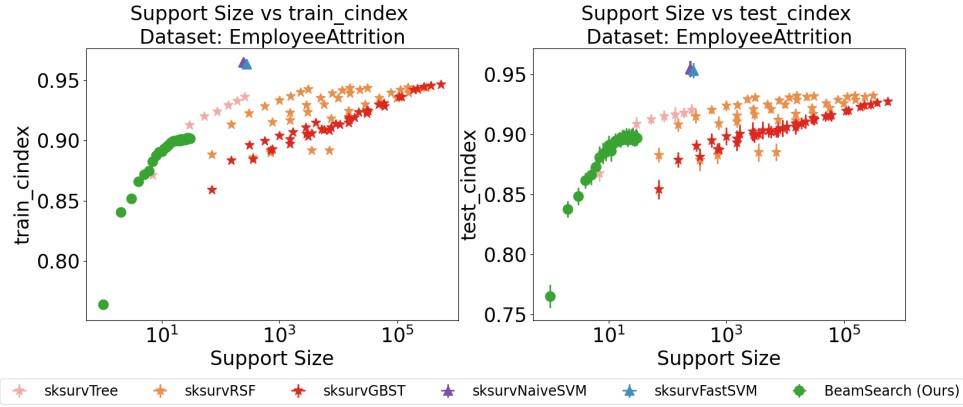

Figure 27: 5-fold Cross-validation on *EmployeeAttrition* dataset. Comparision with non-cox models, metric: CIndex

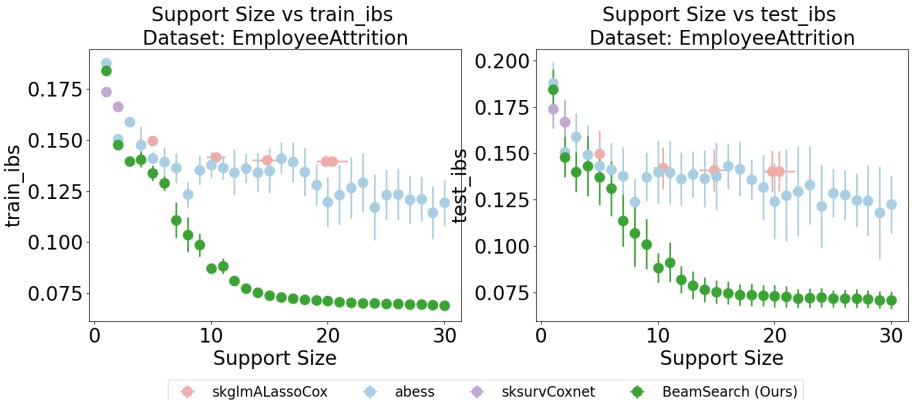

Figure 28: 5-fold Cross-validation on *EmployeeAttrition* dataset. Comparision with other cox models, metric: IBS

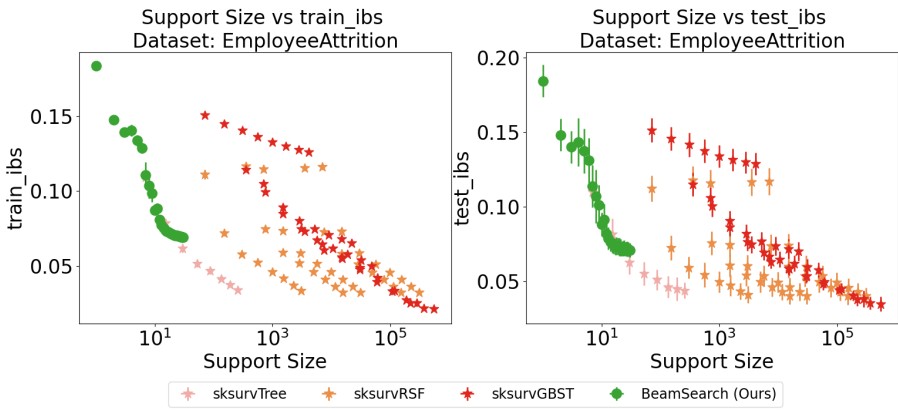

Figure 29: 5-fold Cross-validation on *EmployeeAttrition* dataset. Comparision with non-cox models, metric: IBS

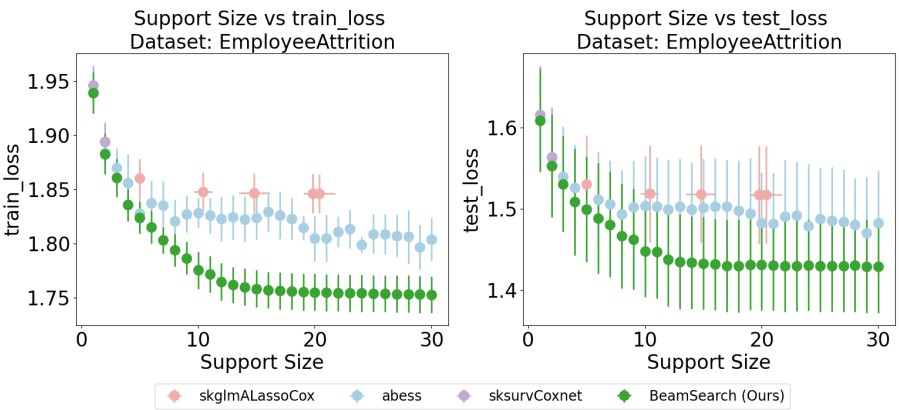

Figure 30: 5-fold Cross-validation on *EmployeeAttrition* dataset. Comparision with other cox models, metric: CPH Loss

### D.2.3 Results on Kickstarter1

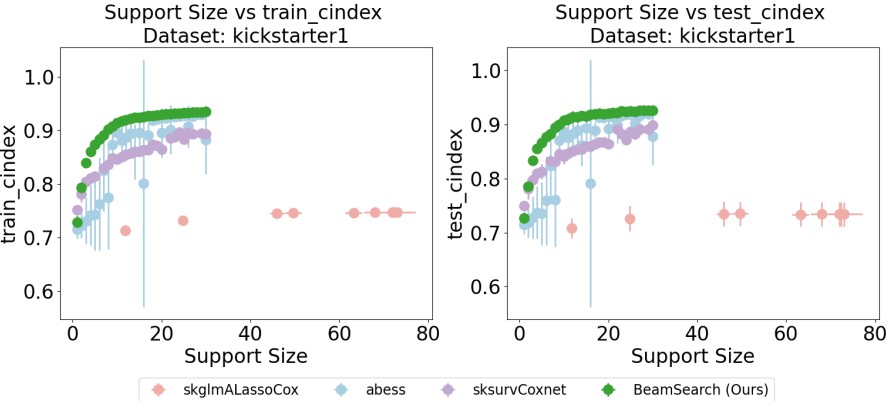

Figure 31: 5-fold Cross-validation on *kickstarter1* dataset. Comparision with other cox models, metric: CIndex

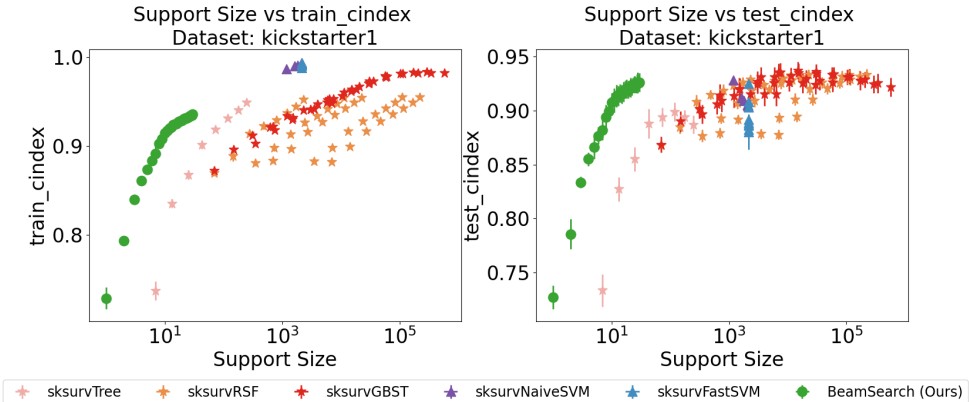

Figure 32: 5-fold Cross-validation on *kickstarter1* dataset. Comparision with non-cox models, metric: CIndex

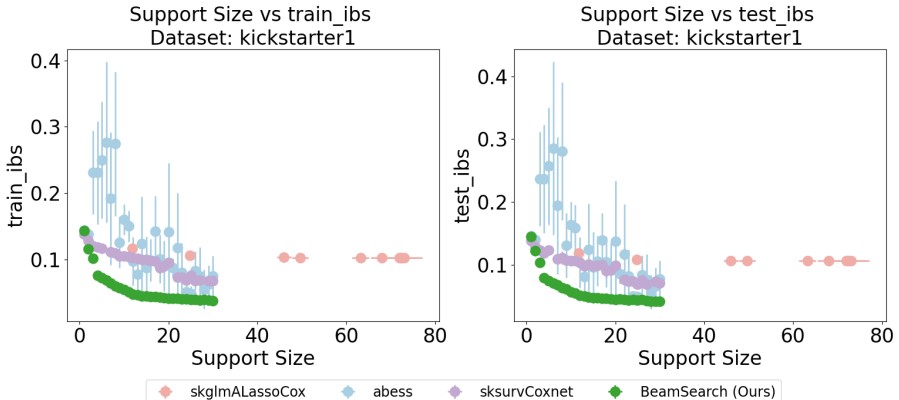

Figure 33: 5-fold Cross-validation on *kickstarter1* dataset. Comparision with other cox models, metric: IBS

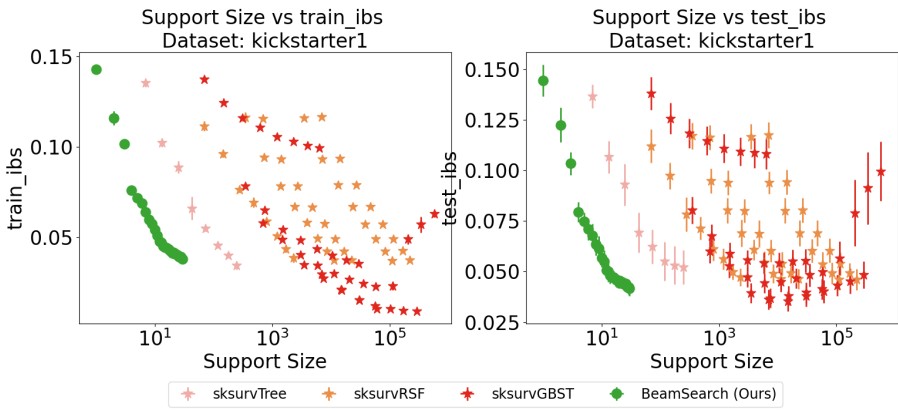

Figure 34: 5-fold Cross-validation on *kickstarter1* dataset. Comparision with non-cox models, metric: IBS

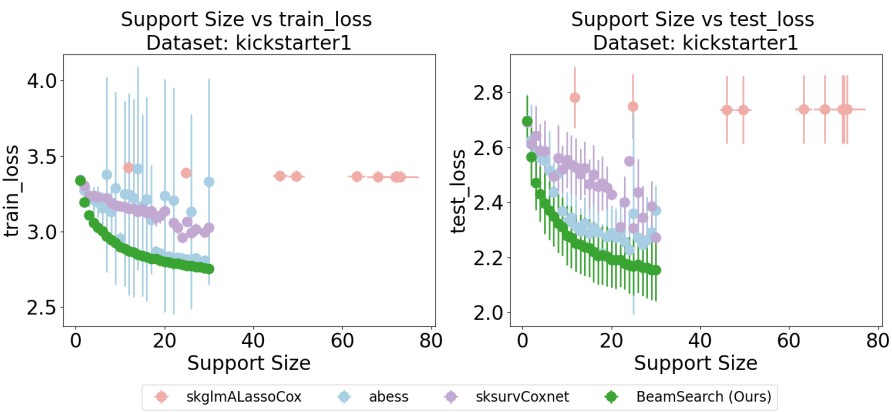

Figure 35: 5-fold Cross-validation on *kickstarter1* dataset. Comparision with other cox models, metric: CPH Loss

