# OpenReview forum: "FastSurvival: Hidden Computational Blessings in Training Cox Proportional Hazards Models"
_NeurIPS.cc/2024/Conference — NeurIPS 2024 poster_

### Official Review · Reviewer_qYft · 2024-06-17

**Soundness:** 2
**Presentation:** 2
**Contribution:** 2
**Rating:** 4
**Confidence:** 4

**Summary:**

The authors propose an alternative optimization method for the Cox proportional hazards model. They derive quadratic and cubic upper bounds on the loss and minimize these upper bounds with respect to a single model parameter at a time (similar to coordinate descent) via explicit formulas. They test their method on standard survival analysis benchmarks and also apply it to a feature selection problem with highly correlated features.

**Strengths:**

The topic addressed by the paper (survival analysis with large datasets/high dimensionality) is relevant to the community. The prose of the paper was generally easy to follow. The background information provided on survival analysis should also make it easier for a non-expert to understand the paper.

**Weaknesses:**

I do not believe the paper makes a significant contribution to the literature.

The runtime reduction results for computing derivatives of the partial likelihood are trivial. The example given on lines 147-148, showing a "surprising" improvement from $O(n^2)$ to $O(n)$ computation time, seems to use a deliberately inefficient method for computing derivatives is a point of comparison: there is no need to compute the full Hessian (which is where the authors derive the $O(n^2)$ runtime) when we only want some diagonal terms, and indeed one would expect computing derivatives of a function of the form $\sum_{i=1}^n f_i(x)$ to take $O(n)$ time provided the derivatives of each function in the summation can be computed in constant time. To that end, the only optimization over a naive implementation is the use of partial sums to avoid repeated calculation of sums with $O(n)$ terms, which is a very standard technique.

The most important problem is that the experimental results on runtime improvements are misleading. The Flchain dataset is a standard benchmark in survival analysis. It contains 7874 datapoints with 9 features, 4 of which are numeric. After standard 1-hot encoding of the categorical features, the features become 38-dimensional. Requiring 10-20 seconds for convergence (Fig. 1, rightmost plot) on such a modestly-sized dataset does not seem reasonable.

I tested this myself using the scikit-survival (sksurv) library. Sksurv implements Newton's method for fitting the Cox model with an L2 penalty, as can be seen from their source code: https://github.com/sebp/scikit-survival/blob/v0.22.2/sksurv/linear_model/coxph.py#L189

I ran the following code based on the sksurv documentation:

```
from sksurv.datasets import load_flchain
from sksurv.linear_model import CoxPHSurvivalAnalysis
from sksurv.preprocessing import OneHotEncoder
from sklearn.impute import SimpleImputer

X, y = load_flchain() # Load the data
Xt = OneHotEncoder().fit_transform(X) # One-hot encode categorical variables
imputer = SimpleImputer().fit(Xt) # Dataset contains missing values, impute with mean
Xt = imputer.transform(Xt)

cph = CoxPHSurvivalAnalysis(alpha=1.0, verbose=1) # Fit the model with regularization lambda_2 = 1 as in Fig. 1
cph.fit(Xt, y)
```

The fitting procedure converged for me on my laptop (Apple M2 Pro chip) in a Jupyter notebook in 9 iterations and 0.6s wall-clock time. I also had no problems getting the model fit to converge even with no regularization, contrary to the claim that "the losses blow up when regularization is weak" for Newton's method (caption of Fig. 1a). Thus, the apparent gains in performance are most likely due to poor implementations of Newton's method/other baselines.

In addition to these major issues, there are several other problems:
1. In equation (6), $\ell$ had been defined as a function of a $p$-dimensional vector $\beta$, so it is not clear what is meant by making it a function of the $n$-dimensional vector $\eta$. The reader can figure out what the authors probably meant, but this should be fixed and defined precisely.

2. The probabilistic interpretations given by the authors for some of the derivative expressions have appeared in very early works on the Cox model, see e.g. Section 2 of [1].

3. Equation (17) is exactly a Newton step, just in a single coordinate and with an upper bound on the second derivative instead of the second derivative itself. It's then not at all clear why we should expect this to be an improvement over Newton's method, especially for the specific cases the authors mention (e.g. high correlations among the features).

4. I'm not sure what is meant by "the analytical solution to this cubic surrogate function has not been well studied" (lines 189-190). The cubic formula, which can be used to solve a general cubic equation, has existed for several hundred years (https://en.wikipedia.org/wiki/Cubic_equation).

5. Lines 191-192: The claim that minimizing a convex upper bound of the original function will lead to a decrease in the original function as well is incorrect, barring some important additional assumptions. In particular, I don't believe the authors have justified their claim that the proposed method will "ensure monotonic loss decrease and global convergence".

**References:**

[1] Schoenfeld, David. Partial Residuals for the Proportional Hazards Regression Model. Biometrika (1982).

**Questions:**

Is there a mathematical justification for why we should expect the method to be robust to highly correlated features?

Please also address the concerns listed in the Weaknesses section.

**Limitations:**

Limitations are briefly discussed (lines 278-282), mostly regarding limitations of the Cox model itself.

---

> ### Author Rebuttal · Authors · 2024-08-07
>
> Thank you very much for reviewing our paper. Please see below for our answers to your questions.
>
> 1. **Runtime reduction results for computing derivatives are trivial**.
>
> Please look at Line 146.
> The result is surprising *especially for computing the exact 2nd order partial derivatives*.
>
> Even for the quasi and proximal Newton methods which make diagonal approximation of Hessian (Line 93-94), the complexity is still high.
> We **need to calculate the full Hessian** in the space of $\boldsymbol{\beta}$ via the formula $\boldsymbol{X}^T H(\boldsymbol{\eta})     \boldsymbol{X}$, whose complexity is $O(n p^2)$ when $H(\boldsymbol{\eta})$ is diagonal.
> Plus, the Newton method requires $O(p^3)$ to solve a linear system via the LDL method. In total, it is $O(np^2 + p^3)$.
>
> In contrast, we optimize each coordinate with complexity $O(n)$, and in total $O(np)$ for $p$ coordinates.
> This reduction is not at all trivial, and we support this claim through empirical results on real-world datasets.
>
> 2. **Experimental results on runtime are misleading**
>
> *Your preprocessing of the dataset is different from ours; from running on the wrong preprocessed datasets, one can't conclude our experimental results are wrong*.
> As written in Line 257-259, we perform binary thresholding to preprocess the data.
> You didn't do thresholding for continuous variables, which is what makes the variables correlated and challenging.
> In Appendix C2, our preprocessed Flchain dataset has 333 encoded binary features.
> Since you have only 38 features, you're using a totally different dataset.
>
> The sksurvCPH you used is also not the exact Newton method.
> They used a trick on Line 463 of the file coxph.py on the GitHub repo.
> They take half the exact Newton step size if the loss goes up.
> This trick does not work in general.
> When $\lambda_2 = 0.001$, sksurvCPH runs over 100 seconds without converging.
> The loss is 14522.97.
> In contrast, our method runs under 40 seconds, and the loss is 10659.91 (high precision solution).
>
> We uploaded to Anonymous GitHub the complete code (existing Newton methods and preprocessing steps) to support our claims.
> Due to the rebuttal policy, we only shared the link with the Area Chair, who can verify results in Section 4.1 and this response by running our codes.
>
> 3. **Eq. (6), how do we get from $\boldsymbol{\beta}$ to $\boldsymbol{\eta}?$**
>
> As mentioned in Line 89, we use an intermediate variable $\boldsymbol{\eta}$ with $\boldsymbol{\eta = X \beta}$.
> By the chain rule, we have $\nabla_{\boldsymbol{\beta}} \ell(\boldsymbol{\beta}) = \boldsymbol{X}^T \nabla_{\boldsymbol{\eta}} \ell(\boldsymbol{\eta})$ and $\nabla_{\boldsymbol{\beta}}^2 \ell(\boldsymbol{\beta}) = \boldsymbol{X}^T \nabla_{\boldsymbol{\eta}}^2 \ell(\boldsymbol{\eta}) \boldsymbol{X}$.
>
> 4. **Probabilistic interpretations are not novel**
>
> Thank you for pointing out this paper; it does not overlap with the novel contributions in this submission.
> We will cite this reference during revision.
> The reference you cited has only basics.
> The equation at the bottom of page 1 boils down to the optimality condition when the gradient is equal to 0.
> **Our contributions are substantially different**. We show that
> - The 2nd- and 3rd-order partial derivatives have probabilistic interpretations, which is unexpected.
> - The 2nd- and 3rd-order partial derivatives are in the same formulas as the 2nd- and 3rd-moment calculations.
> - The Lipschitz constants can be computed explicitly for the 1st- and 2nd-order partial derivatives.
>
> 5. **Why is Eq.17 an improvement over and different from the existing Newton methods?**
>
> See the Abstract (Line 7-9), Introduction (Line 33-39), and Preliminaries (Line 95-104). To reiterate:
>
> **Problems with existing Newton methods**
>
> a) They all have trouble converging due to the vanishing Hessian.
> This is a well-known issue for Newton-based methods (see left two plots in Fig 1).
>
> b) Even if they converge, none of them can converge with high precision fast enough (Line 100-104; see right two plots in Fig 1).
>
> **Difference over previous Newton methods**
>
> a) We always converge to optimal solutions with high precision. For highly correlated features, other methods could have the loss blow up.
>
> b) Without our formula to calculate $L_2$ in Eq.13, it is impossible to implement Eq.17.
>
> c) Without our formula to calculate $L_3$ in Eq.14, it is impossible to implement for Eq.18.
>
> 6. **The cubic equation and formula have existed for several hundred years**
>
> The cubic equation has almost nothing to do with our cubic surrogate function, despite sharing the same word ``cubic''.
> The tasks are completely different.
> In our work, we *minimize* a special polynomial whose highest order is 3.
> One core step is to *find roots to the second-order polynomial* (see Appendix A4 for details).
> In contrast, the cubic equation and cubic formula are used to *find the roots to the third-order polynomial*.
> We never used the cubic formula in our work.
> We also solve $\ell_1$-regularized cubic surrogate problem, which has not been studied before.
>
> 7. **Why is the method robust to highly correlated features in variable selection, and is there any math theory?**
>
> At the core, our methods work because they allow solutions to converge with high precision quickly.
> During beam search, having high-precision solutions allows us to compare solutions and pass only the best ones to the next stage of support expansion.
> If one doesn't get high-precision solutions, some medium- or low-quality solutions can be passed to the next stage, which hurts the overall solution quality.
> For theories, there are two relevant papers ([13] and [47] in our reference).
> We can combine their proofs to show our beam search on the Cox model is an approximation algorithm, which means there is some quality guarantee on the final solution.
>
> Hopefully we have answered all your questions and cleared all your doubts!

---

> > ### Comment · Reviewer_qYft · 2024-08-10
> >
> > Thanks to the authors for their detailed response. My questions related to points 3 & 4, as well as the monotonic decrease in the objective have all been answered, and I have raised my score accordingly. I still lean towards rejection for the following reasons:
> >
> > 1. I understand that naively computing the second order derivatives naively using the formula on line 146 would lead to a high runtime. However, using equation (8) and standard methods for reducing runtime when computing multiple partial sums, the runtime reduction is not surprising.
> >
> > 2. I now understand the reason for the difference in wall clock runtime from my implementation vs. yours, and I believe that the baselines were implemented correctly. However, I am not convinced that the preprocessing used in the paper is realistic. Turning continuous features into highly correlated categorical features is the opposite of what one would usually do in practice. Is there any benefit to this procedure? For instance, does it improve the downstream performance of the model, e.g. in terms of the C-index on the test data? Barring some good justification for this (other than that it leads to better performance for the proposed method over existing baselines), this experiment doesn't seem convincing.
> >
> > 6. I agree with the authors that the cubic equation does not apply to this case and I apologize for the misunderstanding. That said, minimizing a cubic by finding the roots of a quadratic equation still does not constitute significant mathematical novelty. Even with the addition of L1 regularization, this is reasonably behaved one-dimensional optimization problem which should not pose much challenge for standard approaches.

---

> > > ### Author Response · Authors · 2024-08-13
> > >
> > > 1. **Your claim that using Eq.8 and standard methods for reducing runtime when computing multiple partial sums, the runtime reduction is not surprising**
> > >
> > > Your opinion on runtime reduction not being surprising is based on the assumption that you already have access to Eq.8.
> > > However, before our submission, no one knows we could possibly write the 2nd-order partial derivatives in such a way.
> > > Eq.8 is unique to the Cox model (Eq.8) and does not hold for a generic loss function.
> > > **Since Eq.8 is not something already known/established, it's not possible to conclude our work ''is not surprising'' considering only what follows it**.
> > > As mentioned before, the previous approaches rely on $\boldsymbol{\eta}^T \nabla_{\boldsymbol{\eta}} \ell(\boldsymbol{\eta}) \boldsymbol{\eta}$.
> > > This is why we can reduce the complexity from $O(n^2)$ to $O(n)$ and why we say the runtime reduction is surprising.
> > >
> > > 2. **What is the benefit of our preprocessing?**
> > >
> > > We do not perform this for the sake of creating a challenging dataset.
> > > It does have a potential benefit for producing better models.
> > > Due to the character limit, please see another post where we support this claim through experiments.
> > >
> > > The origin of our preprocessing can be traced back to additive models in statistics (see Line 27, 257-258, and 280, with citations [1, 11, 27, 28, 49, 67]).
> > > Additive models have been extensively used to capture nonlinear relationships between the target and continuous variables.
> > > For example, on the ICU patients, death risks may not increase linearly w.r.t age, a continuous variable; the risk for older patients may increase at a much higher rate.
> > > Using the original continuous variable would fail to capture this relationship.
> > >
> > > Lastly, additive models can be created by other preprocessing procedures, such as splines and polynomials (see [27, 28, 67]).
> > > However, no matter which procedure we use, the preprocessed features are likely much more correlated than the original ones.
> > > As shown in Section 4.2 (variable selection), our method can handle highly correlated features much better than existing methods.
> > >
> > > 3. **Minimizing a cubic by finding the roots of a quadratic equation still does not constitute significant mathematical novelty**
> > >
> > > We try to minimize the following cubic surrogate function:
> > > $$h_x(\Delta x):= f(x) + f'(x) \Delta x + \frac{1}{2} f''(x) \Delta x^2 + \frac{1}{6} L_3 \vert \Delta x \vert^3.$$
> > > Your opinion on our method not constituting significant mathematical novelty is based on the assumption that *$h_x(\Delta x)$ already exists*.
> > > Specifically, you assume that $f'(x)$, $f''(x)$, and $L_3$ have already been given to you.
> > > However, this is not the case, due to two things mentioned in our initial rebuttal:
> > >
> > > a) $f''(x)$can be computed in $O(n)$ time. Previously, the best way takes $O(n^2)$. *If you compute $f''(x)$ in the old-fashioned way, it takes much longer to formulate $h_x(\Delta x)$*.
> > >
> > > b) Previously, it is unknown, regarding the Cox model, whether $L_3$ (Lipschitz constant for $f''(x)$) exists and, if so, whether there exists an explicit way to compute $L_3$. In our work, we answer both questions positively, through Eq.9 and Eq.14. *Without our work, you cannot even compute $L_3$ and thus formulate $h_x(\Delta x)$, let alone minimize it*.
> > >
> > > Moreover, *obtaining Eq.9 and Eq.14 are nontrivial mathematical tasks*.
> > > No one has expected these nice structures (probabilistic formula for the 3rd-order partial derivatives and its connection to the 3rd-order central moment calculation) could pop up out of the Cox model.
> > >
> > > Lastly, there are subjective opinions and objective facts.
> > > Regarding subjective opinions, we accept that different researchers can have different research tastes on novelty.
> > > However, objectively, the fact is that no one has written Eq.18, the analytical minimizer to $h_x(\Delta x)$. If you think they have, please point us to the literature where they publish this equation.
> > >
> > > 4. **Even with the addition of L1 regularization, this is reasonably behaved one-dimensional optimization problem which should not pose much challenge for standard approaches**
> > >
> > > Our reasoning above also applies here.
> > > *Without our work, you cannot even formulate this L1-regularized cubic surrogate function, let alone minimize it.*
> > > Previous researchers didn't know that the cubic surrogate function could be useful to solving the L1-regularized problem.
> > > All they had are the quasi and proximal Newton methods.
> > > Again, we make a contribution here because Eq.22 has never been presented.
> > > Moreover, our Eq.22 is one step further beyond the traditional soft-thresholding operator (well known as solving the L1-regularized problem with the quadratic surrogate function).
> > > Our work opens the door to solving the L1-regularized problem with the cubic surrogate function.

---

> > > > ### Author Response · Authors · 2024-08-13
> > > > **Follow-up on what is the benefit of our preprocessing**
> > > >
> > > > Please see the results below for the comparison between Cox models trained on the original and preprocessed features, respectively.
> > > > The results correspond to our Figure 3 and Figure 4 in the main paper.
> > > > The trend is very clear: the models trained on the preprocessed features achieve better metrics than the ones trained on the original features (for CIndex, the higher the better; for IBS, the lower the better).
> > > > Note that the Employee Attrition dataset has 17 original features while the Dialysis dataset has only 7 original features.
> > > >
> > > > 1. **Results on Employee Attrition; see Figure 3**
> > > >
> > > > | Use Thresholding as Preprocessing | Support Size | Method | Train CIndex | Test CIndex | Train IBS | Test IBS |
> > > > | :---: | :----: | :---: | :---: | :---: | :---: | :---: |
> > > > | no | 5 | ABESS | 0.855 $\pm$ 0.002 | 0.855 $\pm$ 0.004 | 0.259 $\pm$ 0.003 | 0.260 $\pm$ 0.025 |
> > > > | yes | 5 | ABESS | 0.919 $\pm$ 0.006 | 0.918 $\pm$ 0.011 | 0.204 $\pm$ 0.010 | 0.206 $\pm$ 0.030 |
> > > > | no | 10 | ABESS | 0.856 $\pm$ 0.001 | 0.855 $\pm$ 0.007 | 0.253 $\pm$ 0.003 | 0.254 $\pm$ 0.021 |
> > > > | yes | 10 | ABESS | 0.927 $\pm$ 0.004 | 0.925 $\pm$ 0.013 | 0.199 $\pm$ 0.012 | 0.201 $\pm$ 0.032 |
> > > > | no | 15 | ABESS | 0.857 $\pm$ 0.002 | 0.856 $\pm$ 0.007 | 0.252 $\pm$ 0.002 | 0.252 $\pm$ 0.022 |
> > > > | yes | 15 | ABESS | 0.925 $\pm$ 0.007 | 0.924 $\pm$ 0.013 | 0.200 $\pm$ 0.018 | 0.201 $\pm$ 0.030 |
> > > > | no | 17 | ABESS | 0.856 $\pm$ 0.002 | 0.856 $\pm$ 0.006 | 0.252 $\pm$ 0.002 | 0.252 $\pm$ 0.022 |
> > > > | yes | 17 | ABESS | 0.924 $\pm$ 0.006 | 0.922 $\pm$ 0.011 | 0.209 $\pm$ 0.016 | 0.209 $\pm$ 0.023 |
> > > > | yes | 20 | ABESS | 0.931 $\pm$ 0.004 | 0.928 $\pm$ 0.012 | 0.189 $\pm$ 0.021 | 0.194 $\pm$ 0.039 |
> > > > | yes | 25 | ABESS | 0.935 $\pm$ 0.006 | 0.934 $\pm$ 0.007 | 0.182 $\pm$ 0.021 | 0.188 $\pm$ 0.022 |
> > > > | yes | 30 | ABESS | 0.939 $\pm$ 0.004 | 0.937 $\pm$ 0.010 | 0.172 $\pm$ 0.017 | 0.176 $\pm$ 0.034 |
> > > > | no | 5 | ours | 0.854 $\pm$ 0.002 | 0.854 $\pm$ 0.005 | 0.252 $\pm$ 0.002 | 0.252 $\pm$ 0.020 |
> > > > | yes | 5 | ours | 0.920 $\pm$ 0.001 | 0.919 $\pm$ 0.009 | 0.199 $\pm$ 0.006 | 0.201 $\pm$ 0.026 |
> > > > | no | 10 | ours | 0.854 $\pm$ 0.002 | 0.853 $\pm$ 0.007 | 0.246 $\pm$ 0.002 | 0.246 $\pm$ 0.020 |
> > > > | yes | 10 | ours | 0.943 $\pm$ 0.002 | 0.942 $\pm$ 0.007 | 0.127 $\pm$ 0.002 | 0.128 $\pm$ 0.013 |
> > > > | no | 15 | ours | 0.855 $\pm$ 0.002 | 0.854 $\pm$ 0.007 | 0.245 $\pm$ 0.002 | 0.245 $\pm$ 0.020 |
> > > > | yes | 15 | ours | 0.951 $\pm$ 0.002 | 0.950 $\pm$ 0.007 | 0.107 $\pm$ 0.003 | 0.109 $\pm$ 0.010 |
> > > > | no | 17 | ours | 0.855 $\pm$ 0.002 | 0.854 $\pm$ 0.007 | 0.245 $\pm$ 0.002 | 0.245 $\pm$ 0.020 |
> > > > | yes | 17 | ours | 0.951 $\pm$ 0.002 | 0.950 $\pm$ 0.007 | 0.105 $\pm$ 0.003 | 0.107 $\pm$ 0.010 |
> > > > | yes | 20 | ours | 0.952 $\pm$ 0.001 | 0.950 $\pm$ 0.008 | 0.103 $\pm$ 0.002 | 0.105 $\pm$ 0.009 |
> > > > | yes | 25 | ours | 0.953 $\pm$ 0.002 | 0.951 $\pm$ 0.007 | 0.101 $\pm$ 0.002 | 0.103 $\pm$ 0.007 |
> > > > | yes | 30 | ours | 0.954 $\pm$ 0.002 | 0.952 $\pm$ 0.007 | 0.099 $\pm$ 0.002 | 0.102 $\pm$ 0.007 |
> > > >
> > > > 2. **Results on Dialysis; see Figure 4**
> > > >
> > > > | Use Thresholding as Preprocessing | Support Size | Method | Train CIndex | Test CIndex | Train IBS | Test IBS |
> > > > | :---: | :----: | :---: | :---: | :---: | :---: | :---: |
> > > > | no | 5 | ABESS | 0.649 $\pm$ 0.002 | 0.642 $\pm$ 0.008 | 0.173 $\pm$ 0.001 | 0.176 $\pm$ 0.010 |
> > > > | yes | 5 | ABESS | 0.666 $\pm$ 0.011 | 0.650 $\pm$ 0.020 | 0.178 $\pm$ 0.002 | 0.182 $\pm$ 0.012 |
> > > > | no | 7 | ABESS | 0.651 $\pm$ 0.002 | 0.646 $\pm$ 0.008 | 0.173 $\pm$ 0.001 | 0.176 $\pm$ 0.010 |
> > > > | yes | 7 | ABESS | 0.680 $\pm$ 0.008 | 0.670 $\pm$ 0.011 | 0.175 $\pm$ 0.001 | 0.178 $\pm$ 0.010 |
> > > > | yes | 10 | ABESS | 0.692 $\pm$ 0.004 | 0.683 $\pm$ 0.012 | 0.170 $\pm$ 0.002 | 0.174 $\pm$ 0.009 |
> > > > | yes | 15 | ABESS | 0.701 $\pm$ 0.008 | 0.693 $\pm$ 0.018 | 0.166 $\pm$ 0.008 | 0.170 $\pm$ 0.013 |
> > > > | yes | 20 | ABESS | 0.709 $\pm$ 0.008 | 0.694 $\pm$ 0.015 | 0.164 $\pm$ 0.003 | 0.171 $\pm$ 0.011 |
> > > > | yes | 25 | ABESS | 0.724 $\pm$ 0.013 | 0.710 $\pm$ 0.007 | 0.156 $\pm$ 0.005 | 0.163 $\pm$ 0.011 |
> > > > | yes | 30 | ABESS | 0.731 $\pm$ 0.003 | 0.719 $\pm$ 0.010 | 0.153 $\pm$ 0.003 | 0.159 $\pm$ 0.010 |
> > > > | no | 5 | ours | 0.651 $\pm$ 0.003 | 0.644 $\pm$ 0.010 | 0.179 $\pm$ 0.003 | 0.181 $\pm$ 0.010 |
> > > > | yes | 5 | ours | 0.672 $\pm$ 0.005 | 0.654 $\pm$ 0.014 | 0.176 $\pm$ 0.001 | 0.181 $\pm$ 0.001 |
> > > > | no | 7 | ours | 0.651 $\pm$ 0.003 | 0.646 $\pm$ 0.010 | 0.175 $\pm$ 0.003 | 0.177 $\pm$ 0.009 |
> > > > | yes | 7 | ours | 0.686 $\pm$ 0.003 | 0.669 $\pm$ 0.013 | 0.171 $\pm$ 0.001 | 0.176 $\pm$ 0.009 |
> > > > | yes | 10 | ours | 0.703 $\pm$ 0.007 | 0.690 $\pm$ 0.017 | 0.165 $\pm$ 0.002 | 0.170 $\pm$ 0.011 |
> > > > | yes | 15 | ours | 0.725 $\pm$ 0.011 | 0.704 $\pm$ 0.007 | 0.158 $\pm$ 0.006 | 0.164 $\pm$ 0.009 |
> > > > | yes | 20 | ours | 0.739 $\pm$ 0.005 | 0.722 $\pm$ 0.014 | 0.152 $\pm$ 0.003 | 0.157 $\pm$ 0.010 |
> > > > | yes | 25 | ours | 0.742 $\pm$ 0.004 | 0.722 $\pm$ 0.014 | 0.150 $\pm$ 0.002 | 0.156 $\pm$ 0.011 |
> > > > | yes | 30 | ours | 0.745 $\pm$ 0.003 | 0.719 $\pm$ 0.014 | 0.149 $\pm$ 0.002 | 0.157 $\pm$ 0.010 |

---

> > > > > ### Comment · Reviewer_qYft · 2024-08-13
> > > > >
> > > > > I have read the additional comments and maintain my score.

---

### Official Review · Reviewer_GBdX · 2024-07-02

**Soundness:** 4
**Presentation:** 4
**Contribution:** 3
**Rating:** 9
**Confidence:** 4

**Summary:**

This paper explores the optimisation of the Cox model. Through careful mathematical analysis, the authors identified efficient ways to calculate the exact derivatives and surrogate loss functions necessary for efficient optimisation, addressing existing strategies' imprecision and time limitations.

**Strengths:**

**Clarity**
Despite not being familiar with the literature on optimising the Cox model, the problem is well-described, and the mathematical expressions are rigorous and well-detailed.

**Relevance**
The Cox model remains a pillar in multiple fields; ensuring its stability and quick convergence is a significant contribution.

**Weaknesses:**

Excellent paper, only a few minor typos and questions, see below.

**Questions:**

- It would be beneficial to add a pointer on the proof of convexity (line 82) of the nll.
- Doesn't the cumulative reverse sum require times to be sorted, i.e. an initial O(n log(n)) computation?
- Typo Line 195 - 'wise' should be 'wide'.

- In Appendix A1.1.1, the 3rd equation, should have for second element $\eta_i$ not $\eta_j$; then no difference between 4th and 5th

**Limitations:**

As an optimization paper, the authors mention that the work's limitations are the same as those of the standard Cox model.

---

> ### Author Rebuttal · Authors · 2024-08-07
>
> Thank you very much for reviewing our paper. Please see below for our answers to your questions.
>
> 1. **Proof of convexity of negative log likelihood (nll)**
>
> The essential part of the nll is the logSumExp function, which is defined as $f(\boldsymbol{x}) = \log(\sum_{i=1}^m \exp(x_m))$.
> It is well known the logSumExp function is convex (see Optimization Models and Applications by Laurent El Ghaoui, 2017).
>
> Moreover, the composition of a convex function with an affine (also linear) function is still convex (see the proof on Math Stack Exchange with question ID 654201). Therefore, $\log(\sum_{j \in R_i} e^{\boldsymbol{x}_j^T \boldsymbol{\beta}})$ is convex.
>
> Lastly, the sum of two convex functions is still a convex function.
> Therefore, $\sum_{i=1}^n \delta_i \left[ \log(\sum_{j \in R_i} e^{\boldsymbol{x}_j^T \boldsymbol{\beta}}) - \boldsymbol{x}_i^T \boldsymbol{\beta} \right]$ is convex.
>
> 2. **Is sorting needed during optimization?**
>
> Please look at our answer in the general response.
>
> 3. **Page 15, math typos associated with Appendix A.1.1**
>
> Please look at our answer in the general response.
>
> 4. **Minor writing typo**
>
> Thank you for your careful reading. We will fix this (wise $\rightarrow$ wide) during revision.

---

> > ### Comment · Reviewer_GBdX · 2024-08-12
> > **Maintaining score**
> >
> > Thank you for your answers, I am maintaining my score

---

### Official Review · Reviewer_GRak · 2024-07-05

**Soundness:** 4
**Presentation:** 4
**Contribution:** 3
**Rating:** 7
**Confidence:** 3

**Summary:**

This paper presents a new optimization algorithm for the Cox model, which is a classical algorithm for survival analysis presented in 1972.

**Strengths:**

This paper is well-written.  I am not an expert on optimization algorithms for convex functions, but I think I could understand the proposed algorithm and I enjoyed reading this paper.

The proposed algorithm converges faster than other methods due to the new idea of exploiting high order derivatives (up to the third order derivatives) in the Cox model, and it utilizes the surrogate functions for the second and third derivatives.  The proposed algorithm is reasonable and the experimental results are convincing.

**Weaknesses:**

As a researcher in the field of survival analysis, I would like to note that the impact of this paper is moderate within the field of survival analysis. As mentioned on page 9 of this paper, the Cox model is based on a strong assumption, the proportional hazard assumption, which is unlikely to hold in practice. Therefore, many state-of-the-art models have been developed using much weaker assumptions, and the Cox model is becoming obsolete. However, I would also like to note that the Cox model still shows a certain level of popularity thanks to its simplicity and interpretability.

Misc:
- Line 52: we are -> ours are
- Lines 93-94: the formula for quasi Newton method can be simplified by using the diag() function as in the formula for the proximal Newton method.
- Line 116: result result
- Line 124: a comma is missing
- Line 195: wise -> wide
- Line 257: it is better to clarify which correlation coefficient is used: Pearson, Spearman, or any other one?
- Page 15: nothing is changed in the statement "move the partial derivative operator"
- Lines 539-540: Redundant vertical lines in the inequality.

**Questions:**

N/A

**Limitations:**

Discussed in page 9.

---

> ### Author Rebuttal · Authors · 2024-08-07
>
> Thank you very much for reviewing our paper. Please see below for our answers to your questions.
>
> 1. **Utility of Cox Model**
>
>  We want to emphasize that we agree with what the reviewer has commented and simply want to continue this conversation. We think there are three aspects that make the Cox model still worthwhile and relevant:
>
> (1) The Cox model is relevant in high stakes domains such as medicine. In these domains, professionals need to interpret the results and understand feature importance. One way to achieve this is to obtain as small number of coefficients as possible without losing predictive performance. In Section 4.2, we have pushed the frontier of this variable selection task.
>
> (2) The Cox model is useful when served as a baseline to compare against more advanced and complicated models. If the more complicated model doesn't outperform the Cox model, we probably want to use the simpler Cox model, according to the principle of Occam's Razor.
>
> (3) Even if we do not deploy the Cox model in the end, it is still beneficial to use during the data exploration stage. When data scientists deal with messy real-world datasets for the first time, they want to perform assessments to see whether there are any abnormalities. It is cumbersome to explore if they use a much more complicated model. Moreover, fixing the abnormalities found by the Cox model could potentially boost the performance of the complicated one.
>
> We totally respect the reviewer's expertise in survival analysis. We agree with the reviewer that there exist survival models that are not based on the proportional harzard assumption. In Appendeix B Related Work, we have discussed some SOTA survival models. For example, OST and OSST are two decision tree methods. Besides the Cox model, there are also the accelerated failure time (AFT) model and the Aalen’s additive model. Lastly, other models include the ensemble method (SurvivalQuilts) and neural networks (DeepSurv). If you think there are other SOTA survival models we missed, please let us know, we are happy to cite them during the revision.
>
> 2. **Line 93-94, formula for quasi-Newton method**
>
> Thank you for this suggestion.
> We are afraid our current notation cannot be further simplified.
> In our definition, $\text{diag}(\cdot)$ (see Line 94) takes a vector as input and outputs a square matrix whose diagonal equals to this input and other entries should be $0$. However, $\nabla^2_{\boldsymbol{\eta}} \ell(\boldsymbol{\eta})$ is a square matrix.
> Thus, we cannot send $\nabla_{\boldsymbol{\eta}}^2 \ell(\boldsymbol{\eta})$ as input to $\text{diag}(\cdot)$.
> In the paper, we had some comments (in green color) to help clarify things.
> This was the most succinct notation we could come up. If you have another idea to simplify Line 93-94, please let us know, and we are very happy to incorporate that during revision.
>
> 3. **Line 257, definition of feature correlation**
>
> The correlation used in our work is neither based on Pearson nor Spearman. Please see Line 709-714 in Appendix C2 for the context where the feature correlation is used. For the synthetic dataset, the feature $\boldsymbol{x}_i$ is sampled from a Gaussian distribution:
>
> $\boldsymbol{x}_i \sim \mathcal{N}(0, \boldsymbol{\Sigma})$,
>
> where $\boldsymbol{\Sigma}$ is the the (Toeplitz) covariance matrix with $\Sigma_{jl} = \rho^{|j - l|}$, and $\rho \in (0, 1]$ is the correlation parameter.
> When $\rho$ is large, like $\rho=0.9$ in our paper, the features become highly correlated.
> This serve as a great benchmark in Section 4.2 to test how well different algorithms can recover the sparse coefficients.
> As you have seen, our new method significantly outperforms other methods.
>
> 4. **Page 15, math typos associated with Appendix A.1.1**
>
> Please look at our answer in the general response.
>
> 5. **Other minor writing typos**
>
>  We will fix them during revision. Thank you very much for point these out!

---

> > ### Comment · Reviewer_GRak · 2024-08-10
> >
> > Thank you for your comments. I will maintain my original score.

---

### Official Review · Reviewer_EDwF · 2024-07-11

**Soundness:** 3
**Presentation:** 3
**Contribution:** 3
**Rating:** 7
**Confidence:** 3

**Summary:**

The authors propose an optimization of the Cox proportional hazards model based on minimizing surrogate functions obtained from exploiting the Lipschitz continuity property of the first and second order partial derivatives of the loss wrt coefficients. The authors show that the optimization works for sparse penalties or constrained problems like using a cardinality constraint. The authors conduct experiments to show the efficiency and stability of the proposed optimization procedure and empirically show high performance against CoxPH with Newton-Raphson as well as non-linear predictors.

**Strengths:**

- The size of modern data sets wrt n and p can be a limiting factor for applying the CoxPH model, so innovation in optimization strategy is needed
- Strongly correlated variables can be an issue for solving the CoxPH model due to colinearity
- The method works well compared to standard CoxPH empirircally
- Introducing beamsearch for l0 regularization and covariate selection due to a well behaved loss function leads to interesting performance improvements for large data sets and prevents overfitting

**Weaknesses:**

- Comparison to an O(n) gradient descent framework like fastCPH (Yang et al) and BigSurvSGD would be helpful in the experiment section (including the lassonet implementation) to substantiate the claim that it is slow (in runtime) compared the proposed procedure and compare stability.
- When comparing performances wrt survival metrics (c-index, ibs), including the lasso version of the optimization model would be helpful to understand the exact difference between standard lasso cox and the proposed optimization
- A discussion of the tightness of the bound seems necessary when optimizing the bound instead of the loss function

**Questions:**

- With the proposed method, is it possible to obtain the variance-covariance matrix to construct confidence intervals for the coefficients? If so, the calibration of these would be interesting to look at.
- Do I understand it correctly that the survival times have to be sorted for the first and second derivative to be O(n) in the current implementation? Would that imply that the complexity of the algorithm is O(n log n)?
- In future outlook time-varying features are named as an application, does this amount to just changing the risk set in the derivatives as is the case for the loss function?

**Limitations:**

The authors have addressed some of the limitations of the model.

---

> ### Author Rebuttal · Authors · 2024-08-07
>
> Thank you very much for reviewing our paper. Please see below for our answers to your questions.
>
> 1. **Comparison with fastCPH and BigSurvSGD on computational efficiency**
>
> Thank you for pointing out these two baselines.
> We will cite them during revision.
>
> Regarding BigSurvSGD, below is a comparison between BigSurvSGD and our methods on Experiment 4.1.
> BigSurvSGD finishes running in a few seconds with low-quality solutions.
> To compare convergence speed, we set the time limit for our methods so that they run with the same or less amount of time than BigSurvSGD.
> We can see that our methods achieve much smaller training losses than bigSurvSGD within the same or less wall-clock time, especially using the cubic method.
> We also want to point out that we implement our methods in python (we loop through all coordinates sequentially) whereas BigSurvSGD is implemented in C++.
>
> | Method      | lambda1 | lambda 2 | train loss | time (s) |
> | ----------- | ----------- | ----------- | ----------- | ----------- |
> | bigSurvSGD | 0 | 1 | 12274 | 18.1 |
> | ours (quadratic) | 0 | 1 | 11054 | 4.2 |
> | ours (quadratic) | 0 | 1 | 10872 | 8.4 |
> | ours (quadratic) | 0 | 1 | 10775 | 16.7 |
> | ours (cubic) | 0 | 1 | 10760 | 4.0 |
> | ours (cubic) | 0 | 1 | 10707 | 7.9 |
> | ours (cubic) | 0 | 1 | 10690 | 15.7 |
> | - | - | - | - | - |
> | bigSurvSGD | 1 | 5 | 10859 | 17.6 |
> | ours (quadratic) | 1 | 5 | 11100 | 4.1 |
> | ours (quadratic) | 1 | 5 | 10935 | 7.6 |
> | ours (quadratic) | 1 | 5 | 10854 | 14.5 |
> | ours (cubic) | 1 | 5 | 10855 | 3.9 |
> | ours (cubic) | 1 | 5 | 10808 | 7.15 |
> | ours (cubic) | 1 | 5 | 10795 | 14.6 |
>
> Note that bigSurvSGD uses stochastic gradient descent.
> Stochasticity introduces random noise and does not ensure monotonic loss decrease.
> This is problematic if we want to use SGD inside beamsearch for the variable selection task in presence of highly correlated features.
> Moreover, for gradient descent, we have mentioned in Line 84-85 that it is hard to pick the right step size, which significantly impacts the convergence speed.
>
> Regarding FastCPH (LassoNetCoxRegressor from the lassonet package), this uses a neural network instead of a pure linear model.
> It is not allowed to specify no hidden layers for FastCPH (\# of neural network layers $>$2; see Line 19 in the file model.py in the lassonet package on GitHub).
> Therefore, we cannot use this package to train a linear Cox model.
>
> 2. **Comparison with lasso on solution quality**
>
> In Section 4.2, we have already included the lasso baseline.
> See the baseline sksurvCoxnet (the default $\ell_1$ ratio is $1.0$).
> During revision, we will make it clearer that skusrvCoxnet is the lasso baseline.
>
> 3. **Tightness of bounds when optimizing the bound instead of the loss function**
>
> At each iteration, all current methods (existing Newton methods or our methods) try to minimize a function that approximates the original loss function.
>
> **Existing Newton methods** They approximate the original loss function through a 2nd-order Taylor expansion (see Line 87-94).
> The exact Newton method computes the Hessian exactly while quasi and proximal Newton methods compute the Hessian approximately.
>
> **Our quadratic surrogate function** Here, we also make a 2nd-order Taylor expansion but with two differences.
> Firstly, we only do the approximation in a single coordinate.
> Secondly, the expansion's 2nd-order coefficient is an upper bound on the exact 2nd-order derivative.
> For tightness, if we use the exact 2nd-order derivative instead of the upper bound, we can get a better approximation, which we discuss next.
>
> **Our cubic surrogate function**
> Here, we make a 3rd-order Taylor expansion on a single coordinate.
> We use the exact 1st- and 2nd -order partial derivatives and also use an upper bound on the 3rd-order partial derivative.
> For tightness, we obtain a more accurate approximation of the Cox function than our quadratic surrogate function, *at the neighborhood of $x$*.
>
> Both of our methods ensure monotonic loss decrease and global convergence while existing Newton methods do not.
> Please see the proof sketch in our general response.
>
> 4. **Can we construct confidence intervals for coefficients?**
>
> This is not the goal of our paper, but yes, it is possible, *after the optimal solution is obtained*.
>
> After the optimal solution $\boldsymbol{\beta}^*$ is obtained, we first calculate the Hessian $\boldsymbol{H}$ with respect to the loss function.
> Next, we calculate the standard error vector $\boldsymbol{s} = \text{sqrt}\left[ (\boldsymbol{H}^{-1}).\text{diag}() \right]$, where $\boldsymbol{Z}.\text{diag}()$ means taking the diagonal of the square matrix $\boldsymbol{Z}$ as a vector.
> The confidence interval is then $\boldsymbol{\beta}^* \pm \alpha \boldsymbol{s}$, where $\alpha \in \mathbb{R}_+$ corresponds to different confidence levels.
> Please see the book In All Likelihood by Yudi Pawitan for details.
> Also, please see an implementation in the lifelines package on GitHub.
>
> 5. **Is sorting needed during optimization?**
>
> Please look at the our answer in the general response.
>
> 6. **How can this work be extended to time-varying features?**
>
> It takes more than preprocessing and passing a new dataset to our algorithm.
> Time-varying features are not our focus in this work, but we will try our best to answer this question.
> For details, please see the lifelines documentation page on this topic.
>
> For time-invariant features, we only care about time duration, and reverse cumulative sum works fine.
> However, for time-varying features, we cannot use the vanilla reverse cumulative sum.
> At different times, new features can enter the loss function (while old features exit).
> Our optimization techniques are still applicable, but we need to write a customized reverse cumulative sum function.

---

> > ### Comment · Reviewer_EDwF · 2024-08-07
> >
> > Thank you for the response, I have increased the score to accept.

---

### Author Rebuttal · Authors · 2024-08-07

We would like to thank the reviewers for their detailed reviews.
We will use this general response to address some common questions and concerns.

1. **Is sorting needed during optimization?**
Thanks to both Reviewer EDwF and Reviewer GBdX for asking this question.

We never perform sorting during any iterations.
We perform sorting \textbf{only once} at the beginning as a preprocessing step.
When sorting, we can either (1) rearrange the row orders of $\boldsymbol{X}$ and $\boldsymbol{y}$ or (2) record the sorting order and do reverse cumulative sum in this new order.
Thus, the complexity at each iteration is $O(n)$ instead of $O(n \text{log}(n))$. Well-known open-source GitHub packages (scikit-survival, lifelines, skglm, etc) also use sorting as a preprocessing step, so we do not take more preprocessing time than other methods.

2. **Page 15, math typos associated with Appendix A.1.1**

Thanks to both Reviewer GRak and Reviewer GBdX for finding the typos.

To fix them, we will delete the 3rd equation with the comment ''move the partial derivative operator'' since the line above this already moves the partial derivative operator inside.
Moreover, from the 3rd to the 8th equations, we will change $-\eta_j \Rightarrow - \eta_i$ and $\frac{\partial}{\partial \eta_{k_1}}(\eta_j) \Rightarrow \frac{\partial}{\partial \eta_{k_1}}(\eta_i)$.
These typos do not affect the end result.
We will correct them during revision.

3. **Monotonic Decrease and Global Convergence**

Reviewer qYft asked this question, but the answer here also complements our reply to Reviewer EDwF regarding the tightness of our surrogate functions (bounds).

a. **monotonic loss decrease**:
We use the quadratic surrogate function $g_x(\Delta x)$ defined in Equation (15) to illustrate this.
The reasoning also applies to the cubic surrogate function $h_x(\Delta x)$ defined in Equation (16).
From Equation (15), we know two facts:

$$f(x + \Delta x) \leq g_x(\Delta x) \text{ for any } \Delta x, \; \text{ and } f(x) = g_x(0).$$

If we define $\Delta \tilde{x} := \text{argmin}_{\Delta x} g_x(\Delta x)$.
Then we have the inequalities below:

$$f(x + \Delta \tilde{x}) \leq g_x(\Delta \tilde{x}) \leq  g_x(0) = f(x).$$

Thus, we have $f(x + \Delta \tilde{x}) \leq f(x)$.
Since $\Delta \tilde{x}$ is the step we take to minimize $g_x(\cdot)$, our original loss function $f(\cdot)$ will decrease monotonically.

b. **global convergence**
We use the Monotone Convergence Theorem (MCT) from basic real analysis to prove convergence.
Let $\{x^t\}$ and $\{f(x^t)\}$ be the sequence of solutions and loss values generated by the iterative procedure, where $t$ is the number of iterations.
From last part, we know that the sequence $\{f(x^t)\}$ is monotonically decreasing.
Furthermore, this sequence is bounded below by $f(x^*)$, where $x^*$ is the optimal solution.
Then, by the MCT, the sequence $\{f(x^t)\}$ converges.

c. **Converging to the optimal value**
We show the sequence $\{f(x^t)\}$ will converge to the optimal value $f(x^*)$.
Because the Cox function $f(\cdot)$ is continuous, convergence of $\{f(x^t)\}$ implies the convergence of $\{x^t\}$.
If we define $\Delta x^t := x^{t+1} - x^{t}$, this implies that  $\Delta x^t \xrightarrow{t \to \infty} 0$.
From Equation (17) or Equation (18), we can  deduce that the first order partial derivatives $f'(x^t)$ converges to 0, for every coordinate.
Note that this is the optimality condition for a convex function.
Thus, we have $\lim_{t \rightarrow \infty} f(x^t) = f(x^*)$.

---

### Decision · Program_Chairs · 2024-09-25

**Decision:**

Accept (poster)

**Comment:**

Cox proportional hazards models are a mainstay of statistical modeling in healthcare and beyond. This paper proposes a simple but important improvement to the stability and speed of optimizing these models, and so represents an important advance in applied data science.